# Demystifying Online Clustering of Bandits: Enhanced Exploration Under Stochastic and Smoothed Adversarial Contexts

**Zhuohua Li**[1,2]***Maoli Liu**[2]***Xiangxiang Dai**[2]**, John C.S. Lui**[2]
[1]Guangzhou Institute of Technology, Xidian University
[2]Department of Computer Science and Engineering, The Chinese University of Hong Kong
{zhli,mlliu,xxdai23,cslui}@cse.cuhk.edu.hk

## Abstract

The contextual multi-armed bandit (MAB) problem is crucial in sequential decision-making. A line of research, known as online clustering of bandits, extends contextual MAB by grouping similar users into clusters, utilizing shared features to improve learning efficiency. However, existing algorithms, which rely on the upper confidence bound (UCB) strategy, struggle to gather adequate statistical information to accurately identify unknown user clusters. As a result, their theoretical analyses require several strong assumptions about the "diversity" of contexts generated by the environment, leading to impractical settings, complicated analyses, and poor practical performance. Removing these assumptions has been a long-standing open problem in the clustering of bandits literature. In this work, we provide two partial solutions. First, we introduce an additional exploration phase to accelerate the identification of clusters. We integrate this general strategy into both graph-based and set-based algorithms and propose two new algorithms, `UniCLUB` and `UniSCLUB`. Remarkably, our algorithms require substantially weaker assumptions and simpler theoretical analyses while achieving superior cumulative regret compared to previous studies. Second, inspired by the smoothed analysis framework, we propose a more practical setting that eliminates the requirement for *i.i.d.* context generation used in previous studies, thus enhancing the performance of existing algorithms for online clustering of bandits. Extensive evaluations on both synthetic and real-world datasets demonstrate that our proposed algorithms outperform existing approaches.

## 1 Introduction

The stochastic multi-armed bandit (MAB) problem is an online sequential decision-making problem, where at each time step, the learner selects an action (a.k.a. arm) and observes a reward generated from an unknown probability distribution associated with that arm. The goal of the learner is to maximize cumulative rewards (or equivalently, minimize cumulative regrets) in the long run. The contextual linear bandit problem (Li et al., 2010; Chu et al., 2011) extends the MAB framework by associating each action with a feature vector and a corresponding unknown linear reward function.

*Online clustering of bandits*, first introduced by Gentile et al. (2014), generalizes contextual linear bandits by utilizing preference relationships among users. It adaptively partitions users into clusters and leverages the collaborative effect of similar users to enhance learning efficiency. This approach has many applications in computational advertising, web page content optimization, and recommendation systems (Li & Zhang, 2018). Different from conventional MAB problems that focus solely on regret minimization, online clustering of bandits has two *simultaneous* goals. Firstly, it infers the underlying cluster structures among users by sequentially recommending arms and receiving user feedback. Secondly, based on the inferred clusters, it minimizes the cumulative regret along

---

*Zhuohua Li and Maoli Liu contributed equally to this work. Maoli Liu is the corresponding author.
An updated version of this paper is available on arXiv at https://arxiv.org/abs/2501.00891.

the learning trajectory. These dual goals significantly influence the algorithm design, as the learner must balance the accurate cluster inference and effective regret minimization.

Most existing studies such as Gentile et al. (2014); Li & Zhang (2018) employ an Upper Confidence Bound (UCB)-based strategy (Abbasi-Yadkori et al., 2011) to balance exploration (for cluster inference) and exploitation (for regret minimization). While this strategy is intuitive and standard for stochastic linear bandits, the least squares estimator used in the UCB strategy does not directly yield a precise estimation of underlying parameters, leading to insufficient statistical information for cluster inference. As a result, to ensure correct cluster inference, existing algorithms for clustering of bandits require several strong assumptions on "data diversity" for their regret analysis, such as contexts being independently generated from a fixed random process (we refer to it as stochastic context setting) with lower bounded minimum eigenvalue of a covariance matrix and upper bounded variance (See Section 3.2 for details). Unfortunately, these assumptions result in impractical settings, overly complicated theoretical analysis, suboptimal regret incurred by cluster inference, and more importantly, poor performance in practice. Based on these challenges, a natural question arises:

> *Can we design new algorithms or propose new settings for the online clustering of bandits that eliminate the restrictive assumptions while achieving similar regrets?*

Several efforts have been made to relax these assumptions. However, previous attempts have not addressed the fundamental issues, and some have resulted in deteriorated regret that grows exponentially with the number of arms, in exchange for milder assumptions (See details in Section 2).

In this paper, we demystify the process of cluster inference and provide two (partial) solutions to the aforementioned open problem. We show that merely employing a UCB algorithm is insufficient for simultaneously achieving both cluster inference and regret minimization. Specifically, the lack of sufficient exploration in a UCB-like strategy hinders efficient inference of the underlying cluster structures. To tackle this problem, we propose two approaches, one focusing on algorithmic design and the other on problem setup:

**Algorithmic Design Perspective**: We maintain the stochastic context setting as in Gentile et al. (2014) (with some minor changes, see Section 3.1), and propose new algorithms with an additional pure exploration phase before applying UCB, leading to more reasonable settings, significantly relaxed assumptions, simpler proofs, and improved cumulative regret. Intuitively, the additional exploration phase gathers more information about the underlying clusters, preventing the subsequent UCB strategy from exploiting inaccurate clusters. This technique is quite general and applicable to both graph-based (Gentile et al., 2014) and set-based (Li et al., 2019) algorithms. It may also be of independent interest to other research on multi-objective MAB problems.

**Problem Setup Perspective**: We eliminate the need for stochastic context generation by proposing a new setup based on the smoothed analysis framework (Spielman & Teng, 2004), where the contexts can be chosen by a "smoothed" adversary. This setting interpolates between two extremes: the *i.i.d.* context generation used in Gentile et al. (2014) and the adversarial context generation used in Abbasi-Yadkori et al. (2011). We show that with some minor changes, existing algorithms (such as CLUB proposed by Gentile et al. (2014)) achieve better performance in this setting.

To the best of our knowledge, our work is the first to clarify the inferior results due to the restrictive assumptions in the clustering of bandits literature (Gentile et al., 2014), propose new algorithms/settings to eliminate the assumptions, and obtain improved cumulative regrets.

**Main contributions**. Our contributions are highlighted as follows:

- Following the stochastic context setting established in Gentile et al. (2014), we propose two algorithms: a graph-based algorithm `UniCLUB` based on CLUB (Gentile et al., 2014) and a set-based algorithm `UniSCLUB` based on SCLUB (Li et al., 2019). Both algorithms incorporate an initial exploration phase before the conventional UCB strategy. Benefiting from our design and novel analysis techniques, we substantially relax the assumptions required in our theoretical analysis.

- We show that `UniCLUB` and `UniSCLUB` enjoy a regret bound of $\widetilde{O}\left(u\frac{d}{\widetilde{\gamma}^2\lambda_x} + d\sqrt{mT}\right)$, where the first term improves the state-of-the-art regret in existing literature (Gentile et al., 2014; Li & Zhang, 2018; Li et al., 2019; Liu et al., 2022; Wang et al., 2023a;b), and the second term matches the minimax near-optimal regret bound of contextual linear bandits (Abbasi-Yadkori et al., 2011).

- Besides the stochastic context setting, we propose a new setting called the smoothed adversarial context setting, where in each round, the context is chosen by an adversary but is then randomly perturbed. This setup is more practical and aligns more closely with the original setting of contextual linear bandits (Abbasi-Yadkori et al., 2011). We give two associated algorithms SACLUB, SASCLUB and prove that they enjoy a regret of $\widetilde{O}\left(u\frac{d}{\gamma^2\widetilde{\lambda}_x} + d\sqrt{mT}\right)$ where $\widetilde{\lambda}_x = O\left(\frac{1}{\log K}\right)$.

- We perform extensive evaluations on both synthetic and real-world datasets. The results demonstrate that our algorithms outperform all the baseline algorithms, validating their effectiveness and practical applicability in various settings.

Table 1 summarizes the comparison between our algorithms and existing studies with different assumptions and cumulative regrets. Detailed explanations of the diversity conditions and regret analysis are given in Section 3.2 and Section 5.

Table 1: Comparison between our algorithms and existing studies with different design choices.

| Algorithms | Context Generation | Diversity Assumption | Regret incurred by clustering | Constants |
|---|---|---|---|---|
| CLUB (Gentile et al., 2014), CLUB-cascade (Li & Zhang, 2018), SCLUB (Li et al., 2019), FCLUB (Liu et al., 2022) | i.i.d. | $\lambda_{\min}(\mathbb{E}[\boldsymbol{X}\boldsymbol{X}^{\mathsf{T}}]) = \lambda_x$, $(\boldsymbol{z}^{\mathsf{T}}\boldsymbol{X})^2$ is $\sigma^2$-sub-Gaussian, $\sigma^2 \leq \frac{\lambda_x^2}{8\log(4K)}$. | $\widetilde{O}\left(u\left(\frac{d}{\gamma^2\lambda_x} + \frac{1}{\lambda_x^2}\right)\right)$ | $\lambda_x = O(\frac{1}{d})$ |
| RCLUMB, RSCLUMB (Wang et al., 2023a), RCLUB-WCU (Wang et al., 2023b), FedC³UCB-H (Yang et al., 2024) | i.i.d. | $\lambda_{\min}(\mathbb{E}[\boldsymbol{X}\boldsymbol{X}^{\mathsf{T}}]) = \lambda_x$, $(\boldsymbol{z}^{\mathsf{T}}\boldsymbol{X})^2$ is $\sigma^2$-sub-Gaussian. | $\widetilde{O}\left(u\left(\frac{d}{\gamma^2\widetilde{\lambda}_x} + \frac{1}{\widetilde{\lambda}_x^2}\right)\right)$ | $\widetilde{\lambda}_x = O(\frac{1}{d^{2K+1}})$ |
| UniCLUB, UniSCLUB (Ours, Theorems 1 and 2) | i.i.d. | $\lambda_{\min}(\mathbb{E}[\boldsymbol{X}\boldsymbol{X}^{\mathsf{T}}]) = \lambda_x$, The minimum gap between clusters $\gamma$ has a known lower bound $\widetilde{\gamma}$. | $\widetilde{O}\left(\frac{ud}{\widetilde{\gamma}^2\lambda_x}\right)$ | $\lambda_x = O(\frac{1}{d})$ |
| SACLUB, SASCLUB (Ours, Theorem 4) | Adversarial | Gaussian noise perturbation | $\widetilde{O}\left(\frac{ud}{\gamma^2\lambda_x}\right)$ | $\widetilde{\lambda}_x = O\left(\frac{1}{\log K}\right)$ |

$K$, $u$, and $d$ denote the number of arms, the number of users, and the dimension, respectively. $\gamma$ and $\widetilde{\gamma}$ are defined in Assumption 2. $\boldsymbol{X}$ and $\lambda_x$ are defined in Assumption 3. $\boldsymbol{z} \in \mathbb{R}^d$ represents an arbitrary unit vector.

The rest of the paper is organized as follows. In Section 2, we review related work, emphasizing the restrictive assumptions in the original setting and discussing prior attempts to eliminate these assumptions. In Section 3, we present our solution for removing these assumptions in the original setting. In Section 4, we introduce an alternative approach based on the smoothed analysis framework, which aligns more closely with the original linear bandits setting (Abbasi-Yadkori et al., 2011). Finally, in Section 5 and Section 6, we present the theoretical analysis and experiment results.

## 2 RELATED WORK

Our work is closely related to the literature of online clustering of bandits. Since the seminal work by Gentile et al. (2014), which first formulated the clustering of bandits problem and proposed a graph-based algorithm, there has been a line of follow-up studies. For example, Li et al. (2016) considers the collaborative effects that arise from user-item interactions. Gentile et al. (2017) implements the underlying feedback-sharing mechanism by estimating user neighborhoods in a context-dependent manner. Li & Zhang (2018) consider the clustering of bandits problem in the cascading bandits setting with random prefix feedback. Li et al. (2019) propose a set-based algorithm and consider users with non-uniform arrival frequencies. Ban & He (2021) introduce local clustering which does not assume that users within the same cluster share exactly the same parameter. Liu et al. (2022) extend the clustering of bandits problem to the federated setting and consider privacy preservation. Wang et al. (2023a;b) investigate clustering of bandits under misspecified and corrupted user models.

However, all of these studies adhere to Gentile et al. (2014)'s original setting and theoretical analysis framework, which imposes several restrictive assumptions on the generation process of contexts, such as (a) the feature vector of each arm is independently sampled from a fixed distribution; (b) the covariance matrix constructed on specific context-action features is full rank with the minimum eigenvalue greater than 0; and (c) the square of contexts projected in a fixed direction is sub-Gaussian with bounded conditional variance. Constructing a natural context generation distribution that satisfies all these assumptions simultaneously is *highly challenging* (if not impossible), and none of

the aforementioned papers provide any concrete examples. There have been some attempts to relax these assumptions. For example, Wang et al. (2023a;b); Yang et al. (2024) propose a more relaxed assumption regarding the variance of contexts. However, these approaches result in a regret that grows exponentially with the number of arms $K$ (as shown in Table 1). We refer interested readers to more related works about leveraging similar assumptions in Appendix A.

Our smoothed adversarial setting and algorithms SACLUB, SASCLUB are inspired by the smoothed analysis framework, introduced by Spielman & Teng (2004). This framework studies algorithms where some instances are chosen by an adversary, but are then perturbed randomly, representing an interpolation between worst-case and average-case analyses. It was originally proposed to analyze the running time of algorithms. Kannan et al. (2018) first introduce the "smoothed adversary" setting in multi-armed bandits and study how the regret bound of greedy algorithms behave on smoothed bandit instances. This setting has been extended to structured linear bandits (i.e., the unknown preference vector has structures such as sparsity, group sparsity, or low rank) (Sivakumar et al., 2020), linear bandits with knapsacks (Sivakumar et al., 2022), and Bayesian regret (Raghavan et al., 2023). All these studies show that the greedy algorithm almost matches the best possible regret bound. The core idea is that the inherent diversity in perturbed data (contexts) makes explicit exploration unnecessary. In contrast to the analysis of greedy algorithms, our work introduces the smoothed analysis framework into the clustering of bandits setting, where the UCB strategy lacks sufficient exploration for identifying unknown user clusters. We demonstrate that the inherent diversity of contexts in the smoothed adversary setting eliminates the impractical requirement for *i.i.d.* context generation and enhances the cumulative regrets of existing algorithms (e.g., CLUB (Gentile et al., 2014)).

## 3 STOCHASTIC CONTEXT SETTING

In this section, we study the online clustering of bandits problem under the stochastic context setting, building upon the seminal work of Gentile et al. (2014) but with substantially relaxed assumptions and improved cumulative regrets. We begin by introducing the problem setting, which largely follows Gentile et al. (2014) with the stringent assumptions replaced by a mild assumption that the minimum gap between clusters has a known lower bound. Then we provide the intuition of our key techniques. Finally, we present our proposed algorithms.

### 3.1 PROBLEM SETTING

In the following, we use boldface letters for vectors and matrices. We denote $[M] := \{1, \ldots, M\}$ for $M \in \mathbb{N}^+$. For any real vector $\boldsymbol{x}, \boldsymbol{y}$ and positive semi-definite (PSD) matrix $\boldsymbol{V}$, $\|\boldsymbol{x}\|$ denotes the $\ell_2$ norm of $\boldsymbol{x}$, $\langle \boldsymbol{x}, \boldsymbol{y} \rangle = \boldsymbol{x}^\mathsf{T} \boldsymbol{y}$ denotes the dot product of vectors, $\langle \boldsymbol{x}, \boldsymbol{y} \rangle_{\boldsymbol{V}} = \boldsymbol{x}^\mathsf{T} \boldsymbol{V} \boldsymbol{y}$ denotes the weighted inner product, and $\|\boldsymbol{x}\|_{\boldsymbol{V}}$ denotes the Mahalanobis norm $\sqrt{\boldsymbol{x}^\mathsf{T} \boldsymbol{V} \boldsymbol{x}}$. We use $\lambda_{\min}(\cdot)$ and $\lambda_{\max}(\cdot)$ to denote the minimum and maximum eigenvalue.

In the online clustering of bandit problem, there are $u$ users, denoted by set $[u] = \{1, 2, \ldots, u\}$. Each user $i \in [u]$ is associated with an *unknown* preference feature vector $\boldsymbol{\theta}_i \in \mathbb{R}^d$ with $\|\boldsymbol{\theta}_i\|_2 \leq 1$. There is an underlying cluster structure among all the users. Specifically, the users are separated into $m$ clusters $\mathcal{I}_1, \mathcal{I}_2, \ldots, \mathcal{I}_m$ ($m \ll u$), where $\bigcup_{i \in [m]} \mathcal{I}_i = [u]$ and $\mathcal{I}_i \cap \mathcal{I}_j = \emptyset$ for $i \neq j$, such that users lying in the same cluster share the same preference feature vector (i.e., they have similar behavior) and users lying in different clusters have different preference feature vector (i.e., they have different behavior). Formally, let $\boldsymbol{\theta}^k$ denote the common preference vector for cluster $\mathcal{I}_k$ and $j(i) \in [m]$ denote the index of the cluster that user $i$ belongs to. In other words, for any user $i \in \mathcal{I}_k$, we have $\boldsymbol{\theta}_i = \boldsymbol{\theta}^k = \boldsymbol{\theta}^{j(i)}$. The underlying partition of users and the number of clusters $m$ are *unknown* to the learner, and need to be learned during the algorithm.

The learning procedure operates as follows: At each round $t = 1, 2, \ldots, T$, the learner receives a user index $i_t \in [u]$ and a finite set of arms $\mathcal{A}_t \subseteq \mathcal{A}$ where $|\mathcal{A}_t| = K$. Each arm $a \in \mathcal{A}$ is associated with a feature vector $\boldsymbol{x}_a \in \mathbb{R}^d$, and we denote $\mathcal{D}_t = \{\boldsymbol{x}_a\}_{a \in \mathcal{A}_t} \subseteq \mathbb{R}^d$. $\mathcal{D}_t$ is also called *context*. When we need to emphasize the index of arms, we also denote $\mathcal{A}_t = \{a_{t,i}\}_{i=1}^K$ and $\mathcal{D}_t = \{\boldsymbol{x}_{t,i}\}_{i=1}^K$. Then the learner assigns an appropriate cluster $V_t$ for user $i_t$ and recommends an arm $a_t \in \mathcal{A}_t$ based on the aggregated data gathered from cluster $V_t$. After receiving the recommended arm, user $i_t$ sends a random reward $r_t \in [-1, 1]$ back to the learner. The reward is assumed to have a linear structure: $r_t = \boldsymbol{x}_{a_t}^\mathsf{T} \boldsymbol{\theta}_{i_t} + \eta_t$, where $\eta_t$ is a zero-mean, 1-sub-Gaussian noise term.

Let $a_t^* = \arg\max_{a \in \mathcal{A}_t} \boldsymbol{x}_a^\mathsf{T} \boldsymbol{\theta}_{i_t}$ be the optimal arm with the highest expected reward at time step $t$. The goal of the learner is to minimize the expected cumulative regret defined as follows:

$$\mathbb{E}[R(T)] = \mathbb{E}\left[\sum_{t=1}^{T}\Big(\boldsymbol{x}_{a_t^*}^\mathsf{T}\boldsymbol{\theta}_{i_t} - \boldsymbol{x}_{a_t}^\mathsf{T}\boldsymbol{\theta}_{i_t}\Big)\right],$$

where the expectation is taken over both the arms $(a_t)$ and users $(i_t)$ chosen during the process.

We assume the users and clusters satisfy the assumptions as follows:

**Assumption 1** (User uniformness). At each time step $t$, the user $i_t$ is drawn uniformly from the set of all users $[u]$, independently over the past.

**Assumption 2** (Well-separatedness among clusters). All users in the same cluster $\mathcal{I}_j$ share the same preference vector $\boldsymbol{\theta}^j$. For users in different clusters, there is a fixed but *unknown* gap $\gamma$ between their preference vectors, with a *known* lower bound $\widetilde{\gamma}$. Specifically, for any cluster indices $i \neq j$,

$$\left\|\boldsymbol{\theta}^i - \boldsymbol{\theta}^j\right\|_2 \geq \gamma \geq \widetilde{\gamma} > 0.$$

**Remark 1.** Unlike the original setting in Gentile et al. (2014), where the gap $\gamma$ is *completely unknown*, we assume that $\gamma$ has a *known* lower bound $\widetilde{\gamma}$. We emphasize that this adjustment is natural and results in a more practical setting. On the one hand, the clustering of bandits problem intrinsically requires a lower bound for $\gamma$. Specifically, to ensure successful clustering, it is necessary to have $\frac{1}{\gamma^2} \ll T$, where $T$ is the total time horizon. Without this condition, no algorithm can effectively identify the underlying clusters within time $T$.[1] On the other hand, in real-world clustering tasks, $\gamma$ is usually a pre-defined parameter used to determine whether two items belong to the same cluster (Ban & He, 2021; Ban et al., 2022). Therefore, incorporating a known lower bound for $\gamma$ is both reasonable and aligned with practical scenarios.

**Remark 2.** We also consider the case where $\gamma$ is completely unknown. In this scenario, without the stringent assumptions used in previous studies, algorithms are unable to determine whether clustering has been successful. As a result, we have to deal with the additional regret incurred by misclustering, similar to the misspecified linear bandits setting (Ghosh et al., 2017), which typically leads to regret that grows linearly with $T$ (Lattimore et al., 2020). To overcome this challenge, we leverage the structure of the clustering problem and propose a phase-based algorithm, `PhaseUniCLUB`, with carefully designed phase lengths. We demonstrate that our algorithm achieves sublinear regret of order $\widetilde{O}(T^{\frac{2}{3}})$ (see details in Appendix D).

In the stochastic context setting, we assume the feature vectors (i.e., contexts) are independently sampled from a fixed distribution, but we completely remove the restricted assumptions on the sub-Gaussian distribution and variance of the arm generation process as mentioned in prior works.

**Assumption 3** (Context diversity for stochastic contexts). At each time step $t$, the feature vectors in $\mathcal{D}_t$ are drawn independently from a fixed distribution $\boldsymbol{X}$ with $\|\boldsymbol{X}\| \leq L$, and $\mathbb{E}[\boldsymbol{X}\boldsymbol{X}^\mathsf{T}]$ is of full rank with minimum eigenvalue $\lambda_x > 0$.

**Remark 3.** Intuitively, the minimum eigenvalue indicates how "diverse" the distribution $\boldsymbol{X}$ is, depicting how certain the feature vectors span the full $\mathbb{R}^d$ space. Having a lower bound on the minimum eigenvalue means that $\boldsymbol{X}$ has non-zero variance in all directions, which is necessary for the least squares estimator to converge to the true parameter. Note that Assumption 3 only maintains the minimum eigenvalue assumption[2] and *completely* removes the additional stringent assumptions as in previous studies (Gentile et al., 2014; Li & Zhang, 2018; Li et al., 2019; Wang et al., 2023a;b; Liu et al., 2022; Yang et al., 2024).

## 3.2 DIVERSITY CONDITIONS IN PREVIOUS STUDIES AND KEY TECHNIQUES

Before delving into detailed algorithms, we first examine the stringent statistical assumptions in previous studies (Gentile et al., 2014; Li & Zhang, 2018; Li et al., 2019; Wang et al., 2023a;b; Liu

---

[1]To distinguish between two hypotheses about the bias of a coin with a difference $\Delta$ in their probabilities of heads, approximately $1/\Delta^2$ samples are needed.

[2]This assumption is inevitable, since if the minimum eigenvalue is zero, the covariance matrix is not of full rank, and thus $\boldsymbol{\theta}$ cannot be uniquely determined.

et al., 2022; Yang et al., 2024) and explain their necessity for the theoretical analysis of the existing UCB-based algorithms. Then we provide insights on how these assumptions can be eliminated.

The key requirement of online clustering of bandits is the precise estimation of the preference vectors $\boldsymbol{\theta}_i$ for each user $i$, which is essential for correctly identifying the unknown user clusters. However, the convergence of the least squares estimator relies on sufficiently diverse data. Intuitively, when data points span a broad range of values and cover the spectrum of possible predictors, the model can better capture the true underlying relationships, leading to more reliable parameter estimates. Mathematically, diverse data help ensure that user $i$'s design matrix $\boldsymbol{S}_{i,t} = \sum_{s \in [t]:i_s=i} \boldsymbol{x}_{a_s} \boldsymbol{x}_{a_s}^\top$ is well-conditioned, resulting in a more stable matrix inverse, which in turn reduces the variance of the estimated preference vector $\widehat{\boldsymbol{\theta}}_i = \boldsymbol{S}_{i,t}^{-1} \sum_{s \in [t]:i_s=i} r_s \boldsymbol{x}_{a_s}$. Therefore, all the previous studies rely on the diverse stochastic context assumption (Assumption 3), which states that the feature vector of each arm is drawn independently from a fixed distribution $\boldsymbol{X}$ with $\lambda_{\min}(\mathbb{E}[\boldsymbol{X}\boldsymbol{X}^\top]) = \lambda_x > 0$. However, diverse contexts do not necessarily lead to a well-conditioned design matrix because *Assumption 3 only guarantees the diversity of arm set $\mathcal{D}_t$, but not that the arms selected by the UCB strategy are diverse*. As a result, previous studies impose an additional assumption, requiring that for any fixed unit vector $\boldsymbol{z} \in \mathbb{R}^d$, random variable $(\boldsymbol{z}^\top \boldsymbol{X})^2$ has sub-Gaussian tails with variance parameter $\sigma^2 \leq \frac{\lambda_x^2}{8\log(4K)}$. This assumption, however, contradicts the diverse stochastic context assumption (Assumption 3), as the bounded variance condition restricts the diversity of $\boldsymbol{X}$. In fact, it is extremely difficult to construct a natural example of $\boldsymbol{X}$ such that all these assumptions are satisfied simultaneously, and the aforementioned papers also do not provide any.

In summary, the key insight for eliminating the additional assumption is to ensure a well-conditioned design matrix $\boldsymbol{S}_{i,t}$, i.e., the selected arms are sufficiently diverse. To this end, we introduce an additional pure exploration phase which uniformly selects arms in the arm set $\mathcal{D}_t$. In Lemma 2, we will show that this explicit exploration guarantees that the minimum eigenvalue of the design matrix $\boldsymbol{S}_{i,t}$ grows linearly with the number of times user $i$ appears.

## 3.3 ALGORITHMS FOR STOCHASTIC CONTEXT SETTING

For the stochastic context setting, we introduce two algorithms: a graph-based algorithm called Uniform Exploration Clustering of Bandits (UniCLUB, Algorithm 1) and a set-based algorithm called Uniform Exploration Set-based Clustering of Bandits (UniSCLUB, Algorithm 2). Due to space constraints, we focus on UniCLUB in the main paper, leaving the details and regret analysis of UniSCLUB in Appendix C.

As shown in Algorithm 1, inspired by the CLUB algorithm proposed in Gentile et al. (2014), UniCLUB maintains a dynamic undirected graph $G_t = ([u], E_t)$ representing the current estimated cluster structures of all users. The main difference is that UniCLUB includes an additional uniform exploration phase to promote cluster identification. At the beginning, $G_t$ is initialized as a complete graph, indicating that all users are considered in a single cluster. Then at each round $t$, a user $i_t \in [u]$ comes to be served with a feasible arm set $\mathcal{A}_t$ from which the learner has to choose. The algorithm operates in the following two phases depending on whether the current time step $t \leq T_0$, and the arm selection strategy differs between these phases.

**Pure exploration phase**. In the first $T_0$ rounds, the algorithm uniformly select arm $a_t$ from $\mathcal{A}_t$ (Line 4). This arm selection strategy ensures selecting sufficiently *diverse* arms so that the minimum eigenvalue of the design matrix grows linearly in time (Lemma 2). In Lemma 3, we will show that the phase length $T_0$ is chosen to guarantee that this phase gathers sufficient statistics to estimate each user's preference vector and correctly infer the underlying user clusters with high probability.

**Exploration-exploitation phase**. After $T_0$, the algorithm constructs the connected component $V_t$ containing user $i_t$ in the graph $G_{t-1}$ (Line 6), and computes the estimated preference vector $\widehat{\boldsymbol{\theta}}_{V_t,t-1}$ based on historical information associated with $V_t$ using the least squares estimator with regularization parameter $\lambda > 0$ (Line 8). The algorithm then recommends an arm using the upper confidence bound (UCB) strategy (Abbasi-Yadkori et al., 2011) to balance exploration and exploitation (Line 9):

$$a_t = \arg\max_{a \in \mathcal{A}_t} \widehat{\boldsymbol{\theta}}_{V_t,t-1}^\top \boldsymbol{x}_a + \beta \sqrt{\boldsymbol{x}_a^\top \overline{\boldsymbol{M}}_{V_t,t-1}^{-1} \boldsymbol{x}_a},$$

---

**Algorithm 1:** `UniCLUB`: Uniform Exploration Clustering of Bandits

---

**Input:** $\lambda, \beta, \lambda_x, \delta, L, \widetilde{\gamma}$
**Initialization:** Let $G_0 = ([u], E_0)$ be a complete graph.
Let $\boldsymbol{S}_{i,0} = \boldsymbol{0}_{d \times d}, \boldsymbol{b}_{i,0} = \boldsymbol{0}_{d \times 1}, T_{i,0} = 0, \forall i \in [u]$.
Define $f(T_{i,t}) \triangleq (\sqrt{2 \log(u/\delta) + d \log(1 + \frac{T_{i,t}L^2}{\lambda d})} + \sqrt{\lambda}) / \sqrt{\lambda + T_{i,t}\lambda_x/2}$.
Define $T_0 \triangleq 16u \log\left(\frac{u}{\delta}\right) + 4u \max\left\{ \frac{8L^2}{\lambda_x} \log\left(\frac{ud}{\delta}\right), \frac{512d}{\widetilde{\gamma}^2 \lambda_x} \log\left(\frac{u}{\delta}\right) \right\}$.

1 **for** $t = 1, 2, \ldots$ **do**
2      Receive user index $i_t$ and arm set $\mathcal{A}_t$
3      **if** $t \leq T_0$ **then**
4          Select $a_t$ uniformly at random from $\mathcal{A}_t$
5      **else**
6          Find the connected component $V_t$ for $i_t$ in $G_{t-1}$
7          $\boldsymbol{M}_{V_t,t-1} = \sum_{i \in V_t} \boldsymbol{S}_{i,t-1}, \overline{\boldsymbol{M}}_{V_t,t-1} = \lambda \boldsymbol{I} + \boldsymbol{M}_{V_t,t-1}, \boldsymbol{b}_{V_t,t-1} = \sum_{i \in V_t} \boldsymbol{b}_{i,t-1}$
8          $\widehat{\boldsymbol{\theta}}_{V_t,t-1} = \overline{\boldsymbol{M}}_{V_t,t-1}^{-1} \boldsymbol{b}_{V_t,t-1}$
9          Select arm $a_t = \arg\max_{a \in \mathcal{A}_t} \widehat{\boldsymbol{\theta}}_{V_t,t-1}^\mathsf{T} \boldsymbol{x}_a + \beta \sqrt{\boldsymbol{x}_a^\mathsf{T} \overline{\boldsymbol{M}}_{V_t,t-1}^{-1} \boldsymbol{x}_a}$
10      Receive reward $r_t$
11      Update statistics for user $i_t$, others remain unchanged:
         $\boldsymbol{S}_{i_t,t} = \boldsymbol{S}_{i_t,t-1} + \boldsymbol{x}_{a_t} \boldsymbol{x}_{a_t}^\mathsf{T}, \quad \boldsymbol{b}_{i_t,t} = \boldsymbol{b}_{i_t,t-1} + r_t \boldsymbol{x}_{a_t}$
         $T_{i_t,t} = T_{i_t,t-1} + 1, \quad \widehat{\boldsymbol{\theta}}_{i_t,t} = (\lambda \boldsymbol{I} + \boldsymbol{S}_{i_t,t})^{-1} \boldsymbol{b}_{i_t,t}$
12      Delete edge $(i_t, \ell) \in E_{t-1}$ if

$$\left\| \widehat{\boldsymbol{\theta}}_{i_t,t} - \widehat{\boldsymbol{\theta}}_{\ell,t} \right\| > f(T_{i_t,t}) + f(T_{\ell,t})$$

     and obtain an updated graph $G_t = ([u], E_t)$

---

where the first term is the estimated reward of arm $a$ at time $t$ and the second term is the confidence radius of arm $a$ at time $t$ with parameter $\beta = \sqrt{d \log(1 + \frac{TL^2}{d\lambda}) + 2 \log(\frac{1}{\delta})} + \sqrt{\lambda}$.

After receiving the feedback $r_t$ from user $i_t$, the learner updates the statistics for user $i_t$ while keeping other users' statistics unchanged. Note that the estimated preference vector $\widehat{\boldsymbol{\theta}}_{i_t}$ is computed using historical information associated with user $i_t$ (Line 11). Finally, the algorithm updates the inferred clusters by deleting edges in graph $G_{t-1}$ if it determines that two users belong to different clusters. Specifically, for every user $\ell \in [u]$ that has an edge to user $i_t$, the algorithm checks if the distance between the estimated preference vectors of users $\ell$ and $i_t$ exceeds a specific threshold (Line 12). If so, the algorithm deletes the edge $i_t, \ell$ to split them apart and update the graph.

## 4   SMOOTHED ADVERSARIAL CONTEXT SETTING

Although `UniCLUB` offers significant theoretical improvements, it requires a change in the arm selection strategy during its execution, which might not be desirable in some circumstances. Additionally, the stochastic context setting necessitates an *i.i.d.* context generation process, which might be impractical in real-world applications. To overcome these limitations, based on the intuition in Section 3.2, we propose the smoothed adversarial context setting to eliminate the need for explicit pure exploration. This setting interpolates between the two extremes: the *i.i.d.* context generation in Gentile et al. (2014) and the adversarial context generation in Abbasi-Yadkori et al. (2011). The intrinsic diversity of contexts makes explicit exploration unnecessary, thereby ensuring a well-conditioned design matrix (Lemma 11). This approach allows existing algorithms in previous studies, such as CLUB (Gentile et al., 2014) and SCLUB (Li et al., 2019), which consistently employ the UCB strategy, to perform more effectively.

## 4.1 PROBLEM SETTING

In the smoothed adversarial setting, we retain Assumption 1, while replacing Assumptions 2 and 3 with Assumptions 4 and 5, respectively. As detailed below, since there is no need to switch arm selection strategies during execution, Assumption 4 follows the original assumption in Gentile et al. (2014), which does not require knowledge of a lower bound for $\gamma$. Meanwhile, Assumption 5 allows feature vectors (i.e., contexts) to be arbitrarily chosen by an adversary, but with some random perturbation to ensure the resulting contexts remain sufficiently diverse. This approach maintains enough data diversity to support effective learning while avoiding the need for explicit pure exploration.

**Assumption 4** (Well-separatedness among clusters). All users in the same cluster $\mathcal{I}_j$ share the same preference vector $\boldsymbol{\theta}^j$. For users in different clusters, there is a fixed but *unknown* gap $\gamma$ between their preference vectors. Specifically, for any cluster indices $i \neq j$,

$$\left\| \boldsymbol{\theta}^i - \boldsymbol{\theta}^j \right\|_2 \geq \gamma > 0.$$

**Assumption 5** (Context diversity for adversarial contexts). At each time step $t$, the feature vector $\boldsymbol{x}_a \in \mathcal{D}_t$ for each arm $a \in \mathcal{A}_t$ is drawn by a "smoothed" adversary, meaning that the adversary first chooses an arbitrary vector $\boldsymbol{\mu}_a \in \mathbb{R}^d$ with $\|\boldsymbol{\mu}_a\| \leq 1$, then samples a noise vector $\boldsymbol{\varepsilon}_a \in \mathbb{R}^d$ from a truncated multivariate Gaussian distribution where each dimension is truncated within $[-R, R]$, i.e., $\boldsymbol{\varepsilon}_a \sim \mathcal{N}(0, \sigma^2 \boldsymbol{I})$ conditioned on $|(\boldsymbol{\varepsilon}_a)_j| \leq R, \forall j \in [d]$. And the feature vector $\boldsymbol{x}_a = \boldsymbol{\mu}_a + \boldsymbol{\varepsilon}_a$.

**Remark 4.** The truncation in Assumption 5 is used to guarantee that the length of each feature vector is bounded, which is a standard requirement in the linear bandits literature. In fact, if we are only concerned with high-probability regret, we can also use a Gaussian distribution (without truncation) and show that the length of each feature vector is bounded with high probability. Note that Assumption 5 is more similar to the original linear bandit setting (Abbasi-Yadkori et al., 2011), except that we require each arm to be perturbed by Gaussian noise. It remains an open problem whether a fully adversarial setting can be achieved.

## 4.2 ALGORITHMS FOR SMOOTHED ADVERSARIAL CONTEXT SETTING

Our proposed algorithms for the smoothed adversarial context setting, SACLUB and SASCLUB, are essentially CLUB (Gentile et al., 2014) and SCLUB (Li et al., 2019) with $\lambda_x$ replaced by $\widetilde{\lambda}_x$ and $L$ replaced by $1 + \sqrt{d}R$ in the edge deletion threshold. Due to space constraints, we omit the full details of SACLUB and SASCLUB here, but the complete proofs are provided in Appendix E.

## 5 THEORETICAL ANALYSIS

In this section, we present the theoretical results of our algorithms, with detailed proofs provided in Appendices B, C, D, and E. For clarity, we ignore the constants but they are fleshed out in the proofs. Note that $\lambda_x$ appears in the denominator of the regret expressions. This is due to the assumption of bounded contexts ($\|\boldsymbol{X}\|_2$ is bounded) in the stochastic context setting, resulting in $\lambda_x = O(1/d)$, and therefore it is important to track the dependency of our regret bounds on $\lambda_x$.

**Theorem 1** (Regret of UniCLUB). *Under the stochastic context setting (Assumptions 1, 2, 3), the expected regret of the* UniCLUB *(Algorithm 1) satisfies:*

$$\mathbb{E}[R(T)] = O\left( \frac{ud}{\widetilde{\gamma}^2 \lambda_x} \log(T) + d\sqrt{mT} \log(T) \right).$$

**Theorem 2** (Regret of UniSCLUB). *Under the stochastic context setting (Assumptions 1, 2, 3), the expected regret of the* UniSCLUB *(Algorithm 2) satisfies:*

$$\mathbb{E}[R(T)] = O\left( \frac{ud}{\widetilde{\gamma}^2 \lambda_x} \log(T) + d\sqrt{mT} \log(T) \right).$$

**Remark 5.** Our results comprise two components: the first term is associated with cluster identification, and the second term aligns with the minimax near-optimal regret of linear contextual bandits. Notably, thanks to the careful design of UniCLUB and UniSCLUB, our results enhance the first term by $\widetilde{O}(u/\lambda_x^2)$, offering a significant improvement over existing studies, as detailed in Table 1.

**Theorem 3** (Regret of `PhaseUniCLUB`). *Under the stochastic context setting (Assumptions 1, 2, 3), but $\widetilde{\gamma}$ is unknown, the expected regret of algorithm `PhaseUniCLUB` (Algorithm 5) satisfies:*

$$\mathbb{E}[R(T)] = O\left(\frac{ud}{\gamma^2 \lambda_x^2}\log(T) + \left(\frac{ud}{\lambda_x}\right)^{\frac{1}{3}} T^{\frac{2}{3}} \log^{\frac{1}{3}}(T) + d\sqrt{mT}\log(T)\right).$$

**Remark 6.** Without knowledge of $\widetilde{\gamma}$ (i.e., a lower bound of $\gamma$), `PhaseUniCLUB` suffers a regret of order $\widetilde{O}(d\sqrt{mT} + d^{\frac{1}{3}}T^{\frac{2}{3}})$. As noted in Remark 2, this deteriorated result is expected since the algorithm has to contend with the issue of misclustering, similar to the challenges seen in misspecified linear bandits (Ghosh et al., 2017; Lattimore et al., 2020). Despite this challenge, we carefully design the phase lengths to exploit the property of clustering and achieve sublinear regret, which is a significant improvement over the linear regret typically expected in misspecified linear bandits. We emphasize that, as discussed in Remark 1, $\gamma$ not only has an intrinsic lower bound but also is typically predefined in real-world applications. Therefore, our study of the scenario where $\widetilde{\gamma}$ is unknown is primarily of theoretical interest. Further details about the algorithm can be found in Appendix D.

**Theorem 4** (Regret of `SACLUB` and `SASCLUB`). *Under the smoothed adversarial context setting (Assumptions 1, 4, 5), the expected regrets of `SACLUB` and `SASCLUB` both satisfy:*

$$\mathbb{E}[R(T)] = O\left(\frac{ud}{\gamma^2 \widetilde{\lambda}_x}\log(T) + d\sqrt{mT}\log(T)\right),$$

*where $\widetilde{\lambda}_x = c_1 \frac{\sigma^2}{\log K}$ for some constant $c_1$.*

**Remark 7.** For the smoothed adversarial context setting, we prove that the intrinsic diversity of contexts guarantees a lower bound on the minimum eigenvalue of $\mathbb{E}\left[\boldsymbol{x}_{a_t}\boldsymbol{x}_{a_t}^\mathsf{T}\right]$ (Lemma 11). This allows us to apply techniques similar to those used in the stochastic context setting to get this result.

## 6 PERFORMANCE EVALUATION

In this section, we present the evaluation results of our algorithms. We focus on the stochastic context setting in the main paper since the smoothed adversarial setting serves mainly for theoretical analysis, and algorithms `SACLUB`/`SASCLUB` are minor modifications of CLUB (Gentile et al., 2014)/SCLUB (Li et al., 2019). Nonetheless, we provide detailed evaluations of the smoothed adversarial setting and an ablation study on different arm set sizes and user numbers in Appendix G.

### 6.1 EXPERIMENT SETUP

We compare our algorithms against the following state-of-the-art algorithms for clustering of bandits: (1) LinUCB-One: LinUCB (Li et al., 2010) with a single preference vector shared across all users. (2) LinUCB-Ind: LinUCB (Li et al., 2010) with separate preference vectors estimated for each user. (3) CLUB (Gentile et al., 2014): A graph-based algorithm that consistently employs the UCB strategy. (4) SCLUB (Li et al., 2019): A set-based algorithm with improved practical performance. In addition to these clustering-based baselines, we also evaluate our approach against two graph-based algorithms from a related but distinct setting in Appendix G: GOB.Lin (Cesa-Bianchi et al., 2013), which incorporates user similarities using Laplacian regularization, and GraphUCB (Yang et al., 2020), which utilizes the random-walk Laplacian matrix to encode user relationships. Note that neither GOB.Lin nor GraphUCB explicitly performs clustering. All the experiments were conducted on a device equipped with a 3.60 GHz Intel Xeon W-2223 CPU and 32GB RAM. Each experiment was repeated over 5 random seeds, and the results are reported with confidence intervals calculated by dividing the standard deviation by the square root of the number of seeds.

### 6.2 DATASETS GENERATION AND PREPROCESSING

In our experiments, we employ one synthetic dataset and three real-world datasets, MovieLens-25M (Harper & Konstan, 2015), Last.fm (Cantador et al., 2011), and Yelp (Yelp, 2023). We generate the synthetic dataset and preprocess the real-world datasets following the same method in previous studies (Zhang et al., 2020; Li et al., 2024; 2025; Dai et al., 2024a;b).

To generate the synthetic dataset, we set the dimension $d = 50$, the number of users $u = 200$, and the total number of arms $|\mathcal{A}| = 5,000$. The feature vector $\boldsymbol{x}_a \in \mathbb{R}^d$ of each arm $a \in \mathcal{A}$ and the preference vector $\boldsymbol{\theta}_i \in \mathbb{R}^d$ for each user $i \in [u]$ are generated by independently sampling each dimension from a uniform distribution $\mathcal{U}(-1, 1)$ and then normalizing to unit length.

For the real-world datasets, MovieLens-25M, Last.fm, and Yelp, we regard movies/artists/businesses as arms. We extract a subset of $|\mathcal{A}| = 5,000$ arms with the most quantity of user-assigned ratings/tags, and a subset of $u = 200$ users who assign the most quantity of ratings/tags. Using the data extracted above, we create a *feedback matrix* $\boldsymbol{R}$ of size $u \times |\mathcal{A}|$, where each element $\boldsymbol{R}_{i,j}$ represents the user $i$'s feedback to arm $j$. We assume that the user's feedback is binary. For the MovieLens and Yelp datasets, a user's feedback for a movie/business is 1 if the user's rating is higher than 3. For the Last.fm dataset, a user's feedback for an artist is 1 if the user assigns a tag to the artist. We generate feature vectors and preference vectors by decomposing $\boldsymbol{R}$ using Singular Value Decomposition (SVD) as $\boldsymbol{R} = \boldsymbol{\Theta} \boldsymbol{S} \boldsymbol{A}^\mathsf{T}$, where $\boldsymbol{\Theta} = \{\boldsymbol{\theta}_i\}_{i \in [u]}$, and $\boldsymbol{A} = \{\boldsymbol{x}_a\}_{a \in \mathcal{A}}$. Then the top $d = 50$ dimensions of these vectors associated with the highest singular values in $\boldsymbol{S}$ are extracted.

### 6.3 EXPERIMENT RESULTS

In the experiment, to incorporate user clustering, we randomly select 50 users and partition them into 10 clusters. For each cluster $j$, we calculate the mean preference vector across all users within that cluster to serve as $\boldsymbol{\theta}^j$. Note that the number of clusters is unknown to the algorithms. At each round $t$, we uniformly draw a user $i_t$ from the 50 users, and randomly select 100 arms from $\mathcal{A}$ to form the arm set $\mathcal{A}_t$. The results are presented in Figure 1.

As shown in Figure 1, the algorithms that employ user clustering significantly outperform LinUCB-Ind and LinUCB-One, which do not consider the similarity among users or cluster users. More importantly, our graph-based algorithm `UniCLUB` consistently outperforms the graph-based baseline CLUB, and our set-based algorithm `UniSCLUB` is better than the set-based baseline SCLUB across all four datasets. It is important to note that the modest advantage of `UniCLUB`/`UniSCLUB` over CLUB/SCLUB is both expected and reasonable, given the logarithmic improvement in the regret upper bound. The results demonstrate the effect of uniform exploration and the robustness of our proposed algorithms across various datasets. More evaluation under the smoothed adversarial context setting and ablation study can be found in Appendix G.

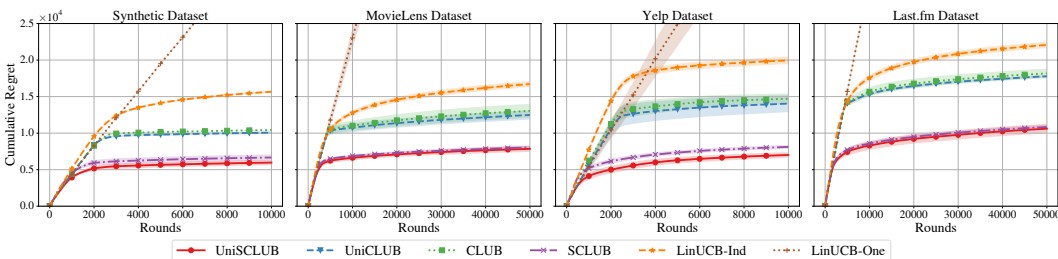

Figure 1: Comparison of cumulative regrets in the stochastic context setting.

## 7 CONCLUSION

In this paper, we proposed two new algorithms for online clustering of bandits, `UniCLUB` and `UniSCLUB`, which incorporate an additional pure exploration phase to enhance the identification of user clusters. Notably, our algorithms require significantly weaker assumptions while achieving superior cumulative regret compared to previous studies. Furthermore, we introduced the smoothed adversarial context setting, which interpolates between *i.i.d.* and fully adversarial context generation. We showed that with minor modifications, existing algorithms perform better in this new setting. Finally, we conducted extensive evaluations to validate the effectiveness of our methods.

ACKNOWLEDGMENTS

The work of John C.S. Lui was supported in part by the RGC's SRFS2122-4S02.

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

## A    MORE DISCUSSIONS ON RELATED WORK

Leveraging data diversity (i.e., conditions refer to the minimum eigenvalue of a design matrix) in stochastic linear contextual bandits has two lines of research. The first line involves using additional "diversity conditions" to improve cumulative regrets. For example, Bastani et al. (2021) introduce a condition for the disjoint-parameter case, and prove that a non-explorative greedy algorithm achieves $O(\log T)$ problem-dependent regret on a 2-arm bandit instance. Hao et al. (2020) give a condition and prove a constant problem-dependent regret for LinUCB in the shared-parameter case. Wu et al. (2020) show that under some diversity conditions, LinUCB achieves constant expected regret in the disjoint-parameter case. Ghosh & Sankararaman (2022) use a condition similar to Gentile et al. (2014) and achieve a problem-independent logarithmic regret for linear contextual bandits. The second line of research focuses on achieving concurrent statistical inference and regret minimization (i.e., multi-objective MAB). This involves performing additional tasks on top of regret minimization, such as clustering (Gentile et al., 2014), model selection (Chatterji et al., 2020; Ghosh et al., 2021a), personalization (Ghosh et al., 2021b), and exploration under safety constraints (Amani et al., 2019).

## B    THEORETICAL ANALYSIS OF UNICLUB

**Lemma 1.** *With probability at least $1 - \delta$ for any $\delta \in (0, 1)$, $\forall t \in [T]$ and $\forall i \in [u]$,*

$$\left\|\widehat{\boldsymbol{\theta}}_{i,t} - \boldsymbol{\theta}^{j(i)}\right\|_2 \leq \frac{\sqrt{2 \log\left(\frac{u}{\delta}\right) + d \log\left(1 + \frac{T_{i,t}L^2}{\lambda d}\right)} + \sqrt{\lambda}}{\sqrt{\lambda + \lambda_{\min}(\boldsymbol{S}_{i,t})}}.$$

*Proof.* Fix a user $i \in [u]$, for all $t \in [T]$, we have

$$\widehat{\boldsymbol{\theta}}_{i,t} - \boldsymbol{\theta}^{j(i)} = \left(\lambda \boldsymbol{I} + \sum_{\tau \in [t]: i_\tau = i} \boldsymbol{x}_{a_\tau} \boldsymbol{x}_{a_\tau}^\mathsf{T}\right)^{-1} \left(\sum_{\tau \in [t]: i_\tau = i} \boldsymbol{x}_{a_\tau} (\boldsymbol{x}_{a_\tau}^\mathsf{T} \boldsymbol{\theta}^{j(i)} + \eta_\tau)\right) - \boldsymbol{\theta}^{j(i)}$$

$$= \left(\lambda \boldsymbol{I} + \sum_{\tau \in [t]: i_\tau = i} \boldsymbol{x}_{a_\tau} \boldsymbol{x}_{a_\tau}^\mathsf{T}\right)^{-1} \left[\left(\lambda \boldsymbol{I} + \sum_{\tau \in [t]: i_\tau = i} \boldsymbol{x}_{a_\tau} \boldsymbol{x}_{a_\tau}^\mathsf{T}\right) \boldsymbol{\theta}^{j(i)} + \sum_{\tau \in [t]: i_\tau = i} \boldsymbol{x}_{a_\tau} \eta_\tau - \lambda \boldsymbol{\theta}^{j(i)}\right] - \boldsymbol{\theta}^{j(i)}$$

$$= \boldsymbol{\theta}^{j(i)} + (\lambda \boldsymbol{I} + \boldsymbol{S}_{i,t})^{-1} \sum_{\tau \in [t]: i_\tau = i} \boldsymbol{x}_{a_\tau} \eta_\tau - \lambda (\lambda \boldsymbol{I} + \boldsymbol{S}_{i,t})^{-1} \boldsymbol{\theta}^{j(i)} - \boldsymbol{\theta}^{j(i)}$$

$$= \overline{\boldsymbol{S}}_{i,t}^{-1} \sum_{\tau \in [t]: i_\tau = i} \boldsymbol{x}_{a_\tau} \eta_\tau - \lambda \overline{\boldsymbol{S}}_{i,t}^{-1} \boldsymbol{\theta}^{j(i)},$$

where we denote $\overline{\boldsymbol{S}}_{i,t} \triangleq \lambda \boldsymbol{I} + \boldsymbol{S}_{i,t} = \lambda \boldsymbol{I} + \sum_{\tau \in [t]: i_\tau = i} \boldsymbol{x}_{a_\tau} \boldsymbol{x}_{a_\tau}^\mathsf{T}$.

For any vector $\boldsymbol{x} \in \mathbb{R}^d$,

$$\boldsymbol{x}^\mathsf{T}\left(\widehat{\boldsymbol{\theta}}_{i,t} - \boldsymbol{\theta}^{j(i)}\right) = \boldsymbol{x}^\mathsf{T} \overline{\boldsymbol{S}}_{i,t}^{-1} \sum_{\tau \in [t]: i_\tau = i} \boldsymbol{x}_{a_\tau} \eta_\tau - \lambda \boldsymbol{x}^\mathsf{T} \overline{\boldsymbol{S}}_{i,t}^{-1} \boldsymbol{\theta}^{j(i)}$$

$$= \left\langle \boldsymbol{x}, \sum_{\tau \in [t]: i_\tau = i} \boldsymbol{x}_{a_\tau} \eta_\tau \right\rangle_{\overline{\boldsymbol{S}}_{i,t}^{-1}} - \lambda \left\langle \boldsymbol{x}, \boldsymbol{\theta}^{j(i)} \right\rangle_{\overline{\boldsymbol{S}}_{i,t}^{-1}}.$$

Therefore, by the Cauchy–Schwarz inequality,

$$\left|\boldsymbol{x}^\mathsf{T}\left(\widehat{\boldsymbol{\theta}}_{i,t} - \boldsymbol{\theta}^{j(i)}\right)\right| \leq \|\boldsymbol{x}\|_{\overline{\boldsymbol{S}}_{i,t}^{-1}} \left(\left\|\sum_{\tau \in [t]: i_\tau = i} \boldsymbol{x}_{a_\tau} \eta_\tau\right\|_{\overline{\boldsymbol{S}}_{i,t}^{-1}} + \lambda \|\boldsymbol{\theta}^{j(i)}\|_{\overline{\boldsymbol{S}}_{i,t}^{-1}}\right) \qquad (1)$$

Next, we bound the two terms in the parenthesis. For the first term, consider the $\sigma$-algebra $\mathcal{F}_t = \sigma(i_1, \boldsymbol{x}_{a_1}, \eta_1, \ldots, i_t, \boldsymbol{x}_{a_t}, \eta_t, i_{t+1}, \boldsymbol{x}_{a_{t+1}})$, where $i_t$ and $\boldsymbol{x}_{a_t}$ are $\mathcal{F}_{t-1}$-measurable, and $\eta_t$ is $\mathcal{F}_t$-measurable. Let $\{\mathcal{F}_t\}_{t=1}^\infty$ be a filtration. By applying Theorem 1 of Abbasi-Yadkori et al. (2011)

and using the union bound over all users $i \in [u]$, we have that for all $i \in [u]$ and for all $t \in [T]$, with probability $\geq 1 - \delta$,

$$\left\| \sum_{\tau \in [t]:i_\tau = i} \boldsymbol{x}_{a_\tau} \eta_\tau \right\|_{\overline{\boldsymbol{S}}_{i,t}^{-1}} \leq \sqrt{2 \log \left( \frac{u \det(\overline{\boldsymbol{S}}_{i,t})^{\frac{1}{2}} \det(\lambda \boldsymbol{I})^{-\frac{1}{2}}}{\delta} \right)}$$

$$\leq \sqrt{2 \log \left( \frac{u}{\delta} \right) + d \log \left( 1 + \frac{T_{i,t} L^2}{\lambda d} \right)},$$

where we use the standard derivation due to Lemma 16: $\det(\overline{\boldsymbol{S}}_{i,t}) \leq \left( \frac{\mathrm{tr}(\overline{\boldsymbol{S}}_{i,t})}{d} \right)^d \leq \left( \lambda + \frac{T_{i,t} L^2}{d} \right)^d$ and the fact that $\det(\lambda \boldsymbol{I}) = \lambda^d$.

For the second term, by the property of the Rayleigh quotient, for any invertible PSD matrix $\boldsymbol{V}$ and non-zero vector $\boldsymbol{w}$, we have

$$\frac{\|\boldsymbol{w}\|_{\boldsymbol{V}^{-1}}^2}{\|\boldsymbol{w}\|_2^2} = \frac{\boldsymbol{w}^\mathsf{T} \boldsymbol{V}^{-1} \boldsymbol{w}}{\boldsymbol{w}^\mathsf{T} \boldsymbol{w}} \leq \lambda_{\max}(\boldsymbol{V}^{-1}) = \frac{1}{\lambda_{\min}(\boldsymbol{V})}.$$

Therefore, by the fact that $\lambda_{\min}(\overline{\boldsymbol{S}}_{i,t}) \geq \lambda$, and the assumption that $\|\boldsymbol{\theta}^{j(i)}\|_2 \leq 1$, we have

$$\lambda \|\boldsymbol{\theta}^{j(i)}\|_{\overline{\boldsymbol{S}}_{i,t}^{-1}} \leq \sqrt{\lambda} \|\boldsymbol{\theta}^{j(i)}\|_2 \leq \sqrt{\lambda}. \tag{2}$$

Plugging in $\boldsymbol{x} = \overline{\boldsymbol{S}}_{i,t} \left( \widehat{\boldsymbol{\theta}}_{i,t} - \boldsymbol{\theta}^{j(i)} \right)$ to Equation (1), we have

$$\left| \boldsymbol{x}^\mathsf{T} \left( \widehat{\boldsymbol{\theta}}_{i,t} - \boldsymbol{\theta}^{j(i)} \right) \right| = \left\| \widehat{\boldsymbol{\theta}}_{i,t} - \boldsymbol{\theta}^{j(i)} \right\|_{\overline{\boldsymbol{S}}_{i,t}}^2$$

$$\leq \left\| \overline{\boldsymbol{S}}_{i,t} \left( \widehat{\boldsymbol{\theta}}_{i,t} - \boldsymbol{\theta}^{j(i)} \right) \right\|_{\overline{\boldsymbol{S}}_{i,t}^{-1}} \left( \sqrt{2 \log \left( \frac{u}{\delta} \right) + d \log \left( 1 + \frac{T_{i,t} L^2}{\lambda d} \right)} + \sqrt{\lambda} \right)$$

$$= \left\| \widehat{\boldsymbol{\theta}}_{i,t} - \boldsymbol{\theta}^{j(i)} \right\|_{\overline{\boldsymbol{S}}_{i,t}} \left( \sqrt{2 \log \left( \frac{u}{\delta} \right) + d \log \left( 1 + \frac{T_{i,t} L^2}{\lambda d} \right)} + \sqrt{\lambda} \right).$$

Again, by the property of the Rayleigh quotient, for any PSD matrix $\boldsymbol{V}$ and non-zero vector $\boldsymbol{w}$,

$$\lambda_{\min}(\boldsymbol{V}) \leq \frac{\boldsymbol{w}^\mathsf{T} \boldsymbol{V} \boldsymbol{w}}{\boldsymbol{w}^\mathsf{T} \boldsymbol{w}} = \frac{\|\boldsymbol{w}\|_{\boldsymbol{V}}^2}{\|\boldsymbol{w}\|_2^2} \implies \|\boldsymbol{w}\|_2 \leq \frac{\|\boldsymbol{w}\|_{\boldsymbol{V}}}{\sqrt{\lambda_{\min}(\boldsymbol{V})}}.$$

Therefore, dividing $\left\| \widehat{\boldsymbol{\theta}}_{i,t} - \boldsymbol{\theta}^{j(i)} \right\|_{\overline{\boldsymbol{S}}_{i,t}}$ on both sides and applying the above inequality, we get

$$\left\| \widehat{\boldsymbol{\theta}}_{i,t} - \boldsymbol{\theta}^{j(i)} \right\|_2 \leq \frac{\sqrt{2 \log \left( \frac{u}{\delta} \right) + d \log \left( 1 + \frac{T_{i,t} L^2}{\lambda d} \right)} + \sqrt{\lambda}}{\sqrt{\lambda_{\min}(\overline{\boldsymbol{S}}_{i,t})}}$$

$$\leq \frac{\sqrt{2 \log \left( \frac{u}{\delta} \right) + d \log \left( 1 + \frac{T_{i,t} L^2}{\lambda d} \right)} + \sqrt{\lambda}}{\sqrt{\lambda + \lambda_{\min}(\boldsymbol{S}_{i,t})}},$$

where in the last inequality we use $\lambda_{\min}(\lambda \boldsymbol{I} + \boldsymbol{S}_{i,t}) \geq \lambda_{\min}(\lambda \boldsymbol{I}) + \lambda_{\min}(\boldsymbol{S}_{i,t})$, due to Weyl's inequality. $\qquad \square$

**Lemma 2.** *In the uniform exploration phase of Algorithm 1, with probability at least $1 - \delta$ for any $\delta \in (0, 1)$, if $T_{i,t} \geq \frac{8L^2}{\lambda_x} \log \left( \frac{ud}{\delta} \right)$ for all users $i \in [u]$, we have*

$$\lambda_{\min}(\boldsymbol{S}_{i,t}) \geq \frac{\lambda_x T_{i,t}}{2}, \forall i \in [u].$$

*Proof.* To apply the matrix Chernoff bound (Lemma 12), we first verify the required two conditions for the self-adjoint matrices $\boldsymbol{x}_{a_\tau}\boldsymbol{x}_{a_\tau}^\mathsf{T}$ for any $\tau \in [t]$. First, due to the generation process of feature vectors, $\boldsymbol{x}_{a_\tau}\boldsymbol{x}_{a_\tau}^\mathsf{T}$ is independent and obviously positive semi-definite. Second, by the Courant-Fischer theorem, we have

$$\lambda_{\max}(\boldsymbol{x}_{a_\tau}\boldsymbol{x}_{a_\tau}^\mathsf{T}) = \max_{\boldsymbol{w}:\|\boldsymbol{w}\|=1} \boldsymbol{w}^\mathsf{T}\boldsymbol{x}_{a_\tau}\boldsymbol{x}_{a_\tau}^\mathsf{T}\boldsymbol{w} = \max_{\boldsymbol{w}:\|\boldsymbol{w}\|=1} (\boldsymbol{w}^\mathsf{T}\boldsymbol{x}_{a_\tau})^2 \leq \max_{\boldsymbol{w}:\|\boldsymbol{w}\|=1} \|\boldsymbol{w}\|^2\|\boldsymbol{x}_{a_\tau}\|^2 \leq L^2.$$

According to Algorithm 1, in the uniform exploration phase, for all $\tau \in [t]$, $a_\tau$ is uniformly selected in $\mathcal{A}_\tau$. Also, by Assumption 3, all the feature vectors in $\mathcal{A}_\tau$ are independently sampled from $\boldsymbol{X}$, therefore by Lemma 15, $\boldsymbol{x}_{a_\tau}$ follows the same distribution as $\boldsymbol{X}$.

So for a fixed user $i \in [u]$, we can compute

$$\mu_{\min} = \lambda_{\min}\left(\sum_{\tau \in [t]: i_\tau = i} \mathbb{E}[\boldsymbol{x}_{a_\tau}\boldsymbol{x}_{a_\tau}^\mathsf{T}]\right) = \lambda_{\min}\big(T_{i,t}\,\mathbb{E}[\boldsymbol{X}\boldsymbol{X}^\mathsf{T}]\big) = T_{i,t}\lambda_x,$$

where the last equality is due to the minimum eigenvalue in Assumption 3. Now applying Lemma 12, we get for any $\varepsilon \in (0,1)$,

$$\Pr\left[\lambda_{\min}(\boldsymbol{S}_{i,t}) \leq (1-\varepsilon)T_{i,t}\lambda_x\right] \leq d\left[\frac{e^{-\varepsilon}}{(1-\varepsilon)^{1-\varepsilon}}\right]^{T_{i,t}\lambda_x/L^2}.$$

Choosing $\varepsilon = \frac{1}{2}$, we get

$$\Pr\left[\lambda_{\min}(\boldsymbol{S}_{i,t}) \leq \frac{T_{i,t}\lambda_x}{2}\right] \leq d\left(\sqrt{2}e^{-\frac{1}{2}}\right)^{T_{i,t}\lambda_x/L^2}.$$

Letting the RHS be $\frac{\delta}{u}$, we get $T_{i,t} = \frac{L^2 \log(\frac{ud}{\delta})}{\lambda_x(\frac{1}{2} - \log(2))}$. Therefore, for any fixed user $i \in [u]$, $\lambda_{\min}(\boldsymbol{S}_{i,t}) \geq \frac{T_{i,t}\lambda_x}{2}$ holds with probability at least $1 - \frac{\delta}{u}$ when $T_{i,t} \geq \frac{8L^2}{\lambda_x}\log\big(\frac{ud}{\delta}\big)$. The proof follows by a union bound over all users $i \in [u]$. $\qquad\square$

**Lemma 3.** *With probability at least $1 - 3\delta$, Algorithm 1 can cluster all the users correctly after*

$$T_0 \triangleq 16u\log\Big(\frac{u}{\delta}\Big) + 4u\max\left\{\frac{8L^2}{\lambda_x}\log\Big(\frac{ud}{\delta}\Big), \frac{512d}{\widetilde{\gamma}^2\lambda_x}\log\Big(\frac{u}{\delta}\Big)\right\}. \tag{3}$$

*Proof.* Combining Lemma 1 and Lemma 2, we have with probability at least $1 - 2\delta$, when $T_{i,t} \geq \frac{8L^2}{\lambda_x}\log\big(\frac{ud}{\delta}\big)$ for all users $i \in [u]$,

$$\begin{aligned}
\left\|\widehat{\boldsymbol{\theta}}_{i,t} - \boldsymbol{\theta}^{j(i)}\right\|_2 &\leq \frac{\sqrt{2\log\big(\frac{u}{\delta}\big) + d\log\Big(1 + \frac{T_{i,t}L^2}{\lambda d}\Big)} + \sqrt{\lambda}}{\sqrt{\lambda + \lambda_{\min}(\boldsymbol{S}_{i,t})}} \\
&\leq \frac{\sqrt{2\log\big(\frac{u}{\delta}\big) + d\log\Big(1 + \frac{T_{i,t}L^2}{\lambda d}\Big)} + \sqrt{\lambda}}{\sqrt{\lambda + T_{i,t}\lambda_x/2}} \triangleq f(T_{i,t}), \forall i \in [u].
\end{aligned}$$

Next, we find a sufficient time step $T_{i,t}$ such that the following holds:

$$f(T_{i,t}) \triangleq \frac{\sqrt{2\log\big(\frac{u}{\delta}\big) + d\log\Big(1 + \frac{T_{i,t}L^2}{\lambda d}\Big)} + \sqrt{\lambda}}{\sqrt{\lambda + T_{i,t}\lambda_x/2}} \leq \frac{\widetilde{\gamma}}{4}. \tag{4}$$

We assume $\lambda \leq 2\log\big(\frac{u}{\delta}\big) + d\log\Big(1 + \frac{T_{i,t}L^2}{\lambda d}\Big)$, which typically holds. Then a sufficient condition for Equation (4) is

$$\frac{2\log\big(\frac{u}{\delta}\big) + d\log\Big(1 + \frac{T_{i,t}L^2}{\lambda d}\Big)}{\lambda + T_{i,t}\lambda_x/2} \leq \frac{\widetilde{\gamma}^2}{64}.$$

To make the above equation to hold, it suffices to let

$$\frac{2\log\left(\frac{u}{\delta}\right)}{T_{i,t}\lambda_x/2} \leq \frac{\widetilde{\gamma}^2}{128} \quad \text{and} \quad \frac{d\log\left(1 + \frac{T_{i,t}L^2}{\lambda d}\right)}{T_{i,t}\lambda_x/2} \leq \frac{\widetilde{\gamma}^2}{128}.$$

The first inequality holds when $T_{i,t} \geq \frac{512\log\left(\frac{u}{\delta}\right)}{\widetilde{\gamma}^2\lambda_x}$. For the second inequality, by some basic arithmetic (Lemma 14), a sufficient condition is $T_{i,t} \geq \frac{512d}{\widetilde{\gamma}^2\lambda_x}\log\left(\frac{256L^2}{\widetilde{\gamma}^2\lambda\lambda_x}\right)$. By choosing $\delta$ such that $\frac{u}{\delta} \geq \frac{256L^2}{\widetilde{\gamma}^2\lambda\lambda_x}$, we get a sufficient condition for Equation (4): $T_{i,t} \geq \frac{512d}{\widetilde{\gamma}^2\lambda_x}\log\left(\frac{u}{\delta}\right)$.

In summary, in the uniform exploration phase of Algorithm 1, with probability at least $1 - 2\delta$ for some $\delta > 0$, when

$$T_{i,t} \geq \max\left\{\frac{8L^2}{\lambda_x}\log\left(\frac{ud}{\delta}\right), \frac{512d}{\widetilde{\gamma}^2\lambda_x}\log\left(\frac{u}{\delta}\right)\right\}, \forall i \in [u] \tag{5}$$

we have $\left\|\widehat{\boldsymbol{\theta}}_{i,t} - \boldsymbol{\theta}^{j(i)}\right\|_2 \leq f(T_{i,t}) \leq \frac{\widetilde{\gamma}}{4}$ for all users $i \in [u]$.

Because of Assumption 1, users arrive uniformly, so by Lemma 13 and a union bound over all users $i \in [u]$, Equation (5) holds for all $i \in [u]$ with probability at least $1 - \delta$ when

$$t \geq T_0 \triangleq 16u\log\left(\frac{u}{\delta}\right) + 4u\max\left\{\frac{8L^2}{\lambda_x}\log\left(\frac{ud}{\delta}\right), \frac{512d}{\widetilde{\gamma}^2\lambda_x}\log\left(\frac{u}{\delta}\right)\right\}.$$

Therefore, with probability $1 - 3\delta$, we have $\left\|\widehat{\boldsymbol{\theta}}_{i,t} - \boldsymbol{\theta}^{j(i)}\right\|_2 \leq f(T_{i,t}) \leq \frac{\widetilde{\gamma}}{4}, \forall i \in [u]$ when $t \geq T_0$.

Next, we show that under this condition, the algorithm will cluster all the users correctly. To guarantee this, we need to verify two aspects: (1) if users $k, \ell$ are in the same cluster, then the algorithm will not delete edge $(k, \ell)$; (2) if users $k, \ell$ are not in the same cluster, then the algorithm will delete edge $(k, \ell)$. We show the contrapositive of (1): if edge $(k, \ell)$ is deleted, then users $k, \ell$ are not in the same cluster. Due to the triangle inequality and the deletion rule in Algorithm 1, we have

$$\left\|\boldsymbol{\theta}^{j(k)} - \boldsymbol{\theta}^{j(\ell)}\right\|_2 \geq \left\|\widehat{\boldsymbol{\theta}}_{k,t} - \widehat{\boldsymbol{\theta}}_{\ell,t}\right\|_2 - \left\|\widehat{\boldsymbol{\theta}}_{k,t} - \boldsymbol{\theta}^{j(k)}\right\|_2 - \left\|\widehat{\boldsymbol{\theta}}_{\ell,t} - \boldsymbol{\theta}^{j(\ell)}\right\|_2$$
$$\geq \left\|\widehat{\boldsymbol{\theta}}_{k,t} - \widehat{\boldsymbol{\theta}}_{\ell,t}\right\|_2 - f(T_{k,t}) - f(T_{\ell,t}) > 0.$$

So by Assumption 2, $\left\|\boldsymbol{\theta}^{j(k)} - \boldsymbol{\theta}^{j(\ell)}\right\|_2 > 0$ implies that users $k, \ell$ are not in the same cluster. For (2), we show that if $\left\|\boldsymbol{\theta}^{j(k)} - \boldsymbol{\theta}^{j(\ell)}\right\|_2 \geq \widetilde{\gamma}$, the algorithm will delete edge $(k, \ell)$. By the triangle inequality,

$$\left\|\widehat{\boldsymbol{\theta}}_{k,t} - \widehat{\boldsymbol{\theta}}_{\ell,t}\right\|_2 \geq \left\|\boldsymbol{\theta}^{j(k)} - \boldsymbol{\theta}^{j(\ell)}\right\|_2 - \left\|\widehat{\boldsymbol{\theta}}_{k,t} - \boldsymbol{\theta}^{j(k)}\right\|_2 - \left\|\widehat{\boldsymbol{\theta}}_{\ell,t} - \boldsymbol{\theta}^{j(\ell)}\right\|_2$$
$$\geq \widetilde{\gamma} - \frac{\widetilde{\gamma}}{4} - \frac{\widetilde{\gamma}}{4} = \frac{\widetilde{\gamma}}{2} \geq f(T_{k,t}) + f(T_{\ell,t}),$$

which triggers Algorithm 1 to delete edge $(k, \ell)$. $\qquad \square$

**Lemma 4.** *With probability at least $1 - 4\delta$, for all $t > T_0$, we have*

$$\left|\boldsymbol{x}_a^\top\left(\widehat{\boldsymbol{\theta}}_{V_t,t-1} - \boldsymbol{\theta}^{j(i_t)}\right)\right| \leq \beta\|\boldsymbol{x}_a\|_{\overline{\boldsymbol{M}}_{V_t,t-1}^{-1}} \triangleq C_{a,t},$$

*where $\beta = \sqrt{d\log(1 + \frac{TL^2}{d\lambda}) + 2\log(\frac{1}{\delta})} + \sqrt{\lambda}$.*

*Proof.* Assume that after $T_0$, the underlying clusters are identified correctly, meaning that $V_t$ is the true cluster that contains user $i_t$, i.e., $V_t = \mathcal{I}_{j(i_t)}$ then we have

$$\widehat{\boldsymbol{\theta}}_{V_t,t-1} - \boldsymbol{\theta}^{j(i_t)} = \left(\lambda\boldsymbol{I} + \sum_{\tau\in[t-1]:i_\tau\in V_t}\boldsymbol{x}_{a_\tau}\boldsymbol{x}_{a_\tau}^\top\right)^{-1}\left(\sum_{\tau\in[t-1]:i_\tau\in V_t}r_\tau\boldsymbol{x}_{a_\tau}\right) - \boldsymbol{\theta}^{j(i_t)}$$

$$= \left( \lambda \boldsymbol{I} + \sum_{\tau \in [t-1]: i_\tau \in V_t} \boldsymbol{x}_{a_\tau} \boldsymbol{x}_{a_\tau}^\mathsf{T} \right)^{-1} \left( \sum_{\tau \in [t-1]: i_\tau \in V_t} \boldsymbol{x}_{a_\tau} \left( \boldsymbol{x}_{a_\tau}^\mathsf{T} \boldsymbol{\theta}_{i_t} + \eta_\tau \right) \right) - \boldsymbol{\theta}^{j(i_t)} \tag{6}$$

$$= \overline{\boldsymbol{M}}_{V_t,t-1}^{-1} \left[ \left( \lambda \boldsymbol{I} + \sum_{\tau \in [t-1]: i_\tau \in V_t} \boldsymbol{x}_{a_\tau} \boldsymbol{x}_{a_\tau}^\mathsf{T} \right) \boldsymbol{\theta}_{i_t} - \lambda \boldsymbol{\theta}_{i_t} + \sum_{\tau \in [t-1]: i_\tau \in V_t} \boldsymbol{x}_{a_\tau} \eta_\tau \right] - \boldsymbol{\theta}^{j(i_t)}$$

$$= -\lambda \overline{\boldsymbol{M}}_{V_t,t-1}^{-1} \boldsymbol{\theta}_{i_t} + \overline{\boldsymbol{M}}_{V_t,t-1}^{-1} \sum_{\tau \in [t-1]: i_\tau \in V_t} \boldsymbol{x}_{a_\tau} \eta_\tau,$$

where we denote $\overline{\boldsymbol{M}}_{V_t,t-1} = \lambda \boldsymbol{I} + \boldsymbol{M}_{V_t,t-1}$. Equation (6) is because $V_t$ is the true cluster that contains $i_t$, thus by Assumption 2, $\boldsymbol{\theta}_{i_\tau} = \boldsymbol{\theta}_{i_t}, \forall i_\tau \in V_t$. Therefore, we have

$$\left| \boldsymbol{x}_a^\mathsf{T} \left( \widehat{\boldsymbol{\theta}}_{V_t,t-1} - \boldsymbol{\theta}^{j(i_t)} \right) \right| \leq \lambda \left| \boldsymbol{x}_a^\mathsf{T} \overline{\boldsymbol{M}}_{V_t,t-1}^{-1} \boldsymbol{\theta}^{j(i_t)} \right| + \left| \boldsymbol{x}_a^\mathsf{T} \overline{\boldsymbol{M}}_{V_t,t-1}^{-1} \sum_{\tau \in [t-1]: i_\tau \in V_t} \boldsymbol{x}_{a_\tau} \eta_\tau \right|$$

$$= \lambda \left| \left\langle \boldsymbol{x}_a, \boldsymbol{\theta}^{j(i_t)} \right\rangle_{\overline{\boldsymbol{M}}_{V_t,t-1}^{-1}} \right| + \left| \left\langle \boldsymbol{x}_a, \sum_{\tau \in [t-1]: i_\tau \in V_t} \boldsymbol{x}_{a_\tau} \eta_\tau \right\rangle_{\overline{\boldsymbol{M}}_{V_t,t-1}^{-1}} \right|$$

$$\leq \lambda \| \boldsymbol{x}_a \|_{\overline{\boldsymbol{M}}_{V_t,t-1}^{-1}} \left\| \boldsymbol{\theta}^{j(i_t)} \right\|_{\overline{\boldsymbol{M}}_{V_t,t-1}^{-1}} + \| \boldsymbol{x}_a \|_{\overline{\boldsymbol{M}}_{V_t,t-1}^{-1}} \left\| \sum_{\tau \in [t-1]: i_\tau \in V_t} \boldsymbol{x}_{a_\tau} \eta_\tau \right\|_{\overline{\boldsymbol{M}}_{V_t,t-1}^{-1}} \tag{7}$$

$$\leq \| \boldsymbol{x}_a \|_{\overline{\boldsymbol{M}}_{V_t,t-1}^{-1}} \left( \sqrt{\lambda} + \left\| \sum_{\tau \in [t-1]: i_\tau \in V_t} \boldsymbol{x}_{a_\tau} \eta_\tau \right\|_{\overline{\boldsymbol{M}}_{V_t,t-1}^{-1}} \right), \tag{8}$$

where Equation (7) is by the Cauchy–Schwarz inequality and Equation (8) is derived by Equation (2). Following Theorem 1 in Abbasi-Yadkori et al. (2011), with probability at least $1 - \delta$, for a fixed user $i \in [u]$, we have

$$\left\| \sum_{\tau \in [t-1]: i_\tau \in V_t} \boldsymbol{x}_{a_\tau} \eta_\tau \right\|_{\overline{\boldsymbol{M}}_{V_t,t-1}^{-1}} \leq \sqrt{2 \log \left( \frac{\det(\overline{\boldsymbol{M}}_{V_t,t-1})^{\frac{1}{2}} \det(\lambda \boldsymbol{I})^{-\frac{1}{2}}}{\delta} \right)}$$

$$= \sqrt{2 \log \left( \frac{1}{\delta} \right) + \log \left( \frac{\det(\overline{\boldsymbol{M}}_{V_t,t-1})}{\det(\lambda \boldsymbol{I})} \right)}$$

$$\leq \sqrt{2 \log \left( \frac{1}{\delta} \right) + d \log \left( 1 + \frac{TL^2}{\lambda d} \right)}, \tag{9}$$

where in Equation (9), we use $\det(\lambda \boldsymbol{I}) = \lambda^d$ and by Lemma 16, we get

$$\det(\overline{\boldsymbol{M}}_{V_t,t-1}) \leq \left( \frac{\text{tr}(\overline{\boldsymbol{M}}_{V_t,t-1})}{d} \right)^d = \left( \frac{\text{tr}(\lambda \boldsymbol{I}) + \text{tr} \left( \sum_{\substack{\tau \in [t-1] \\ i_\tau \in V_t}} \boldsymbol{x}_{a_\tau} \boldsymbol{x}_{a_\tau}^\mathsf{T} \right)}{d} \right)^d \leq \left( \frac{\lambda d + TL^2}{d} \right)^d.$$

Since the clusters are correctly identified after $T_0$ with probability at least $1 - 3\delta$ (Lemma 3), and Equation (9) holds with probability at least $1 - \delta$. By plugging Equation (9) into Equation (8) and collecting high probability events, we conclude the proof. $\qquad \square$

**Lemma 5.** *Under the high probability event in Lemma 3, for any cluster $j \in [m]$, we have*

$$\sum_{\substack{t=T_0+1 \\ i_t \in \mathcal{I}_j}}^{T} \min \left\{ 1, \| \boldsymbol{x}_{a_t} \|_{\overline{\boldsymbol{M}}_{\mathcal{I}_j,t-1}^{-1}}^2 \right\} \leq 2d \log \left( 1 + \frac{TL^2}{\lambda d + 4L^2} \right),$$

*where* $\overline{\boldsymbol{M}}_{\mathcal{I}_j,t-1}^{-1} \triangleq \lambda \boldsymbol{I} + \sum_{\tau \in [t-1]: i_\tau \in \mathcal{I}_j} \boldsymbol{x}_{a_\tau} \boldsymbol{x}_{a_\tau}^\mathsf{T} = \lambda \boldsymbol{I} + \sum_{\tau \in [t-1]} \mathbb{1} \{ i_\tau \in \mathcal{I}_j \} \boldsymbol{x}_{a_\tau} \boldsymbol{x}_{a_\tau}^\mathsf{T}.$

*Proof.* The proof mainly follows Lemma 9 in Abbasi-Yadkori et al. (2011), but we improve the bound by leveraging Lemma 2.

Consider the covariance matrix of cluster $\mathcal{I}_j$ after time $T$, we have

$$
\begin{aligned}
&\det\big(\overline{\boldsymbol{M}}_{\mathcal{I}_j,T}\big) \\
&= \det\big(\overline{\boldsymbol{M}}_{\mathcal{I}_j,T-1} + \mathbb{1}\{i_T \in \mathcal{I}_j\}\boldsymbol{x}_{a_T}\boldsymbol{x}_{a_T}^\mathsf{T}\big) \\
&= \det\Big(\overline{\boldsymbol{M}}_{\mathcal{I}_j,T-1}^{\frac{1}{2}}\Big(\boldsymbol{I} + \overline{\boldsymbol{M}}_{\mathcal{I}_j,T-1}^{-\frac{1}{2}}\mathbb{1}\{i_T \in \mathcal{I}_j\}\boldsymbol{x}_{a_T}\boldsymbol{x}_{a_T}^\mathsf{T}\overline{\boldsymbol{M}}_{\mathcal{I}_j,T-1}^{-\frac{1}{2}}\Big)\overline{\boldsymbol{M}}_{\mathcal{I}_j,T-1}^{\frac{1}{2}}\Big) \\
&= \det\big(\overline{\boldsymbol{M}}_{\mathcal{I}_j,T-1}\big)\det\Big(\boldsymbol{I} + \mathbb{1}\{i_T \in \mathcal{I}_j\}\overline{\boldsymbol{M}}_{\mathcal{I}_j,T-1}^{-\frac{1}{2}}\boldsymbol{x}_{a_T}\big(\overline{\boldsymbol{M}}_{\mathcal{I}_j,T-1}^{-\frac{1}{2}}\boldsymbol{x}_{a_T}\big)^\mathsf{T}\Big) \\
&= \det\big(\overline{\boldsymbol{M}}_{\mathcal{I}_j,T-1}\big)\Big(1 + \mathbb{1}\{i_T \in \mathcal{I}_j\}\big\|\overline{\boldsymbol{M}}_{\mathcal{I}_j,T-1}^{-\frac{1}{2}}\boldsymbol{x}_{a_T}\big\|^2\Big) \qquad (10) \\
&= \det\big(\overline{\boldsymbol{M}}_{\mathcal{I}_j,T-1}\big)\Big(1 + \mathbb{1}\{i_T \in \mathcal{I}_j\}\|\boldsymbol{x}_{a_T}\|^2_{\overline{\boldsymbol{M}}_{\mathcal{I}_j,T-1}^{-1}}\Big) \\
&= \det(\overline{\boldsymbol{M}}_{\mathcal{I}_j,T_0})\prod_{t=T_0+1}^{T}\Big(1 + \|\boldsymbol{x}_{a_t}\|^2_{\overline{\boldsymbol{M}}_{\mathcal{I}_j,t-1}^{-1}}\Big), \qquad (11)
\end{aligned}
$$

where Equation 10 is due to the property that all the eigenvalues of a matrix of the form $\boldsymbol{I} + \boldsymbol{x}\boldsymbol{x}^\mathsf{T}$ are one except one eigenvalue, which is $1 + \|\boldsymbol{x}\|^2$. And Equation 11 is by telescoping.

Since we assume the high probability event in Lemma 3 happens, when $t > T_0$, we have

$$
T_{i,t} \geq \max\left\{\frac{8L^2}{\lambda_x}\log\Big(\frac{ud}{\delta}\Big), \frac{512d}{\widetilde{\gamma}^2\lambda_x}\log\Big(\frac{u}{\delta}\Big)\right\} \geq \frac{8L^2}{\lambda_x}\log\Big(\frac{ud}{\delta}\Big), \forall i \in [u].
$$

Note that $\overline{\boldsymbol{M}}_{\mathcal{I}_j,T_0} = \lambda\boldsymbol{I} + \sum_{\tau=1:i_\tau \in \mathcal{I}_j}^{T_0}\boldsymbol{x}_{a_\tau}\boldsymbol{x}_{a_\tau}^\mathsf{T} = \lambda\boldsymbol{I} + \sum_{i \in \mathcal{I}_j}S_{i,T_0}$. So by Lemma 2, we have

$$
\begin{aligned}
\operatorname{tr}\big(\overline{\boldsymbol{M}}_{\mathcal{I}_j,T_0}\big) &= \operatorname{tr}\left(\lambda\boldsymbol{I} + \sum_{i \in \mathcal{I}_j}S_{i,T_0}\right) \\
&\geq \lambda d + \sum_{i \in \mathcal{I}_j}\frac{\lambda_x}{2}\frac{8L^2}{\lambda_x}\log\Big(\frac{ud}{\delta}\Big) \geq \lambda d + 4L^2\log\Big(\frac{ud}{\delta}\Big) \qquad (12)
\end{aligned}
$$

Therefore, taking logarithms on both sides of Equation (11), we have

$$
\log\big(\det\big(\overline{\boldsymbol{M}}_{\mathcal{I}_j,T}\big)\big) = \log(\det\big(\overline{\boldsymbol{M}}_{\mathcal{I}_j,T_0}\big)) + \sum_{t=T_0+1}^{T}\log\Big(1 + \|\boldsymbol{x}_{a_t}\|^2_{\overline{\boldsymbol{M}}_{\mathcal{I}_j,t-1}^{-1}}\Big). \qquad (13)
$$

Since $x \leq 2\log(1+x)$ for $x \in [0,1]$, by Equation 13, we get

$$
\begin{aligned}
\sum_{t=T_0+1}^{T}\min\Big\{1, \|\boldsymbol{x}_{a_t}\|^2_{\overline{\boldsymbol{M}}_{\mathcal{I}_j,t-1}^{-1}}\Big\} &\leq 2\sum_{t=T_0+1}^{T}\log\Big(1 + \|\boldsymbol{x}_{a_t}\|^2_{\overline{\boldsymbol{M}}_{\mathcal{I}_j,t-1}^{-1}}\Big) \\
&= 2\big[\log\big(\det\big(\overline{\boldsymbol{M}}_{\mathcal{I}_j,T}\big)\big) - \log\big(\det\big(\overline{\boldsymbol{M}}_{\mathcal{I}_j,T_0}\big)\big)\big] \qquad (14) \\
&\leq 2\left[d\log\Big(\frac{\operatorname{tr}(\overline{\boldsymbol{M}}_{\mathcal{I}_j,T})}{d}\Big) - d\log\Big(\frac{\operatorname{tr}(\overline{\boldsymbol{M}}_{\mathcal{I}_j,T_0})}{d}\Big)\right] \qquad (15) \\
&\leq 2\left[d\log\Big(\frac{\lambda d + TL^2}{d}\Big) - d\log\Big(\frac{\lambda d + 4L^2\log\big(\frac{ud}{\delta}\big)}{d}\Big)\right] \qquad (16) \\
&\leq 2d\log\Big(1 + \frac{TL^2}{\lambda d + 4L^2}\Big).
\end{aligned}
$$

where Equation (14) uses Equation (13). Equation (15) uses the determinant-trace inequality Lemma 16. Equation (16) uses Equation (12). $\qquad\square$

**Theorem 1** (Regret of `UniCLUB`). *Under the stochastic context setting (Assumptions 1, 2, 3), the expected regret of the `UniCLUB` (Algorithm 1) satisfies:*

$$\mathbb{E}[R(T)] = O\left(\frac{ud}{\widetilde{\gamma}^2 \lambda_x} \log(T) + d\sqrt{mT} \log(T)\right).$$

*Proof.* Define events:

$$\mathcal{E}_1 = \{\text{all users are correctly clustered after } T_0\}.$$

$$\mathcal{E}_2 = \left\{\left|\boldsymbol{x}_a^\mathsf{T}\left(\widehat{\boldsymbol{\theta}}_{V_t, t-1} - \boldsymbol{\theta}^{j(i_t)}\right)\right| \leq \beta \|\boldsymbol{x}_a\|_{\overline{\boldsymbol{M}}_{V_t, t-1}^{-1}}, \forall t \geq T_0\right\}.$$

Let $\mathcal{E} = \mathcal{E}_1 \cap \mathcal{E}_2$. By Lemma 3 and 4, $\mathcal{E}$ happens with probability at least $1 - 4\delta$. Then let $\delta = \frac{1}{T}$ and by the law of total expectations:

$$\mathbb{E}[R(T)] = \mathbb{E}[R(T) \mid \mathcal{E}] \Pr[\mathcal{E}] + \mathbb{E}[R(T) \mid \mathcal{E}^c] \Pr[\mathcal{E}^c]$$

$$\leq \mathbb{E}[R(T) \mid \mathcal{E}] \times 1 + T \times \frac{4}{T}. \tag{17}$$

It remains to bound $\mathbb{E}[R(T) \mid \mathcal{E}]$, i.e., the expected cumulative regret conditioned on event $\mathcal{E}$.

Assume $\mathcal{E}$ happens, by the bounded reward assumption, we can upper bound the regret in the first $T_0$ rounds by $T_0$. After $T_0$, the instantaneous regret at round $t$ satisfies:

$$R_t = \boldsymbol{x}_{a_t^*}^\mathsf{T} \boldsymbol{\theta}^{j(i_t)} - \boldsymbol{x}_{a_t}^\mathsf{T} \boldsymbol{\theta}^{j(i_t)}$$

$$= \boldsymbol{x}_{a_t^*}^\mathsf{T}\left(\boldsymbol{\theta}^{j(i_t)} - \widehat{\boldsymbol{\theta}}_{V_t, t-1}\right) + \left(\boldsymbol{x}_{a_t^*}^\mathsf{T} \widehat{\boldsymbol{\theta}}_{V_t, t-1} + C_{a_t^*, t}\right) - \left(\boldsymbol{x}_{a_t}^\mathsf{T} \widehat{\boldsymbol{\theta}}_{V_t, t-1} + C_{a_t, t}\right)$$

$$+ \boldsymbol{x}_{a_t}^\mathsf{T}\left(\boldsymbol{\theta}^{j(i_t)} - \widehat{\boldsymbol{\theta}}_{V_t, t-1}\right) - C_{a_t^*, t} + C_{a_t, t}$$

$$\leq C_{a_t^*, t} + C_{a_t, t} - C_{a_t^*, t} + C_{a_t, t} = 2C_{a_t, t},$$

where the inequality is due to Lemma 4 and the UCB arm selection strategy of our algorithm.

Therefore, we can bound the cumulative regret conditioned on event $\mathcal{E}$ as follows:

$$R(T) = \sum_{t=1}^{T} R_t = \sum_{t=1}^{T_0} R_t + \sum_{t=T_0+1}^{T} R_t$$

$$\leq T_0 + \sum_{t=T_0+1}^{T} \min\{2, R_t\} \tag{18}$$

$$\leq T_0 + \sum_{j=1}^{m} \sum_{\substack{T_0 < t \leq T \\ i_t \in \mathcal{I}_j}} \min\left\{2, 2\beta \|\boldsymbol{x}_{a_t}\|_{\overline{\boldsymbol{M}}_{\mathcal{I}_t, t-1}^{-1}}\right\} \tag{19}$$

$$\leq T_0 + 2\beta \sum_{j=1}^{m} \sum_{\substack{T_0 < t \leq T \\ i_t \in \mathcal{I}_j}} \min\left\{1, \|\boldsymbol{x}_{a_t}\|_{\overline{\boldsymbol{M}}_{\mathcal{I}_t, t-1}^{-1}}\right\}$$

$$\leq T_0 + 2\beta \sum_{j=1}^{m} \sqrt{T_{\mathcal{I}_j} \sum_{\substack{T_0 < t \leq T \\ i_t \in \mathcal{I}_j}} \min\left\{1, \|\boldsymbol{x}_{a_t}\|_{\overline{\boldsymbol{M}}_{\mathcal{I}_t, t-1}^{-1}}^2\right\}} \tag{20}$$

$$\leq T_0 + 2\beta \sum_{j=1}^{m} \sqrt{T_{\mathcal{I}_j}} \sqrt{2d \log\left(1 + \frac{TL^2}{\lambda d + 4L^2}\right)} \tag{21}$$

$$\leq T_0 + 2\beta \sqrt{mT} \sqrt{2d \log\left(1 + \frac{TL^2}{\lambda d + 4L^2}\right)} \tag{22}$$

$$\leq T_0 + 2\left(\sqrt{d \log\left(1 + \frac{TL^2}{d\lambda}\right) + 2\log\left(\frac{1}{\delta}\right)} + \sqrt{\lambda}\right) \sqrt{2dmT \log\left(1 + \frac{TL^2}{\lambda d + 4L^2}\right)}, \tag{23}$$

where Equation (18) uses $R_t \leq 2$. Equation (19) uses Lemma 4. Equation (20) uses the Cauchy-Schwarz inequality, and we denote the number of times cluster $j$ is selected as $T_{\mathcal{I}_j}$. Equation (21) uses Lemma 5. And Equation (22) is due to the Cauchy-Schwarz inequality and $\sum_{j=1}^m T_{\mathcal{I}_j} = T$.

Let $\delta = \frac{1}{T}$ and plug Equations (23) and (3) into Equation (17), we have

$$
\begin{aligned}
\mathbb{E}[R(T)] &\leq 4 + T_0 + 2\left(\sqrt{d\log\left(1 + \frac{TL^2}{d\lambda}\right)} + 2\log(T) + \sqrt{\lambda}\right)\sqrt{2dmT\log\left(1 + \frac{TL^2}{\lambda d + 4L^2}\right)} \\
&\leq 4 + 16u\log(uT) + 4u\max\left\{\frac{8L^2}{\lambda_x}\log(udT), \frac{512d}{\widetilde{\gamma}^2\lambda_x}\log(uT)\right\} \\
&\quad + 2\left(\sqrt{d\log\left(1 + \frac{TL^2}{d\lambda}\right)} + 2\log(T) + \sqrt{\lambda}\right)\sqrt{2dmT\log\left(1 + \frac{TL^2}{\lambda d + 4L^2}\right)} \\
&= O\left(\frac{ud}{\widetilde{\gamma}^2\lambda_x}\log(T) + d\sqrt{mT}\log(T)\right). \qquad\qquad \Box
\end{aligned}
$$

## C THE SET-BASED ALGORITHM UniSCLUB

### C.1 DETAILS OF THE UniSCLUB ALGORITHM

In this section, we introduce a set-based algorithm named Uniform Exploration Set-based Clustering of Bandits (UniSCLUB), which is inspired by SCLUB Li et al. (2019). Instead of using a graph structure to maintain the clustering information, SCLUB uses a set structure for the same purpose. UniSCLUB inherits the set structure from SCLUB, but incorporates the uniform exploration to enhance its performance. The set structure not only supports the split operations which are similar to those in the graph structure, but also enables the merging of two clusters when the algorithm identifies that their estimated preference vectors are closely aligned. By allowing for both split and merge operations, UniSCLUB can adapt to the underlying clusters more flexibly and expedite the overall clustering process.

The details of UniSCLUB are shown in Algorithm 2. The algorithm maintains information at two levels. At the cluster level, a cluster index $\mathcal{J}$ contains the indices of currently existing clusters, and for each cluster $j \in \mathcal{J}$, the algorithm maintains the set of users $\mathcal{C}^j$ in this cluster and other corresponding information such as the estimated preference vector $\widehat{\boldsymbol{\theta}}^j$. Initially, there is only a single cluster containing all users. At the user level, for each user $i$, the algorithm maintains the estimated preference vector $\widehat{\boldsymbol{\theta}}_{i,t}$ at round $t$ and other corresponding information. Additionally, all users and clusters are associated with a "checked" or "unchecked" status to indicate the estimation accuracy of their preference vectors. UniSCLUB proceeds in phases (Line 1) and each phase $s \in \mathcal{N}_+$ consists of $2^{s-1}$ rounds. At the beginning of each phase, all users revert to the "unchecked" status (Line 2). When a user first appears in a phase, it will be marked as "checked" (Line 14). A cluster is marked as "checked" once all its users are checked. The algorithm will only consider merging checked clusters, so as to avoid premature merging because of inaccurate preference vector estimation. At round $t$, a user $i_t$ comes with a set $\mathcal{A}_t$ of items (Line 4). If $t \leq 2T_0$ (with $T_0$ defined in Equation (24)), UniSCLUB uniformly selects the arm $a_t$ from $\mathcal{A}_t$ (Line 9). Otherwise, it determines the cluster $j$ to which the user $i_t$ belongs and selects the item $a_t$ based on the cluster information (Lines 6 and 7). The algorithm updates the corresponding information of both the user and the cluster after receiving the feedback of the selected item (Lines 10 to 12). Then the algorithm determines whether any split or merge operations are necessary (Lines 13 and 15). If the estimated preference vector of any user within the cluster $j$ diverges from that of user $i_t$, the algorithm will split $i_t$ from the cluster (Algorithm 3). If the estimated preference vectors of two checked clusters are closely aligned, a merge operation will be performed (Algorithm 4).

Since no clustering information is utilized during the uniform exploration period, UniSCLUB can only update the user-level information. Then the algorithm clusters all users at round $2T_0$, and continues updating both user and cluster information subsequently. This implementation makes UniSCLUB more efficient and robust compared to SCLUB.

---

**Algorithm 2:** `UniSCLUB`: Uniform Exploration Set-based Clustering of Bandits

---

**Input:** $\lambda$, $\beta$, $\lambda_x$, $\delta$, $L$, $\widetilde{\gamma}$

**Initialization:** Initialize the cluster indexes by $\mathcal{J} = \{1\}$.

Let $\boldsymbol{M}^1 = \boldsymbol{0}_{d \times d}, \overline{\boldsymbol{M}}^1 = \lambda \boldsymbol{I}, \boldsymbol{b}^0 = \boldsymbol{0}_{d \times 1}, T^0 = 0, \mathcal{C}^1 = [u]$.

Let $\boldsymbol{S}_{i,0} = \boldsymbol{0}_{d \times d}, \boldsymbol{b}_{i,0} = \boldsymbol{0}_{d \times 1}, T_{i,0} = 0, \forall i \in [u]$.

Define $f(T_{i,t}) = (\sqrt{2 \log(u/\delta) + d \log(1 + \frac{T_{i,t} L^2}{\lambda d})} + \sqrt{\lambda})/\sqrt{\lambda + T_{i,t} \lambda_x / 2}$.

Define $T_0 \triangleq 16u \log\left(\frac{u}{\delta}\right) + 4u \max\left\{ \frac{8L^2}{\lambda_x} \log\left(\frac{ud}{\delta}\right), \frac{512d}{\widetilde{\gamma}^2 \lambda_x} \log\left(\frac{u}{\delta}\right) \right\}$.

**1** **for** $s = 1, 2, \ldots$ **do**

**2**    Mark every user unchecked for each cluster.

**3**    **for** $t = 2^{s-1}, \ldots, 2^s - 1$ *(terminate when $t > T$)* **do**

**4**      Receive user index $i_t$ and arm set $\mathcal{A}_t$

**5**      **if** $t > 2T_0$ **then**

**6**        Find the cluster $j \in \mathcal{J}$ satisfying $i_t \in C^j$

**7**        Select arm $a_t = \arg\max_{a \in \mathcal{A}_t} (\widehat{\boldsymbol{\theta}}^j)^\mathsf{T} \boldsymbol{x}_a + \beta \sqrt{\boldsymbol{x}_a^\mathsf{T} (\overline{\boldsymbol{M}}^j)^{-1} \boldsymbol{x}_a}$

**8**      **else**

**9**        Select $a_t$ uniformly at random from $\mathcal{A}_t$

**10**      Receive reward $r_t$

**11**      Update statistics for user $i_t$, others remain unchanged:
$$\boldsymbol{S}_{i_t,t} = \boldsymbol{S}_{i_t,t-1} + \boldsymbol{x}_{a_t} \boldsymbol{x}_{a_t}^\mathsf{T}, \quad \boldsymbol{b}_{i_t,t} = \boldsymbol{b}_{i_t,t-1} + r_t \boldsymbol{x}_{a_t}$$
$$T_{i_t,t} = T_{i_t,t-1} + 1, \quad \widehat{\boldsymbol{\theta}}_{i_t,t} = (\lambda \boldsymbol{I} + \boldsymbol{S}_{i_t,t})^{-1} \boldsymbol{b}_{i_t,t}$$

**12**      Update statistics for cluster $j$, others remain unchanged:
$$\boldsymbol{M}^j = \boldsymbol{M}^j + \boldsymbol{x}_{a_t} \boldsymbol{x}_{a_t}^\mathsf{T}, \quad \boldsymbol{b}^j = \boldsymbol{b}^j + r_t \boldsymbol{x}_{a_t}$$
$$T^j = T^j + 1, \quad \overline{\boldsymbol{M}}^j = \boldsymbol{M}^j + \lambda \boldsymbol{I}, \quad \widehat{\boldsymbol{\theta}}^j = (\overline{\boldsymbol{M}}^j)^{-1} \boldsymbol{b}^j$$

**13**      Run `Split`

**14**      Mark user $i_t$ as checked

**15**      Run `Merge`

---

**Algorithm 3:** Split

---

**1** **if** $\exists i' \in \mathcal{C}^j$ s.t. $\left\| \widehat{\boldsymbol{\theta}}_{i_t,t} - \widehat{\boldsymbol{\theta}}_{i',t} \right\| > f(T_{i_t,t}) + f(T_{i',t})$ **then**

**2**    Split user $i_t$ from the cluster $j$:
$$\boldsymbol{M}^j = \boldsymbol{M}^j - \boldsymbol{S}_{i_t,t}, \quad \boldsymbol{b}^j = \boldsymbol{b}^j - \boldsymbol{b}_{i_t,t}, \quad T^j = T^j - T_{i_t,t}, \quad \mathcal{C}^j = \mathcal{C}^j \setminus \{i_t\}$$
$$\overline{\boldsymbol{M}}^j = \boldsymbol{M}^j + \lambda \boldsymbol{I}, \quad \widehat{\boldsymbol{\theta}}^j = (\overline{\boldsymbol{M}}^j)^{-1} \boldsymbol{b}^j$$

**3**    Generate a new cluster $j'$ containing only user $i_t$:
$$\boldsymbol{M}^{j'} = \boldsymbol{S}_{i_t,t}, \quad \boldsymbol{b}^{j'} = \boldsymbol{b}_{i_t,t}, \quad T^{j'} = T_{i_t,t}, \quad \mathcal{C}^{j'} = \{i_t\} \; \overline{\boldsymbol{M}}^{j'} = \boldsymbol{M}^{j'} + \lambda \boldsymbol{I}, \quad \widehat{\boldsymbol{\theta}}^{j'} = \widehat{\boldsymbol{\theta}}_{i_t,t}$$

**4**    $\mathcal{J} = \mathcal{J} \cup \{j'\}$

---

## C.2   THEORETICAL ANALYSIS OF UNISCLUB

**Theorem 2** (Regret of `UniSCLUB`). *Under the stochastic context setting (Assumptions 1, 2, 3), the expected regret of the `UniSCLUB` (Algorithm 2) satisfies:*

$$\mathbb{E}[R(T)] = O\left( \frac{ud}{\widetilde{\gamma}^2 \lambda_x} \log(T) + d\sqrt{mT} \log(T) \right).$$

*Proof.* From Lemma 6, Algorithm 2 will have correct clusers after $2T_0$ with probability $\geq 1 - 3\delta$.

---

**Algorithm 4:** Merge

---
1 **for** *any two checked clusters* $j_1, j_2 \in \mathcal{J}$ **do**
2      **if** $\left\| \widehat{\boldsymbol{\theta}}^{j_1} - \widehat{\boldsymbol{\theta}}^{j_2} \right\| < f(T^{j_1}) + f(T^{j_2})$ **then**
3          Merge clusters $j_1$ and $j_2$:
            $\boldsymbol{M}^{j_1} = \boldsymbol{M}^{j_1} + \boldsymbol{M}^{j_2}, \ \boldsymbol{b}^{j_1} = \boldsymbol{b}^{j_1} + \boldsymbol{b}^{j_2}, \ T^{j_1} = T^{j_1} + T^{j_2}, \ \mathcal{C}^{j_1} = \mathcal{C}^{j_1} \cup \mathcal{C}^{j_2}$
            $\overline{\boldsymbol{M}}^{j_1} = \boldsymbol{M}^{j_1} + \lambda \boldsymbol{I}, \ \widehat{\boldsymbol{\theta}}^{j_1} = (\overline{\boldsymbol{M}}^{j_1})^{-1} \boldsymbol{b}^{j_1}$
4          $\mathcal{J} = \mathcal{J} \setminus \{j_2\}$

---

Based on Lemma 6 and with $T_0$ replaced by $2T_0$, we can derive the counterparts of Lemma 4 and Lemma 5. Then, similar to the proof of Theorem 1, we have

$$\mathbb{E}[R(T)] \leq 4 + 2T_0 + 2\left( \sqrt{d \log\left(1 + \frac{TL^2}{d\lambda}\right) + 2\log(T)} + \sqrt{\lambda} \right) \sqrt{2dmT \log\left(1 + \frac{TL^2}{\lambda d + 4L^2}\right)}$$

$$\leq 4 + 32u \log(uT) + 8u \max\left\{ \frac{8L^2}{\lambda_x} \log(udT), \frac{512d}{\widetilde{\gamma}^2 \lambda_x} \log(uT) \right\}$$

$$+ 2\left( \sqrt{d \log\left(1 + \frac{TL^2}{d\lambda}\right) + 2\log(T)} + \sqrt{\lambda} \right) \sqrt{2dmT \log\left(1 + \frac{TL^2}{\lambda d + 4L^2}\right)}$$

$$= O\left( \frac{ud}{\widetilde{\gamma}^2 \lambda_x} \log(T) + d\sqrt{mT} \log(T) \right). \qquad \square$$

**Lemma 6.** *With probability at least* $1 - 3\delta$, *Algorithm 2 can cluster all the users correctly after* $2T_0$, *where*

$$T_0 \triangleq 16u \log\left(\frac{u}{\delta}\right) + 4u \max\left\{ \frac{8L^2}{\lambda_x} \log\left(\frac{ud}{\delta}\right), \frac{512d}{\widetilde{\gamma}^2 \lambda_x} \log\left(\frac{u}{\delta}\right) \right\}. \qquad (24)$$

*Proof.* From Lemma 3, with probability at least $1 - 3\delta$, we have $\left\| \widehat{\boldsymbol{\theta}}_{i,t} - \boldsymbol{\theta}^{j(i)} \right\|_2 \leq f(T_{i,t}) \leq \frac{\widetilde{\gamma}}{4}, \forall i \in [u]$, when $t \geq T_0$.

We now show that under this condition, Algorithm 2 will split well, i.e, the current clusters are subsets of true clusters. To guarantee this, at round $t$, for the user $i_t$ and the corresponding cluster $j$, we need to verify that: (1) if the current cluster $j$ is a subset of the ground-truth cluster of user $i_t$, then user $i_t$ will not be split from the cluster $j$. (2) if the current cluster $j$ contains users that are not in the same ground-truth cluster as user $i_t$, then user $i_t$ will be split from the cluster $j$.

To prove (1), we show the contrapositive of (1): if user $i_t$ is split from the cluster $j$, the cluster $j$ contains users such that $i_t$ and these users are from different ground-truth clusters.

If user $i_t$ is split from the cluster $j$ by Algorithm 3, i.e., there exists some user $i' \in \mathcal{C}^j$ such that $\left\| \widehat{\boldsymbol{\theta}}_{i_t,t} - \widehat{\boldsymbol{\theta}}_{i',t} \right\| > f(T_{i_t,t}) + f(T_{i',t})$, due to the triangle inequality, we have

$$\|\boldsymbol{\theta}^{j(i_t)} - \boldsymbol{\theta}^{j(i')}\|_2 \geq \|\widehat{\boldsymbol{\theta}}_{i_t,t} - \widehat{\boldsymbol{\theta}}_{i',t}\|_2 - \|\widehat{\boldsymbol{\theta}}_{i_t,t} - \boldsymbol{\theta}^{j(i_t)}\|_2 - \|\widehat{\boldsymbol{\theta}}_{i',t} - \boldsymbol{\theta}^{j(i')}\|_2$$

$$\geq \|\widehat{\boldsymbol{\theta}}_{i_t,t} - \widehat{\boldsymbol{\theta}}_{i',t}\|_2 - f(T_{i_t,t}) - f(T_{i',t}) > 0.$$

By Assumption 3, $\left\| \boldsymbol{\theta}^{j(i_t)} - \boldsymbol{\theta}^{j(i')} \right\|_2 > 0$ implies that users $i_t, i'$ are not in the same true cluster.

To prove (2), we show that if there exists some user $i' \in \mathcal{C}^j$ and $\left\| \boldsymbol{\theta}^{j(i_t)} - \boldsymbol{\theta}^{j(i')} \right\|_2 > \widetilde{\gamma}$, we have

$$\left\| \widehat{\boldsymbol{\theta}}_{i_t,t} - \widehat{\boldsymbol{\theta}}_{i',t} \right\|_2 \geq \left\| \boldsymbol{\theta}^{j(i_t)} - \boldsymbol{\theta}^{j(i')} \right\|_2 - \left\| \widehat{\boldsymbol{\theta}}_{i_t,t} - \boldsymbol{\theta}^{j(i_t)} \right\|_2 - \left\| \widehat{\boldsymbol{\theta}}_{i',t} - \boldsymbol{\theta}^{j(i')} \right\|_2$$

$$> \widetilde{\gamma} - \frac{\widetilde{\gamma}}{4} - \frac{\widetilde{\gamma}}{4} = \frac{\widetilde{\gamma}}{2} \geq f(T_{i_t,t}) + f(T_{i',t}),$$

which satisfies the condition of splitting. Thus, user $i_t$ will be split out from the current cluster $j$.

Therefore, when $t > T_0$, each existing cluster at $t$ will not contain users from different true clusters.

Now we show that Algorithm 2 will merge well so that only correct clusters remain. For two checked clusters $j_1$ and $j_2$, we need to verify that: (1) if $j_1$ and $j_2$ are not merged, they are from different true clusters. (2) if $j_1$ and $j_2$ are merged, they are from the same true cluster.

We only consider the case where the existing clusters are subsets of ground-truth clusters because of splitting. For convenience, we denote the true preference vectors of clusters $j_1$ and $j_2$ by $\boldsymbol{\theta}^{j_1}$ and $\boldsymbol{\theta}^{j_2}$, which is reasonable since each of they only contains users with the same preference vector.

To prove (1), if $j_1$ and $j_2$ are not merged, i.e., $\left\|\widehat{\boldsymbol{\theta}}^{j_1} - \widehat{\boldsymbol{\theta}}^{j_2}\right\| \geq f(T^{j_1}) + f(T^{j_2})$, by the triangle inequality, we have

$$\|\boldsymbol{\theta}^{j_1} - \boldsymbol{\theta}^{j_2}\|_2 \geq \|\widehat{\boldsymbol{\theta}}^{j_1} - \widehat{\boldsymbol{\theta}}^{j_2}\|_2 - \|\widehat{\boldsymbol{\theta}}^{j_1} - \boldsymbol{\theta}^{j_1}\|_2 - \|\widehat{\boldsymbol{\theta}}^{j_2} - \boldsymbol{\theta}^{j_2}\|_2$$
$$\geq \|\widehat{\boldsymbol{\theta}}^{j_1} - \widehat{\boldsymbol{\theta}}^{j_2}\|_2 - f(T^{j_1}) - f(T^{j_2}) > 0,$$

which implies that $j_1$ and $j_2$ are from two different true clusters.

To prove (2), we show a contraction: if $j_1$ and $j_2$ are merged, but they are from different true clusters, i.e., $\|\boldsymbol{\theta}^{j_1} - \boldsymbol{\theta}^{j_2}\|_2 > \widetilde{\gamma}$, we have

$$\|\widehat{\boldsymbol{\theta}}^{j_1} - \widehat{\boldsymbol{\theta}}^{j_2}\|_2 \geq \|\boldsymbol{\theta}^{j_1} - \boldsymbol{\theta}^{j_2}\|_2 - \|\widehat{\boldsymbol{\theta}}^{j_1} - \boldsymbol{\theta}^{j_1}\|_2 - \|\widehat{\boldsymbol{\theta}}^{j_2} - \boldsymbol{\theta}^{j_2}\|_2$$
$$\geq \widetilde{\gamma} - f(T^{j_1}) - f(T^{j_2}) \geq \frac{\widetilde{\gamma}}{2} \geq (f(T^{j_1}) + f(T^{j_2})),$$

because that $f(T^{j_1}) + f(T^{j_2}) \leq \frac{\widetilde{\gamma}}{2}$. However, this contradicts the condition of merging. Therefore, if $j_1$ and $j_2$ are merged, they are from the same true cluster.

We double the time $T_0$ to $2T_0$ to ensure that all users in each cluster can be checked and provide sufficient time for the split and merge operations. Therefore, after $2T_0$, all the clusters are the ground-truth clusters with probability at least $1 - 3\delta$. $\qquad\square$

## D    THE PHASE-BASED ALGORITHM WITHOUT KNOWLEDGE OF $\gamma$

### D.1    DETAILS OF THE PHASEUNICLUB ALGORITHM

In this section, we provide an algorithm PhaseUniCLUB (in Algorithm 5) for the online clustering of bandits when the preference vectors' distance $\gamma$ between clusters is completely unknown. We also theoretically provide an upper bound of its regret in Theorem 3.

For convenience, we first define the following notations used in PhaseUniCLUB:

$$T^{\text{init}} \triangleq 16u \log\left(\frac{u}{\delta}\right) + 4u \cdot \frac{8L^2}{\lambda_x} \log\left(\frac{ud}{\delta}\right), \quad T^{(s)} \triangleq 4u \cdot \frac{512d}{2^{-s}\lambda_x} \log\left(\frac{u}{\delta}\right). \tag{25}$$

Same as UniCLUB (in Algorithm 1), PhaseUniCLUB also maintains a dynamic undirected graph $G_t = ([u], E_t)$ over all users for the purpose of clustering. However, to cope with the unknown $\gamma$, PhaseUniCLUB leverages the idea of the doubling trick. Specifically, as depicted in Algorithm 5, the algorithm first runs for $T^{\text{init}}$ rounds with uniform exploration (Line 1 to Line 5). Then, it proceeds in phases. For each phase $s = 0, 1, \ldots$, we divide it into two subphases: (1) the first subphase (named the exploration subphase), consisting of $T^{(s)}$ rounds, where PhaseUniCLUB selects an item uniformly from the item set for each coming user (Line 11); (2) the second subphase (named the UCB subphase), containing $(2^{s/2} - 1)T^{(s)}$ rounds, where PhaseUniCLUB identifies the cluster of each coming user (Line 13) and selects an item based on the estimated preference vector of the cluster (Line 16). Meanwhile, in each round, after receiving the reward feedback (Line 17), the algorithm updates statistics for user $i_t$ (Line 18), and determine whether to delete any edge between $i_t$ and its neighbors (Line 19).

Due to insufficient information about the distance gap $\gamma$, PhaseUniCLUB keeps uniformly exploring users' item sets. Unlike UniCLUB, PhaseUniCLUB distributes the uniform exploration

---

**Algorithm 5:** `PhaseUniCLUB`: Phase-based Uniform Exploration Clustering of Bandits

---

**Input:** $\lambda$, $\beta$, $\lambda_x$, $\delta$, $L$
**Initialization:** Let $G_0 = ([u], E_0)$ be a complete graph.
Let $\boldsymbol{S}_{i,0} = \boldsymbol{0}_{d \times d}, \boldsymbol{b}_{i,0} = \boldsymbol{0}_{d \times 1}, T_{i,0} = 0, \forall i \in [u]$.
Define $f(T_{i,t}) = (\sqrt{2 \log(u/\delta) + d \log(1 + \frac{T_{i,t} L^2}{\lambda d})} + \sqrt{\lambda})/\sqrt{\lambda + T_{i,t} \lambda_x / 2}$.

1   **for** $t = 1, 2, \ldots, T^{init}$ **do**
2     Receive user index $i_t$ and arm set $\mathcal{A}_t$
3     Select $a_t$ uniformly at random from $\mathcal{A}_t$
4     Receive reward $r_t$
5     Update statistics for user $i_t$: $\boldsymbol{S}_{i_t,t}, \boldsymbol{b}_{i_t,t}, T_{i_t,t}, \widehat{\boldsymbol{\theta}}_{i_t,t}$

6   **for** $s = 0, 1, \ldots$ **do**
7     **for** $\tau = 1, 2, \ldots, 2^{\frac{s}{2}} \cdot T^{(s)}$ *(terminate when $t > T$)* **do**
8       $t = t + 1$
9       Receive user index $i_t$ and arm set $\mathcal{A}_t$
10      **if** $\tau \leq T^{(s)}$ **then**
11        Select $a_t$ uniformly at random from $\mathcal{A}_t$
12      **else**
13        Find all neighbors of user $i_t$ in $G_{t-1}$ and include $i_t$ to form the cluster $V_t$
14        $\boldsymbol{M}_{V_t,t-1} = \sum_{i \in V_t} \boldsymbol{S}_{i,t-1}, \overline{\boldsymbol{M}}_{V_t,t-1} = \lambda \boldsymbol{I} + \boldsymbol{M}_{V_t,t-1}$
15        $\boldsymbol{b}_{V_t,t-1} = \sum_{i \in V_t} \boldsymbol{b}_{i,t-1}, \widehat{\boldsymbol{\theta}}_{V_t,t-1} = \overline{\boldsymbol{M}}_{V_t,t-1}^{-1} \boldsymbol{b}_{V_t,t-1}, T_{V_t,t-1} = \sum_{i \in V_t} T_{i,t}$
16        Select arm $a_t = \arg\max_{a \in \mathcal{A}_t} \widehat{\boldsymbol{\theta}}_{V_t,t-1}^{\mathsf{T}} \boldsymbol{x}_a + \beta \sqrt{\boldsymbol{x}_a^{\mathsf{T}} \overline{\boldsymbol{M}}_{V_t,t-1}^{-1} \boldsymbol{x}_a}$
17      Receive reward $r_t$
18      Update statistics for user $i_t$: $\boldsymbol{S}_{i_t,t}, \boldsymbol{b}_{i_t,t}, T_{i_t,t}, \widehat{\boldsymbol{\theta}}_{i_t,t}$
19      Delete edge $(i_t, \ell) \in E_{t-1}$ if

$$\left\| \widehat{\boldsymbol{\theta}}_{i_t,t} - \widehat{\boldsymbol{\theta}}_{\ell,t} \right\| > f(T_{i_t,t}) + f(T_{\ell,t})$$

      and obtain an updated graph $G_t = ([u], E_t)$

---

across all phases. During the exploration subphase of each phase $s$, `PhaseUniCLUB` focuses on estimating the users' preference vectors to a precision level of $\gamma_s$. In the subsequent UCB subphase, it clusters users based on the current precision level. If $\gamma_s > \gamma$, there is a risk of users being incorrectly assigned to clusters they are not belonging to. To mitigate this risk of misclustering, when a user appears, `PhaseUniCLUB` only identifies the neighbors of the user to form a cluster, rather than finding the user's entire connected component as in `UniCLUB`. Additionally, with carefully chosen phase lengths, the regret of `PhaseUniCLUB` can be bounded sublinearly. The approach of identifying only neighbors is also applied in Wang et al. (2023a), but to avoid misclustering due to model misspecification. Moreover, `PhaseUniCLUB` achieves better regret bound compared to that in Wang et al. (2023a). While their regret term associated with this error is linear with respect to $T$, ours scales as $T^{2/3}$.

### D.2 THEORETICAL ANALYSIS OF PHASEUNICLUB

To bound the cumulative regret of `PhaseUniCLUB`, we present the following lemmas.

**Lemma 7.** *Denote* $\gamma_s \triangleq 2^{-\frac{s}{2}}$ *and* $C_p = \frac{\sqrt{2}\lambda_x}{1024 u d}$. *With probability at least* $1 - 3\log_2\left(C_p \frac{T}{\log(u/\delta)} + 1\right)\delta$, *after the exploration subphase in any phase* $s = 0, 1, \ldots$, *for all users* $i \in [u]$, *we have*

$$\left\| \widehat{\boldsymbol{\theta}}_{i,t} - \boldsymbol{\theta}^{j(i)} \right\|_2 \leq \frac{\gamma_s}{4}.$$

*Proof.* According to Lemma 3, with probability $\geq 1 - 3\delta$, after the exploration subphase in phase $s = 0, 1, \ldots$, for any user $i \in [u]$, we have

$$\left\|\widehat{\boldsymbol{\theta}}_{i,t} - \boldsymbol{\theta}^{j(i)}\right\|_2 \leq \frac{\gamma_s}{4}.$$

Denote the number of phases for `PhaseUniCLUB` by $N_p$. Since phase $s$ contains $2^{s/2}T^{(s)}$ rounds, $N_p$ is the maximum integer satisfying

$$T^{\text{init}} + \sum_{s=0}^{N_p-1} 2^{s/2} \cdot 4u \cdot \frac{512d}{2^{-s}\lambda_x} \log\left(\frac{u}{\delta}\right) \leq T.$$

By some calculations, we have

$$N_p \leq \frac{2}{3} \log_2 \left( \frac{\sqrt{2}\lambda_x (T - T^{\text{init}})}{1024ud \log \frac{u}{\delta}} + 1 \right) = \log_2 \left( \frac{C_p(T - T^{\text{init}})}{\log \frac{u}{\delta}} + 1 \right),$$

where we denote $C_p = \frac{\sqrt{2}\lambda_x}{1024ud}$.

Applying a union bound over all phases, we have that with probability at least $1 - 3\log_2\left(C_p \frac{T}{\log(u/\delta)} + 1\right)\delta$, after the exploration subphase in any phase $s = 0, 1, \ldots$, for all users $i \in [u]$, $\|\widehat{\boldsymbol{\theta}}_{i,t} - \boldsymbol{\theta}^{j(i)}\|_2 \leq \frac{\gamma_s}{4}$ holds. $\qquad\square$

**Lemma 8.** *With probability at least* $1 - 3\log_2\left(C_p \frac{T}{\log(u/\delta)} + 1\right)\delta$, *in any phase $s$, for two users* $i_1, i_2 \in V_t$, *we have*

$$\left\|\boldsymbol{\theta}^{j(i_1)} - \boldsymbol{\theta}^{j(i_2)}\right\|_2 \leq \gamma_s.$$

*Proof.* In phase $s$, for any two users $i_1, i_2 \in V_t$, it satisfies $\left\|\widehat{\boldsymbol{\theta}}_{i_1,t} - \widehat{\boldsymbol{\theta}}_{i_2,t}\right\| \leq f(T_{i_1,t}) + f(T_{i_2,t})$. By the triangle inequality and Lemma 8, with probability at least $1 - 3\log_2\left(C_p \frac{T}{\log(u/\delta)} + 1\right)\delta$, we have

$$\left\|\boldsymbol{\theta}^{j(i_1)} - \boldsymbol{\theta}^{j(i_2)}\right\|_2 \leq \left\|\boldsymbol{\theta}^{j(i_1)} - \widehat{\boldsymbol{\theta}}_{i_1,t}\right\|_2 + \left\|\widehat{\boldsymbol{\theta}}_{i_2,t} - \boldsymbol{\theta}^{j(i_2)}\right\|_2 + \left\|\widehat{\boldsymbol{\theta}}_{i_1,t} - \widehat{\boldsymbol{\theta}}_{i_2,t}\right\|_2$$
$$\leq \frac{\gamma_s}{4} + \frac{\gamma_s}{4} + f(T_{i_1,t}) + f(T_{i_2,t}) \leq \gamma_s. \qquad\square$$

**Lemma 9.** *With probability at least* $1 - 4\log_2\left(C_p \frac{T}{\log(u/\delta)} + 1\right)\delta$, *in any phase $s$, we have*

$$\left| \boldsymbol{x}_a^{\mathsf{T}}\left( \widehat{\boldsymbol{\theta}}_{V_t,t-1} - \boldsymbol{\theta}^{j(i_t)} \right) \right| \leq C_{a,t} + \frac{2L^2\gamma_s}{\lambda_x} \mathbb{1}\{\gamma_s > \gamma\},$$

*where* $C_{a,t} \triangleq \beta \|\boldsymbol{x}_a\|_{\overline{\boldsymbol{M}}_{V_t,t-1}^{-1}}$ *and* $\beta = \sqrt{d\log(1 + \frac{TL^2}{d\lambda}) + 2\log(\frac{1}{\delta})} + \sqrt{\lambda}$.

*Proof.* In phase $s$, the confidence radius for users' preference vectors is $\frac{\gamma_s}{4}$. If $\gamma_s > \gamma$, user $i_t$ may be mistakenly grouped into the wrong cluster, i.e., $V_t \neq V_{j(i_t)}$. Accounting for this error, we analyze the following two cases:

**Case 1**: user $i_t$ is correctly clustered, i.e., $V_t = V_{j(i_t)}$. Same as Lemma 4, we have

$$\left| \boldsymbol{x}_a^{\mathsf{T}}\left( \widehat{\boldsymbol{\theta}}_{V_t,t-1} - \boldsymbol{\theta}^{j(i_t)} \right) \right| \leq \beta \|\boldsymbol{x}_a\|_{\overline{\boldsymbol{M}}_{V_t,t-1}^{-1}},$$

with probability $1 - 4\log_2\left(C_p \frac{T}{\log(u/\delta)} + 1\right)\delta$.

**Case 2**: user $i_t$ is mistakenly clustered, i.e., $V_t \neq V_{j(i_t)}$, which happens when $\gamma_s > \gamma$. Then,

$$\widehat{\boldsymbol{\theta}}_{V_t,t-1} - \boldsymbol{\theta}^{j(i_t)} = \left( \lambda I + \sum_{\tau \in [t-1]:i_\tau \in V_t} \boldsymbol{x}_{a_\tau}\boldsymbol{x}_{a_\tau}^{\mathsf{T}} \right)^{-1} \left( \sum_{\tau \in [t-1]:i_\tau \in V_t} r_\tau \boldsymbol{x}_{a_\tau} \right) - \boldsymbol{\theta}^{j(i_t)}$$

$$= \left( \lambda I + \sum_{\tau \in [t-1]:i_\tau \in V_t} \boldsymbol{x}_{a_\tau} \boldsymbol{x}_{a_\tau}^\mathsf{T} \right)^{-1} \left( \sum_{\tau \in [t-1]:i_\tau \in V_t} \boldsymbol{x}_{a_\tau} (\boldsymbol{x}_{a_\tau}^\mathsf{T} \boldsymbol{\theta}_{i_\tau} + \eta_\tau) \right) - \boldsymbol{\theta}^{j(i_t)}$$

$$= \overline{\boldsymbol{M}}_{V_t,t-1}^{-1} \left( \sum_{\tau \in [t-1]:i_\tau \in V_t} \boldsymbol{x}_{a_\tau} \boldsymbol{x}_{a_\tau}^\mathsf{T} (\boldsymbol{\theta}_{i_\tau} - \boldsymbol{\theta}_{i_t}) + \sum_{\tau \in [t-1]:i_\tau \in V_t} \boldsymbol{x}_{a_\tau} \boldsymbol{x}_{a_\tau}^\mathsf{T} \boldsymbol{\theta}_{i_t} + \sum_{\tau \in [t-1]:i_\tau \in V_t} \boldsymbol{x}_{a_\tau} \eta_\tau \right) - \boldsymbol{\theta}^{j(i_t)}$$

$$= \overline{\boldsymbol{M}}_{V_t,t-1}^{-1} \left( \left( \lambda \boldsymbol{I} + \sum_{\tau \in [t-1]:i_\tau \in V_t} \boldsymbol{x}_{a_\tau} \boldsymbol{x}_{a_\tau}^\mathsf{T} \right) \boldsymbol{\theta}_{i_t} - \lambda \boldsymbol{\theta}_{i_t} + \sum_{\tau \in [t-1]:i_\tau \in V_t} \boldsymbol{x}_{a_\tau} \eta_\tau \right) - \boldsymbol{\theta}^{j(i_t)}$$

$$+ \overline{\boldsymbol{M}}_{V_t,t-1}^{-1} \sum_{\tau \in [t-1]:i_\tau \in V_t} \boldsymbol{x}_{a_\tau} \boldsymbol{x}_{a_\tau}^\mathsf{T} (\boldsymbol{\theta}_{i_\tau} - \boldsymbol{\theta}_{i_t})$$

$$= - \lambda \overline{\boldsymbol{M}}_{V_t,t-1}^{-1} \boldsymbol{\theta}^{j(i_t)} + \overline{\boldsymbol{M}}_{V_t,t-1}^{-1} \sum_{\tau \in [t-1]:i_\tau \in V_t} \boldsymbol{x}_{a_\tau} \eta_\tau + \overline{\boldsymbol{M}}_{V_t,t-1}^{-1} \sum_{\tau \in [t-1]:i_\tau \in V_t} \boldsymbol{x}_{a_\tau} \boldsymbol{x}_{a_\tau}^\mathsf{T} (\boldsymbol{\theta}_{i_\tau} - \boldsymbol{\theta}_{i_t}),$$

where we denote $\overline{\boldsymbol{M}}_{V_t,t-1} = \lambda I + \boldsymbol{M}_{V_t,t-1}$.

Then from Lemma 4, we have

$$\left| \boldsymbol{x}_a^\mathsf{T} \left( \widehat{\boldsymbol{\theta}}_{V_t,t-1} - \boldsymbol{\theta}^{j(i_t)} \right) \right| \leq \beta \|\boldsymbol{x}_a\|_{\overline{\boldsymbol{M}}_{V_t,t-1}^{-1}} + \left| \boldsymbol{x}_a^\mathsf{T} \overline{\boldsymbol{M}}_{V_t,t-1}^{-1} \sum_{\tau \in [t-1]:i_\tau \in V_t} \boldsymbol{x}_{a_\tau} \boldsymbol{x}_{a_\tau}^\mathsf{T} (\boldsymbol{\theta}_{i_\tau} - \boldsymbol{\theta}_{i_t}) \right|. \quad (26)$$

Now we bound the second term in Equation (26).

$$\left| \boldsymbol{x}_a^\mathsf{T} \overline{\boldsymbol{M}}_{V_t,t-1}^{-1} \sum_{\tau \in [t-1]:i_\tau \in V_t} \boldsymbol{x}_{a_\tau} \boldsymbol{x}_{a_\tau}^\mathsf{T} (\boldsymbol{\theta}_{i_\tau} - \boldsymbol{\theta}_{i_t}) \right|$$

$$\leq \|\boldsymbol{x}_a\|_2 \left\| \overline{\boldsymbol{M}}_{V_t,t-1}^{-1} \right\|_2 \left\| \sum_{\tau \in [t-1]:i_\tau \in V_t} \boldsymbol{x}_{a_\tau} \boldsymbol{x}_{a_\tau}^\mathsf{T} (\boldsymbol{\theta}_{i_\tau} - \boldsymbol{\theta}_{i_t}) \right\|_2 \quad (27)$$

$$\leq L \left\| \overline{\boldsymbol{M}}_{V_t,t-1}^{-1} \right\|_2 \sum_{\tau \in [t-1]:i_\tau \in V_t} \left\| \boldsymbol{x}_{a_\tau} \boldsymbol{x}_{a_\tau}^\mathsf{T} \right\|_2 \|\boldsymbol{\theta}_{i_\tau} - \boldsymbol{\theta}_{i_t}\|_2 \quad (28)$$

$$\leq L \gamma_s \lambda_{\max}(\overline{\boldsymbol{M}}_{V_t,t-1}^{-1}) \sum_{\tau \in [t-1]:i_\tau \in V_t} \left\| \boldsymbol{x}_{a_\tau} \boldsymbol{x}_{a_\tau}^\mathsf{T} \right\|_2 \quad (29)$$

$$\leq L \gamma_s \frac{\sum_{\tau \in [t-1]:i_\tau \in V_t} L}{\lambda_{\min}(\overline{\boldsymbol{M}}_{V_t,t-1})} \quad (30)$$

$$\leq L \gamma_s \frac{T_{V_t,t-1} L}{\lambda_x T_{V_t,t-1}/2 + \lambda} \quad (31)$$

$$\leq \frac{2L^2 \gamma_s}{\lambda_x},$$

where $\left\| \overline{\boldsymbol{M}}_{V_t,t-1}^{-1} \right\|_2$ and $\left\| \boldsymbol{x}_{a_\tau} \boldsymbol{x}_{a_\tau}^\mathsf{T} \right\|_2$ are the spectral norm of matrix $\overline{\boldsymbol{M}}_{V_t,t-1}^{-1}$ and $\boldsymbol{x}_{a_\tau} \boldsymbol{x}_{a_\tau}^\mathsf{T}$. Equation (27) is because of the Cauchy–Schwarz inequality and the induced matrix norm inequality. Equation (28) is due to $\|\boldsymbol{x}_a\|_2 \leq L$ and the induced matrix norm inequality. Equation (29) follows Lemma 8 and that $\overline{\boldsymbol{M}}_{V_t,t-1}^{-1}$ is PSD. Equation (30) is because of $\left\| \boldsymbol{x}_{a_\tau} \boldsymbol{x}_{a_\tau}^\mathsf{T} \right\|_2 = L$ and Equation (31) follows Lemma 2.

Therefore, we have

$$\left| \boldsymbol{x}_a^\mathsf{T} \left( \widehat{\boldsymbol{\theta}}_{V_t,t-1} - \boldsymbol{\theta}^{j(i_t)} \right) \right| \leq \beta \|\boldsymbol{x}_a\|_{\overline{\boldsymbol{M}}_{V_t,t-1}^{-1}} + \frac{2L^2 \gamma_s}{\lambda_x} \mathbb{1}\{\gamma_s > \gamma\}.$$

Combining the two cases, we finish the proof of Lemma 9. $\qquad \square$

Now we can prove the regret upper bound of `PhaseUniCLUB`.

**Theorem 3** (Regret of `PhaseUniCLUB`). *Under the stochastic context setting (Assumptions 1, 2, 3), but $\widetilde{\gamma}$ is unknown, the expected regret of algorithm `PhaseUniCLUB` (Algorithm 5) satisfies:*

$$\mathbb{E}[R(T)] = O\left(\frac{ud}{\gamma^2 \lambda_x^2} \log(T) + \left(\frac{ud}{\lambda_x}\right)^{\frac{1}{3}} T^{\frac{2}{3}} \log^{\frac{1}{3}}(T) + d\sqrt{mT} \log(T)\right).$$

*Proof.* For any round $t$ in the UCB subphase of any phase $s$, with probability at least $1 - 4\log_2\left(C_p \frac{T}{\log(u/\delta)} + 1\right)\delta$, we have

$$
\begin{aligned}
R_t &= \boldsymbol{x}_{a_t^*}^\mathsf{T} \boldsymbol{\theta}^{j(i_t)} - \boldsymbol{x}_{a_t}^\mathsf{T} \boldsymbol{\theta}^{j(i_t)} \\
&= \boldsymbol{x}_{a_t^*}^\mathsf{T}\left(\boldsymbol{\theta}^{j(i_t)} - \widehat{\theta}_{V_t,t-1}\right) + \left(\boldsymbol{x}_{a_t^*}^\mathsf{T}\widehat{\theta}_{V_t,t-1} + C_{a_t^*,t}\right) - \left(\boldsymbol{x}_{a_t}^\mathsf{T}\widehat{\theta}_{V_t,t-1} + C_{a_t,t}\right) \\
&\quad + \boldsymbol{x}_{a_t}^\mathsf{T}\left(\boldsymbol{\theta}^{j(i_t)} - \widehat{\theta}_{V_t,t-1}\right) - C_{a_t^*,t} + C_{a_t,t} \\
&\leq C_{a_t^*,t} + \frac{2L^2\gamma_s}{\lambda_x}\mathbb{1}\{\gamma_s > \gamma\} + C_{a_t,t} + \frac{2L^2\gamma_s}{\lambda_x}\mathbb{1}\{\gamma_s > \gamma\} - C_{a_t^*,t} + C_{a_t,t} \\
&= 2C_{a_t,t} + \frac{4L^2\gamma_s}{\lambda_x}\mathbb{1}\{\gamma_s > \gamma\},
\end{aligned}
$$

where the inequality is due to Lemma 9 and the UCB arm selection strategy of our algorithm.

Denote the regret in phase $s$ by $R^{(s)}$. With the same probability, for any phase $s$, we have

$$
\begin{aligned}
R^{(s)} &\leq T^{(s)} + \sum_{t=T^{\text{init}}+\sum_{p=0}^{s-1} 2^{p/2}T^{(p)}+T^{(s)}+1}^{T^{\text{init}}+\sum_{p=0}^{s} 2^{p/2}T^{(p)}} R_t \\
&\leq T^{(s)} + \frac{4L^2\gamma_s}{\lambda_x}\mathbb{1}\{\gamma_s > \gamma\}\cdot(2^{s/2}-1)T^{(s)} + \sum_{t=T^{\text{init}}+\sum_{p=0}^{s-1} 2^{p/2}T^{(p)}+T^{(s)}+1}^{T^{\text{init}}+\sum_{p=0}^{s} 2^{p/2}T^{(p)}} 2C_{a_t,t}.
\end{aligned}
$$

Summing up the regret over all phases, we obtain

$$
\begin{aligned}
R(T) &\leq T^{\text{init}} + \sum_{s=0}^{N_p-1} R^{(s)} \\
&\leq T^{\text{init}} + \sum_{s=0}^{N_p-1}\left(T^{(s)} + \frac{4L^2\gamma_s}{\lambda_x}\mathbb{1}\{\gamma_s > \gamma\}\cdot(2^{s/2}-1)T^{(s)}\right) + \sum_{t=T^{\text{init}}+1}^{T} 2C_{a_t,t} \\
&\leq T^{\text{init}} + \sum_{s=0}^{N_p-1}\left(T^{(s)} + \frac{4L^2\gamma_s}{\lambda_x}\mathbb{1}\{\gamma_s > \gamma\}\cdot(2^{s/2}-1)T^{(s)}\right) + 2\beta\sqrt{2mTd\log\left(1 + \frac{TL^2}{\lambda d + 4L^2}\right)},
\end{aligned}
$$
(32)

where Equation (32) follows Lemma 5 and the proof of Theorem 1.

Now we bound the second term in Equation (32),

$$
\begin{aligned}
\sum_{s=0}^{N_p-1}\left(T^{(s)} + \frac{4L^2\gamma_s}{\lambda_x}\cdot(2^{s/2}-1)T^{(s)}\right) &\leq \sum_{s=0}^{N_p-1}\left(T^{(s)} + \frac{4L^2\gamma_s}{\lambda_x}\mathbb{1}\{\gamma_s > \gamma\}\cdot 2^{s/2}T^{(s)}\right) \\
&= \sum_{s=0}^{N_p-1}\left(1 + \frac{4L^2}{\lambda_x}\mathbb{1}\{\gamma_s > \gamma\}\right)T^{(s)} \\
&\leq \left(\left(\frac{C_p(T-T^{\text{init}})}{\log(\frac{u}{\delta})}+1\right)^{\frac{2}{3}} + \frac{8L^2}{\gamma^2\lambda_x}\right)\frac{2048ud\log(\frac{u}{\delta})}{\lambda_x},
\end{aligned}
$$
(33)

where Equation (33) is due to $\gamma_s = 2^{-\frac{s}{2}}$.

Then, with probability at least $1 - 4\log_2\left(C_p \frac{T}{\log{(u/\delta)}} + 1\right)\delta$, we have

$$R(T) \leq T^{\text{init}} + 2\beta\sqrt{2mTd\log\left(1 + \frac{TL^2}{\lambda d + 4L^2}\right)} + \left(\left(\frac{C_p(T - T^{\text{init}})}{\log(\frac{u}{\delta})} + 1\right)^{\frac{2}{3}} + \frac{8L^2}{\gamma^2\lambda_x}\right)\frac{2048ud\log(\frac{u}{\delta})}{\lambda_x}.$$

Recall that $T^{\text{init}} = 16u\log\left(\frac{u}{\delta}\right) + 4u \cdot \frac{8L^2}{\lambda_x}\log\left(\frac{ud}{\delta}\right)$ and $C_p = \frac{\sqrt{2}\lambda_x}{1024ud}$. Let $\delta = \frac{1}{T}$ and by the law of total expectations, we have

$$\mathbb{E}[R(T)] \leq T^{\text{init}} + 2\beta\sqrt{2mTd\log\left(1 + \frac{TL^2}{\lambda d + 4L^2}\right)} + \left(\left(\frac{C_p(T - T^{\text{init}})}{\log(\frac{u}{\delta})} + 1\right)^{\frac{2}{3}} + \frac{8L^2}{\gamma^2\lambda_x}\right)\frac{2048ud\log(uT)}{\lambda_x}$$

$$+ 4\log_2\left(C_p\frac{T}{\log{(u/\delta)}} + 1\right)\delta \cdot T$$

$$\leq 16u\log(uT) + 4u \cdot \frac{8L^2}{\lambda_x}\log(udT) + 2\beta\sqrt{2mTd\log\left(1 + \frac{TL^2}{\lambda d + 4L^2}\right)}$$

$$+ \left(\left(\frac{C_p(T - T^{\text{init}})}{\log(\frac{u}{\delta})} + 1\right)^{\frac{2}{3}} + \frac{8L^2}{\gamma^2\lambda_x}\right)\frac{2048ud\log(uT)}{\lambda_x} + 4\log_2\left(C_p\frac{T}{\log{(uT)}} + 1\right)$$

$$= \mathcal{O}\left(\frac{ud}{\gamma^2\lambda_x^2}\log(T) + \left(\frac{ud}{\lambda_x}\right)^{\frac{1}{3}}T^{\frac{2}{3}}\log^{\frac{1}{3}}(T) + d\sqrt{mT}\log(T)\right). \qquad \square$$

## E  Theoretical Analysis of SACLUB and SASCLUB

**Lemma 10.** *Under the smoothed adversary setting, SACLUB and SASCLUB have the following lower bound on the expected minimum eigenvalue of $\boldsymbol{x}_{a_t}\boldsymbol{x}_{a_t}^{\mathsf{T}}$:*

$$\lambda_{\min}\left(\mathbb{E}\left[\boldsymbol{x}_{a_t}\boldsymbol{x}_{a_t}^{\mathsf{T}}\right]\right) \geq c_1\frac{\sigma^2}{\log K} \triangleq \widetilde{\lambda}_x,$$

*where $c_1$ is some constant.*

*Proof.* Fix a time $t$. Let $\boldsymbol{Q}$ be a unitary matrix that rotates $\widehat{\boldsymbol{\theta}}_{i_t,t}$ to align it with the x-axis, retaining its magnitude but zeroing out all components other than the first component, i.e., $\boldsymbol{Q}\widehat{\boldsymbol{\theta}}_{i_t,t} = (\|\widehat{\boldsymbol{\theta}}_{i_t,t}\|, 0, 0, \ldots, 0)$. Such $\boldsymbol{Q}$ always exists because it just rotates the space. According to the UCB arm selection strategy, $\boldsymbol{x}_{a_t} = \arg\max_{a \in \mathcal{A}_t}\left(\widehat{\boldsymbol{\theta}}_{i_t,t}^{\mathsf{T}}\boldsymbol{x}_a + C_{a,t}\right)$, where $C_{a,t} = \beta\|\boldsymbol{x}_a\|_{\overline{\boldsymbol{M}}_{V_t,t-1}^{-1}}$. Denote the $i$-th arm in $\mathcal{A}_t$ as $a_{t,i}$, we have

$$\lambda_{\min}\left(\mathbb{E}\left[\boldsymbol{x}_{a_t}\boldsymbol{x}_{a_t}^{\mathsf{T}}\right]\right) = \lambda_{\min}\left(\mathbb{E}\left[\boldsymbol{x}\boldsymbol{x}^{\mathsf{T}} \mid \boldsymbol{x} = \arg\max_{a \in \mathcal{A}_t}\left(\widehat{\boldsymbol{\theta}}_{i_t,t}^{\mathsf{T}}\boldsymbol{x}_a + C_{a,t}\right)\right]\right)$$

$$= \min_{\boldsymbol{w}:\|\boldsymbol{w}\|=1}\boldsymbol{w}^{\mathsf{T}}\mathbb{E}\left[\boldsymbol{x}\boldsymbol{x}^{\mathsf{T}} \mid \boldsymbol{x} = \arg\max_{a \in \mathcal{A}_t}\left(\widehat{\boldsymbol{\theta}}_{i_t,t}^{\mathsf{T}}\boldsymbol{x}_a + C_{a,t}\right)\right]\boldsymbol{w}$$

$$= \min_{\boldsymbol{w}:\|\boldsymbol{w}\|=1}\mathbb{E}\left[(\boldsymbol{w}^{\mathsf{T}}\boldsymbol{x})^2 \mid \boldsymbol{x} = \arg\max_{a \in \mathcal{A}_t}\left(\widehat{\boldsymbol{\theta}}_{i_t,t}^{\mathsf{T}}\boldsymbol{x}_a + C_{a,t}\right)\right]$$

$$\geq \min_{\boldsymbol{w}:\|\boldsymbol{w}\|=1}\text{Var}\left[\boldsymbol{w}^{\mathsf{T}}\boldsymbol{x} \mid \boldsymbol{x} = \arg\max_{a \in \mathcal{A}_t}\left(\widehat{\boldsymbol{\theta}}_{i_t,t}^{\mathsf{T}}\boldsymbol{x}_a + C_{a,t}\right)\right]$$

$$= \min_{\boldsymbol{w}:\|\boldsymbol{w}\|=1}\text{Var}\left[(\boldsymbol{Q}\boldsymbol{w})^{\mathsf{T}}\boldsymbol{Q}\boldsymbol{x} \mid \boldsymbol{x} = \arg\max_{a \in \mathcal{A}_t}\left((\boldsymbol{Q}\widehat{\boldsymbol{\theta}}_{i_t,t})^{\mathsf{T}}\boldsymbol{Q}\boldsymbol{x}_a + C_{a,t}\right)\right] \qquad (34)$$

$$= \min_{\boldsymbol{w}:\|\boldsymbol{w}\|=1}\text{Var}\left[\boldsymbol{w}^{\mathsf{T}}\boldsymbol{Q}\boldsymbol{x} \mid \boldsymbol{x} = \arg\max_{a \in \mathcal{A}_t}\left(\|\widehat{\boldsymbol{\theta}}_{i_t,t}\|(\boldsymbol{Q}\boldsymbol{x}_a)_1 + C_{a,t}\right)\right] \qquad (35)$$

$$= \min_{\boldsymbol{w}:\|\boldsymbol{w}\|=1} \mathrm{Var}\left[\boldsymbol{w}^\mathsf{T}\boldsymbol{Q}\boldsymbol{\varepsilon} \,\Big|\, \boldsymbol{\varepsilon} = \underset{\boldsymbol{\varepsilon}_{t,i}:i\in[K]}{\arg\max}\left((\boldsymbol{Q}\boldsymbol{\mu}_{t,i}+\boldsymbol{Q}\boldsymbol{\varepsilon}_{t,i})_1 + \frac{C_{a_{t,i,t}}}{\|\widehat{\boldsymbol{\theta}}_{i_t,t}\|}\right)\right] \quad (36)$$

$$= \min_{\boldsymbol{w}:\|\boldsymbol{w}\|=1} \mathrm{Var}\left[\boldsymbol{w}^\mathsf{T}\boldsymbol{\varepsilon} \,\Big|\, \boldsymbol{\varepsilon} = \underset{\boldsymbol{\varepsilon}_{t,i}:i\in[K]}{\arg\max}\left((\boldsymbol{Q}\boldsymbol{\mu}_{t,i}+\boldsymbol{\varepsilon}_{t,i})_1 + \frac{C_{a_{t,i,t}}}{\|\widehat{\boldsymbol{\theta}}_{i_t,t}\|}\right)\right] \quad (37)$$

where Equation (34) uses the property of unitary matrices: $\boldsymbol{Q}^\mathsf{T}\boldsymbol{Q} = \boldsymbol{I}$. Equation (35) applies matrix $\boldsymbol{Q}$ so only the first component is non-zero and we use the fact that minimizing over $\boldsymbol{Qw}$ is equivalent to over $\boldsymbol{w}$. Equation (36) follows because each arm $\boldsymbol{x} = \boldsymbol{\mu} + \boldsymbol{\varepsilon}$ and adding a constant a to a random variable does not change its variance. Equation (37) is due to the rotation invariance of symmetrically truncated Gaussian distributions.

Since $\boldsymbol{\varepsilon}_{t,i} \sim \mathcal{N}(0,\sigma^2\boldsymbol{I})$ conditioned on $|(\boldsymbol{\varepsilon}_{t,i})_j| \leq R, \forall j \in [d]$, by the property of (truncated) multivariate Gaussian distributions, the components of $\boldsymbol{\varepsilon}_{t,i}$ can be equivalently regarded as $d$ independent samples from a (truncated) univariate Gaussian distribution, i.e., $(\boldsymbol{\varepsilon}_{t,i})_j \sim \mathcal{N}(0,\sigma^2)$ conditioned on $|(\boldsymbol{\varepsilon}_{t,i})_j| \leq R, \forall j \in [d]$. Therefore, we have

$$\mathrm{Var}\left[\boldsymbol{w}^\mathsf{T}\boldsymbol{\varepsilon}\right] = \mathrm{Var}\left[\sum_{i=1}^d \boldsymbol{w}_i\boldsymbol{\varepsilon}_i\right] = \sum_{i=1}^d \boldsymbol{w}_i^2 \, \mathrm{Var}\left[\boldsymbol{\varepsilon}_i\right],$$

where the exchanging of variance and summation is due to the independence of $\boldsymbol{\varepsilon}_i$. Therefore, let $p_{t,i} = (\boldsymbol{Q}\boldsymbol{\mu}_{t,i})_1 + \frac{C_{a_{t,i,t}}}{\|\widehat{\boldsymbol{\theta}}_{i_t,t}\|}$, we can write

$$\min_{\boldsymbol{w}:\|\boldsymbol{w}\|=1} \mathrm{Var}\left[\boldsymbol{w}^\mathsf{T}\boldsymbol{\varepsilon} \,\Big|\, \boldsymbol{\varepsilon} = \underset{\boldsymbol{\varepsilon}_{t,i}:i\in[K]}{\arg\max}((\boldsymbol{\varepsilon}_{t,i})_1 + p_{t,i})\right]$$

$$= \min_{\boldsymbol{w}:\|\boldsymbol{w}\|=1} \sum_{j=1}^d \boldsymbol{w}_j^2 \, \mathrm{Var}\left[\boldsymbol{\varepsilon}_j \,\Big|\, \boldsymbol{\varepsilon} = \underset{\boldsymbol{\varepsilon}_{t,i}:i\in[K]}{\arg\max}((\boldsymbol{\varepsilon}_{t,i})_1 + p_{t,i})\right]$$

$$= \min_{\boldsymbol{w}:\|\boldsymbol{w}\|=1} \left\{ \boldsymbol{w}_1^2 \, \mathrm{Var}\left[\boldsymbol{\varepsilon}_1 \,\Big|\, \boldsymbol{\varepsilon} = \underset{\boldsymbol{\varepsilon}_{t,i}:i\in[K]}{\arg\max}((\boldsymbol{\varepsilon}_{t,i})_1 + p_{t,i})\right] \right.$$
$$\left. + \sum_{j=2}^d \boldsymbol{w}_j^2 \, \mathrm{Var}\left[\boldsymbol{\varepsilon}_j \,\Big|\, \boldsymbol{\varepsilon} = \underset{\boldsymbol{\varepsilon}_{t,i}:i\in[K]}{\arg\max}((\boldsymbol{\varepsilon}_{t,i})_1 + p_{t,i})\right] \right\}$$

$$= \min_{\boldsymbol{w}:\|\boldsymbol{w}\|=1} \left\{ \boldsymbol{w}_1^2 \, \mathrm{Var}\left[\boldsymbol{\varepsilon}_1 \,\Big|\, \boldsymbol{\varepsilon} = \underset{\boldsymbol{\varepsilon}_{t,i}:i\in[K]}{\arg\max}((\boldsymbol{\varepsilon}_{t,i})_1 + p_{t,i})\right] + \sum_{j=2}^d \boldsymbol{w}_j^2 \, \mathrm{Var}[\boldsymbol{\varepsilon}_j] \right\}$$

$$= \min_{\boldsymbol{w}:\|\boldsymbol{w}\|=1} \left\{ \boldsymbol{w}_1^2 \, \mathrm{Var}\left[\boldsymbol{\varepsilon}_1 \,\Big|\, \boldsymbol{\varepsilon} = \underset{\boldsymbol{\varepsilon}_{t,i}:i\in[K]}{\arg\max}((\boldsymbol{\varepsilon}_{t,i})_1 + p_{t,i})\right] + (1 - \boldsymbol{w}_1^2)\sigma^2 \right\}$$

$$= \min \left\{ \mathrm{Var}\left[\boldsymbol{\varepsilon}_1 \,\Big|\, \boldsymbol{\varepsilon} = \underset{\boldsymbol{\varepsilon}_{t,i}:i\in[K]}{\arg\max}((\boldsymbol{\varepsilon}_{t,i})_1 + p_{t,i})\right], \sigma^2 \right\} \geq c_1 \frac{\sigma^2}{\log K},$$

where in the last inequality, we use Lemma 15 and Lemma 14 in Sivakumar et al. (2020) and get

$$\mathrm{Var}\left[\boldsymbol{\varepsilon}_1 \,\Big|\, \boldsymbol{\varepsilon} = \underset{\boldsymbol{\varepsilon}_{t,i}:i\in[K]}{\arg\max}((\boldsymbol{\varepsilon}_{t,i})_1 + p_{t,i})\right] \geq \mathrm{Var}\left[\boldsymbol{\varepsilon}_1 \,\Big|\, \boldsymbol{\varepsilon} = \underset{\boldsymbol{\varepsilon}_{t,i}:i\in[K]}{\arg\max}(\boldsymbol{\varepsilon}_{t,i})_1\right] \geq c_1 \frac{\sigma^2}{\log K}. \qquad \square$$

**Lemma 11.** *Under the smoothed adversarial setting, for algorithms SACLUB and SASCLUB, with probability at least $1 - \delta$ for any $\delta \in (0,1)$, if $T_{i,t} \geq \frac{8(1+\sqrt{d}R)^2\log(K)}{c_1\sigma^2}\log\left(\frac{ud}{\delta}\right) = \frac{8(1+\sqrt{d}R)^2}{\widetilde{\lambda}_x}\log\left(\frac{ud}{\delta}\right)$ for all users $i \in [u]$, we have*

$$\lambda_{\min}(\boldsymbol{S}_{i,t}) \geq \frac{c_1\sigma^2 T_{i,t}}{2\log K} = \frac{\widetilde{\lambda}_x T_{i,t}}{2}, \forall i \in [u],$$

*where $c_1$ is some constant and $\widetilde{\lambda}_x \triangleq \frac{c_1\sigma^2}{\log K}$.*

*Proof.* The proof follows the same techniques as Lemma 2. The main difference lies in two aspects: (1) in the smoothed adversary setting, the length of feature vectors is bounded by $1 + \sqrt{d}R$ instead of $L$. Therefore, $\lambda_{\max}(\boldsymbol{x}_{a_\tau} \boldsymbol{x}_{a_\tau}^\mathsf{T}) \leq (1 + \sqrt{d}R)^2$. (2) the computation of $\mu_{\min}$ needs to refer to Lemma 10.

Specifically, to compute $\mu_{\min}$, by the super-additivity of the minimum eigenvalue (due to Weyl's inequality) and Lemma 10, we have

$$\mu_{\min} = \lambda_{\min}\left(\sum_{\tau \in [t]: i_\tau = i} \mathbb{E}[\boldsymbol{x}_{a_\tau} \boldsymbol{x}_{a_\tau}^\mathsf{T}]\right) \geq \sum_{\tau \in [t]: i_\tau = i} \lambda_{\min}\left(\mathbb{E}[\boldsymbol{x}_{a_\tau} \boldsymbol{x}_{a_\tau}^\mathsf{T}]\right) \geq T_{i,t} c_1 \frac{\sigma^2}{\log K}.$$

On the other hand, we have

$$\mu_{\max} = \lambda_{\max}\left(\sum_{\tau \in [t]: i_\tau = i} \mathbb{E}[\boldsymbol{x}_{a_\tau} \boldsymbol{x}_{a_\tau}^\mathsf{T}]\right)$$

So by Lemma 12, we have for any $\varepsilon \in (0, 1)$,

$$\Pr\left[\lambda_{\min}(\boldsymbol{S}_{i,t}) \leq (1 - \varepsilon) T_{i,t} c_1 \frac{\sigma^2}{\log K}\right] \leq \Pr\left[\lambda_{\min}(\boldsymbol{S}_{i,t}) \leq (1 - \varepsilon)\mu_{\min}\right]$$

$$\leq d \left[\frac{e^{-\varepsilon}}{(1 - \varepsilon)^{1-\varepsilon}}\right]^{\mu_{\min}/(1+\sqrt{d}R)^2}$$

$$\leq d \left[\frac{e^{-\varepsilon}}{(1 - \varepsilon)^{1-\varepsilon}}\right]^{\frac{T_{i,t} c_1 \sigma^2}{\log(K)(1+\sqrt{d}R)^2}},$$

where the last inequality is because $e^{-x}$ is decreasing. Then the proof follows by the same derivation as Lemma 2, except that $\lambda_x$ is replaced by $\widetilde{\lambda}_x \triangleq \frac{c_1 \sigma^2}{\log K}$. $\qquad \square$

**Theorem 4** (Regret of SACLUB and SASCLUB). *Under the smoothed adversarial context setting (Assumptions 1, 4, 5), the expected regrets of SACLUB and SASCLUB both satisfy:*

$$\mathbb{E}[R(T)] = O\left(\frac{ud}{\gamma^2 \widetilde{\lambda}_x} \log(T) + d\sqrt{mT} \log(T)\right),$$

*where $\widetilde{\lambda}_x = c_1 \frac{\sigma^2}{\log K}$ for some constant $c_1$.*

*Proof.* For algorithm SACLUB, similar to the proof of Theorem 1 under the stochastic context setting, we need to derive the counterparts of Lemma 3, Lemma 4, and Lemma 5 under the smoothed adversary setting according to Lemma 10 and Lemma 11. The proofs use almost the same techniques with $\lambda_x$ replaced by $\widetilde{\lambda}_x$. Similarly, the proof for algorithm SASCLUB requires the counterparts of Lemmas used in the proof of Theorem 2, thus we skip the details. $\qquad \square$

## F   TECHNICAL INEQUALITIES

We present the technical inequalities used throughout the proofs. For inequalities from existing literature, we provide detailed references for readers' convenience.

**Lemma 12** (Matrix Chernoff, Corollary 5.2 in Tropp (2011)). *Consider a finite sequence $\{\boldsymbol{X}_k\}$ of independent, random, self-adjoint matrices with dimension $d$. Assume that each random matrix satisfies*

$$\boldsymbol{X}_k \succeq \boldsymbol{0} \quad and \quad \lambda_{max}(\boldsymbol{X}_k) \leq R \quad almost\ surely.$$

*Define*

$$\boldsymbol{Y} := \sum_k \boldsymbol{X}_k \quad and \quad \mu_{min} := \lambda_{\min}(\mathbb{E}[\boldsymbol{Y}]) = \lambda_{\min}\left(\sum_k \mathbb{E}[\boldsymbol{X}_k]\right).$$

*Then, for any $\delta \in (0, 1)$,*

$$\Pr\left[\lambda_{\min}\left(\sum_k \boldsymbol{X}_k\right) \leq (1 - \delta)\mu_{min}\right] \leq d \left[\frac{e^{-\delta}}{(1 - \delta)^{1-\delta}}\right]^{\mu_{min}/R}.$$

**Lemma 13** (Lemma 8 in Li & Zhang (2018)). *Let $x_1, x_2, \ldots, x_n$ be independent Bernoulli random variables with mean $0 < p \leq \frac{1}{2}$. Let $\delta > 0, B > 0$, then with probability at least $1 - \delta$,*

$$\sum_{s=1}^{t} x_s \geq B, \quad \forall t \geq \frac{16}{p} \log\left(\frac{1}{\delta}\right) + \frac{4B}{p}.$$

**Lemma 14.** *For $a > 0, b > 0, ab \geq 1$, if $t \geq 2a \log(ab)$, then*

$$t \geq a \log(1 + bt).$$

*Proof.* Since $t$ increases faster than $a \log(1 + bt)$, it suffices to prove $t \geq a \log(1 + bt)$ for $t = 2a \log(ab)$. Equivalently, we only need to show:

$$2a \log(ab) \geq a \log(1 + 2ab \log(ab)) = a \log(ab) + a \log\left(2 \log\left(e^{-2ab} ab\right)\right),$$

which follows by observing that $ab \geq 2 \log(e^{-2ab} ab) = \frac{1}{ab} + 2 \log(ab)$ holds when $ab \geq 1$. $\square$

**Lemma 15.** *Let $X_1, X_2, \ldots, X_n$ be a sequence of $n$ independent and identically distributed (i.i.d.) random variables, each following a distribution $P$. Let $Y$ be a random variable that is uniformly selected from the set $\{X_1, X_2, \ldots, X_n\}$, then $Y$ follows the same distribution $P$.*

*Proof.* It suffices to show $\Pr[Y \in A] = \Pr[X_1 \in A]$ for any measurable set $A$. By the law of total probability, we can express the probability that $Y$ falls into the set $A$ as:

$$\Pr[Y \in A] = \sum_{i=1}^{n} \Pr[Y \in A \mid Y = X_i] \Pr[Y = X_i]$$

$$= \sum_{i=1}^{n} \Pr[X_i \in A] \Pr[Y = X_i]$$

$$= \Pr[X_1 \in A],$$

where we use the fact that $X_i$ are i.i.d. and $\Pr[Y = X_i] = 1/n$. $\square$

**Lemma 16** (Determinant-trace inequality, Lemma 10 in Abbasi-Yadkori et al. (2011)). *Suppose $\boldsymbol{X}_1, \boldsymbol{X}_2, \ldots, \boldsymbol{X}_t \in \mathbb{R}^d$ and for any $1 \leq s \leq t, \|\boldsymbol{X}_s\|_2 \leq L$. Let $\overline{\boldsymbol{V}}_t = \lambda \boldsymbol{I} + \sum_{s=1}^{t} \boldsymbol{X}_s \boldsymbol{X}_s^{\mathsf{T}}$ for some $\lambda > 0$. Then,*

$$\det(\overline{\boldsymbol{V}}_t) \leq \left(\lambda + \frac{tL^2}{d}\right)^d.$$

# G MORE EXPERIMENTS

## G.1 EVALUATION UNDER THE SMOOTHED ADVERSARIAL CONTEXT SETTING

To implement the smoothed adversarial contexts, for each arm in the synthetic dataset and three real-world recommendation system datasets, we add a Gaussian noise vector sampled from $\mathcal{N}(0, \boldsymbol{I}/10)$. As illustrated in Figure 2, for all the datasets, SACLUB and SASCLUB (which are essentially adapted versions of CLUB and SCLUB) outperform LinUCB-Ind and LINUCB-One. Additionally, under the smoothed adversarial setting, LinUCB-Ind and LINUCB-One exhibit more significant fluctuations compared to SACLUB and SASCLUB, highlighting the increased complexity of the smoothed adversarial setting relative to the stochastic setting. This corroborates our theoretical results, showing that with some minor changes, the existing algorithms CLUB and SCLUB can be extended to the more practical smoothed adversarial setting, which is closer to the original setting of contextual linear bandits (Abbasi-Yadkori et al., 2011).

## G.2 CUMULATIVE REGRET WITH DIFFERENT ARM SET SIZES

Under the stochastic context setting, we evaluate the impact of the arm set size $|\mathcal{A}_t| = K$ by adjusting $K = 80, 100, 120, 140$ using the Yelp dataset, since it has the largest number of users. As demonstrated in Figure 3, there is no substantial amplification of the cumulative regrets as the arm set size $K$ increases. This observation validates our theoretical results (Theorems 1 and 2), where the regret upper bounds of UniCLUB and UniSCLUB do not involve $K$.

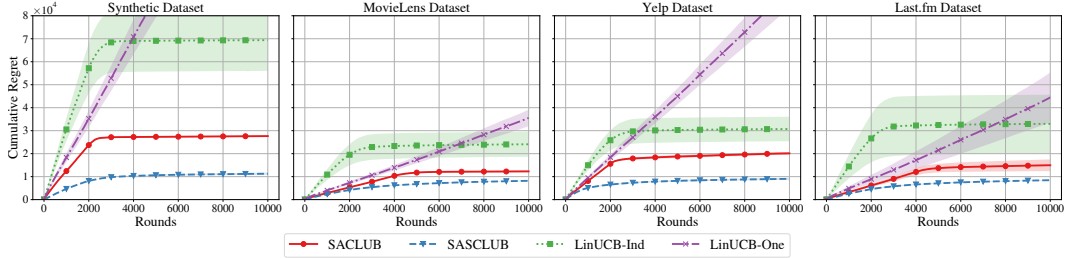

Figure 2: Comparison of cumulative regrets under the smoothed adversarial context setting.

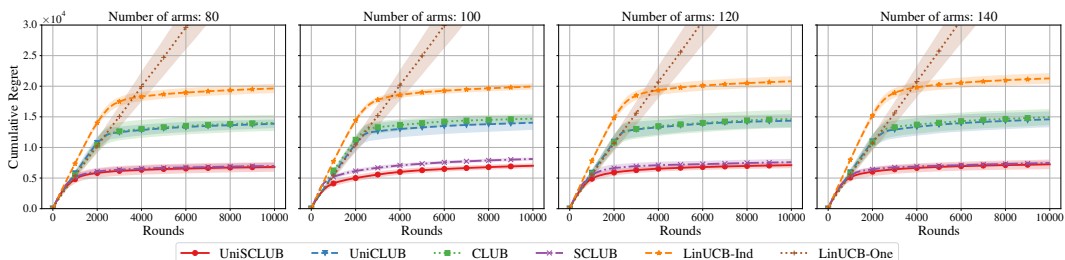

Figure 3: Comparison of cumulative regrets with different arm set sizes.

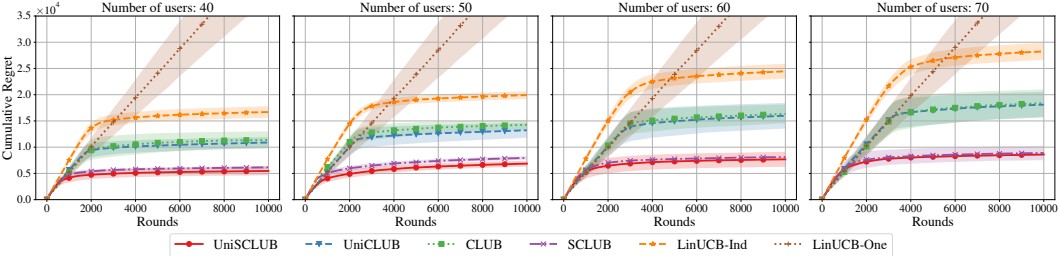

Figure 4: Comparison of cumulative regrets with different numbers of users.

### G.3 CUMULATIVE REGRET WITH DIFFERENT USER NUMBERS

To examine the effect of the number of users, we adjust the number of users $u = 40, 50, 60, 70$ while keeping the number of clusters $m = 10$ using the Yelp dataset. As shown in Figure 4, our proposed algorithms exhibit significant advantages compared to the baselines. Additionally, as the number of users increases, the cumulative regrets also increase. This is expected because a larger number of users poses a greater challenge in learning their preference vectors and identifying the cluster structures.

### G.4 COMPARISON WITH NON-CLUSTERING-BASED BASELINES

In this subsection, we compare our algorithms with two additional graph-based baseline algorithms, GOB.Lin (Cesa-Bianchi et al., 2013) and GraphUCB (Yang et al., 2020), within the stochastic context setting. While both GOB.Lin and GraphUCB leverage user similarities to enhance preference estimation, their methodologies fundamentally differ from ours (and all the other clustering-based algorithms) in two aspects: (1) Neither GOB.Lin nor GraphUCB assume the existence of user clusters or explicitly perform clustering, and (2) both algorithms assume prior knowledge of a user relationship graph. Additionally, it is worth noting that both GOB.Lin and GraphUCB have significantly higher computational complexity compared to clustering-based algorithms, as they require operations such as multiplication and inversion of high-dimensional matrices of size $\mathbb{R}^{ud \times ud}$ for item recommendation, where $u$ is the number of users and $d$ is the feature dimension. In contrast, clustering-based algorithms (including ours) only involve matrix manipulations of size $\mathbb{R}^{d \times d}$.

For the experiments, due to the high computational overhead of GOB.Lin and GraphUCB, we randomly select 20 users (instead of 50 as in other experiments) and divide them into 4 clusters. At each round $t$, a user $i_t$ is uniformly drawn from the 20 users, and 20 items are randomly sampled from the full set of arms to form the arm set $\mathcal{A}_t$. For GOB.Lin and GraphUCB, the user graph is constructed by connecting users within the same cluster.

The cumulative regret results are shown in Figure 5. Both LinUCB-One and LinUCB-Ind exhibit significantly higher cumulative regret than the other algorithms that incorporate user similarity. Among all the clustering-based algorithms, our algorithms `UniCLUB` and `UniSCLUB` consistently outperform their respective counterparts, CLUB and SCLUB, across all four datasets, demonstrating the effectiveness of our proposed uniform exploration strategy. When compared to the new baselines, `UniSCLUB` achieves superior performance compared to GOB.Lin and GraphUCB across all four datasets, and `UniCLUB` also outperforms GOB.Lin. Although GraphUCB slightly surpasses `UniCLUB` on the Last.fm dataset and achieves competitive results on the synthetic dataset, it incurs higher regret on the Movielens and Yelp datasets compared to `UniCLUB`. These results align with expectations, as our setting assumes that users are clustered, which allows clustering-based methods to explicitly exploit this structure for improved performance.

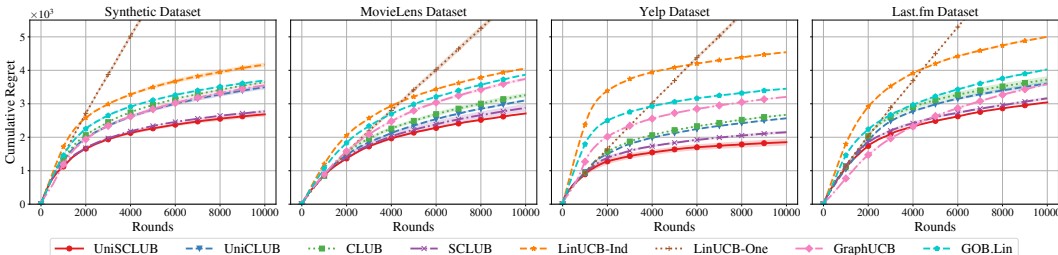

Figure 5: Comparison of cumulative regrets with more baselines.

To demonstrate computational complexity, we also measure the average running time of all algorithms across datasets, with the results summarized in Table 2. As shown in the table, GOB.Lin and GraphUCB exhibit significantly higher average running times, exceeding those of other baselines by more than an order of magnitude. In contrast, our algorithms, `UniCLUB` and `UniSCLUB`, achieve computational efficiency comparable to that of the other baseline methods.

Table 2: Comparison of average running time (in seconds).

| Time(s) \ Algorithms Datasets | LinUCB-One | LinUCB-Ind | CLUB | SCLUB | UniCLUB | UniSCLUB | GraphUCB | GOB.Lin |
|---|---|---|---|---|---|---|---|---|
| **Synthetic** | 2.79 | 2.61 | 3.02 | 7.70 | 3.02 | 8.28 | 128.32 | 204.55 |
| **MovieLens** | 9.78 | 9.62 | 10.11 | 16.96 | 9.96 | 16.80 | 117.25 | 214.96 |
| **Yelp** | 9.81 | 9.59 | 10.17 | 15.05 | 10.19 | 12.13 | 119.86 | 216.35 |
| **Last.fm** | 4.66 | 1.44 | 5.00 | 12.18 | 5.00 | 12.42 | 106.04 | 231.25 |

