# OpenReview forum: "Demystifying Online Clustering of Bandits: Enhanced Exploration Under Stochastic and Smoothed Adversarial Contexts"
_ICLR.cc/2025/Conference — ICLR 2025 Poster_

### Official Review · Reviewer_wksx · 2024-10-21

**Soundness:** 4
**Presentation:** 4
**Contribution:** 3
**Rating:** 8
**Confidence:** 4

**Summary:**

The paper provides novel algorithms to solve the online clustering of bandits problem. Their setting relies on a slightly different set of assumptions w.r.t. the state-of-the-art. They provide theoretical analysis of their algorithms and also provide a proper experimental analysis to show the performance of their approach. Their approach shows little improvement over the performance of state-of-the-art algorithms in the experiments.

**Strengths:**

The writing and presentation of the paper is done with a high quality. The paper highlights an interesting theoretical change w.r.t. the state-of-the-art. It can be even interesting from a practical point of view for real-world applications in large scale.

**Weaknesses:**

I am not sure if it is a weakness of the approach, but it seems [from the experiments] that the presented algorithms do not improve upon state-of-the-art significantly. In addition, I believe the authors could test the performance of their algorithms against more benchmarks such as Gob.Lin [Cesa-Bianchi et al., 2013] and GraphUCB [Yang et al., 2020], but it is not done.

**Questions:**

Why didn't you test the performance of your algorithms for the synthetic data experiment against benchmarks such as  Gob.Lin [Cesa-Bianchi et al., 2013] and GraphUCB [Yang et al., 2020]?

---

> ### Author Response · Authors · 2024-11-18
>
> Thank you for your insightful comments and for acknowledging the quality of our work! Here are our responses:
>
> ---
> **Q1: From the experiments, the presented algorithms do not improve upon state-of-the-art significantly.**\
> **A1:** As discussed in Lines 502-504, the modest improvement observed in our experiments is expected and aligns with our theoretical results. Specifically, while our work achieves improved regret bounds (Theorems 1, 2, and 4), these improvements are logarithmic and relatively minor compared to the dominant term in the regret expression. Consequently, such theoretical gains may not always yield a significant practical difference.
>
> ---
> **Q2: Why didn't you test the performance of your algorithms for the synthetic data experiment against benchmarks such as Gob.Lin [Cesa-Bianchi et al., 2013] and GraphUCB [Yang et al., 2020]?**\
> **A2:** Thank you for your insightful suggestion. Our work primarily aims to relax the strong assumptions introduced in Gentile et al. (2014), which are not used in Gob.Lin and GraphUCB. Additionally, unlike Gob.Lin and GraphUCB, our setting does not require a known user relationship graph. Therefore we did not initially compare our algorithms against these benchmarks.
>
> However, we agree that Gob.Lin and GraphUCB are relevant to our work and it would be valuable to include such comparisons. We will add a discussion in the manuscript to clarify the distinctions between our setting and those of Gob.Lin and GraphUCB. Moreover, we are currently running experiments to compare our algorithms with these benchmarks, and **we will strive to include these results in a revised version of the manuscript** before the discussion phase ends.
>
> ---
> We appreciate your valuable review of our paper and are more than happy to answer any further questions.

---

> ### Author Response · Authors · 2024-11-25
>
> Dear Reviewer wksx,
>
> As the discussion phase is drawing to a close, we kindly ask whether our responses have resolved your concerns or if there are any remaining issues that we can further clarify. Your insights are invaluable in refining our work, and we are eager to ensure that all your concerns are fully addressed.
>
> Thank you once again for your time and effort in reviewing our manuscript.
>
> Best regards,\
> Authors

---

### Official Review · Reviewer_BvdA · 2024-11-03

**Soundness:** 2
**Presentation:** 4
**Contribution:** 2
**Rating:** 5
**Confidence:** 4

**Summary:**

This paper explores the clustering of contextual bandits in both stochastic and adversarial context settings, introducing a new set of assumptions that improve upon some unrealistic assumptions in existing work. The two proposed algorithms are modifications of existing ones, incorporating pure exploration periods. Regret bounds for these algorithms for two setups are also provided.

**Strengths:**

This paper is presented well so that uninitiated readers can understand the work. Also, this work removes the nonsense assumptions in the existing literature and suggests a new set of assumptions for better analysis.

**Weaknesses:**

The statement in L54-61 should be revised as it can be misleading. This is because the iid and minimum eigenvalue assumptions are still chosen in this work. (The assumptions for adversarial setup should be separately stated.) In addition, this work adopts the bounded context ||X||\leq L, which is stronger than the subGaussian assumption for contexts. Lastly, this work has additional assumptions about the parameters, which\|\theta_i - \theta_j\|>gamma. Even though the reviewer acknowledges that the subGaussian assumption (\sigma^2 < lambda_x/8log 4K) in the existing literature does not make sense, the reviewer believes that more substantive contributions beyond introducing new assumptions with an initial pure exploration period are necessary to be sufficiently impactful.

L198-207: the indices should be corrected.

As accurate clustering is a crucial topic in this work, the authors should state the algorithm for clustering, rather than simply referring to it.

**Questions:**

This work provides slightly looser regret bounds for both algorithms. The reviewer conjectures that the regret bounds can be tighter by revising the existing proof techniques.

Logarithmic regrets are known to be attainable for standard contextual bandits under some assumptions. Do the authors think that the regret bounds can be improved to logarithmic ones under some specific assumptions? If not, could we have a square root lower bound?

It is possible that UCB-based algorithms cannot have logarithmic ones. Is it possible for other types of algorithms?

If logarithmic regret bounds are not attainable, could the authors please explain what differences between standard contextual bandits and this work make it infeasible?

L219: Could the authors please clarify with respect to which variables the expectation is taken?

For typical adversarial setups, as contexts are not random, they cannot have expectation and variance. Could you explain more specifically about the adversarial setup that the authors suggest?

---

> ### Author Response · Authors · 2024-11-18
>
> Thank you for your insightful comments! Here are our responses:
>
> ---
> **Q1: The statement in L54-61 should be revised as it can be misleading. This is because the iid and minimum eigenvalue assumptions are still chosen in this work. (The assumptions for adversarial setup should be separately stated.)**\
> **A1:** Thank you for the suggestion. You are correct that the iid and minimum eigenvalue assumptions are retained in the stochastic context setting (Section 3), but they are removed in the smoothed adversarial context setting (Section 4). The statement in L54-61 highlights the contribution of both settings compared to existing work.
>
> We will revise the manuscript to make this point clearer and to prevent any potential misinterpretation.
>
> ---
> **Q2: This work adopts the bounded context $\\|\boldsymbol{X}\\|\leq L$, which is stronger than the subGaussian assumption for contexts.**\
> **A2:** We would like to clarify that this is not correct. It appears there might be a misunderstanding regarding the sub-Gaussian assumptions used in prior work. In fact, prior work assumes that for any unit vector $\boldsymbol{z} \in \mathbb{R}^d$, the random variable $(\boldsymbol{z}^{\top}\boldsymbol{X})^2$ is sub-Gaussian with variance $\sigma^2 \leq \frac{\lambda_x^2}{8\log(4K)}$ (see Lines 279-284 for details). Note that it does not mean that $\boldsymbol{X}$ itself is sub-Gaussian. Clearly, the bounded context assumption $\\|\boldsymbol{X}\\|\leq L$ does not imply the above sub-Gaussian assumptions.
>
> Additionally, the bounded context assumption is standard in the contextual bandit literature, e.g., Abbasi-Yadkori et al. (2011).
>
> ---
> **Q3: L198-207: the indices should be corrected.**\
> **A3:** We have reviewed Lines 198-207 but could not identify any issues with the indices. Could you please provide specific details of the incorrect indices you noticed?
>
> ---
> **Q4: As accurate clustering is a crucial topic in this work, the authors should state the algorithm for clustering, rather than simply referring to it.**\
> **A4:** We believe there might be a misunderstanding. **Our paper does include the full description of our algorithms**, including the clustering procedures, in both the main text and the appendix. For example, in Algorithm 1, the clustering process starts with a complete graph where each node represents a user. At each time step, the algorithm updates this graph by deleting edges based on gathered information (see Lines 346-350). Eventually, the connected components of the resulting graph correspond to the identified clusters.
>
> ---
> **Q5: This work provides slightly looser regret bounds for both algorithms.**\
> **A5:** We believe there might be a misunderstanding regarding the regret bounds of our algorithms. As shown in Table 1 (Lines 119-132), our proposed algorithms for both the stochastic context setting (UniCLUB, UniSCLUB) and the smoothed adversarial context setting (SACLUB, SASCLUB), actually achieve **tighter regret bounds** compared to previous studies.
>
> The only exception is Phase-UniCLUB. However, as discussed in Remark 6 (L432-440), Phase-UniCLUB is designed to work without prior knowledge of the parameter $\widetilde{\gamma}$. The deteriorated regret is an expected trade-off for this generality, since the algorithm must handle potential misclusterings.
>
> ---
> **Q6: Logarithmic regrets are known to be attainable for standard contextual bandits under some assumptions. Do the authors think that the regret bounds can be improved to logarithmic ones under some specific assumptions? If not, could we have a square root lower bound?**\
> **A6:** We suppose you might be referring to the work by Ghosh et al. (2022), titled *"Breaking the $\sqrt{T}$ Barrier: Instance-Independent Logarithmic Regret in Stochastic Contextual Linear Bandits"*. This paper achieves logarithmic regret under assumptions that are **identical** to those used in prior studies on clustering of bandits, such as Gentile et al. (2014), which we aim to relax in our work.
>
> To some extent, the logarithmic regret explains how strong the assumptions are: they yield results that are too good to be true. In fact, it has been proven that the assumptions used in Ghosh et al. (2022) **do not hold** when the number of arms $K \geq 8$, a size that most applications can easily exceed.
>
> Regarding your questions, if we were to adopt the same assumptions as Ghosh et al. (2022), we could also achieve logarithmic regret. However, our objective is to eliminate such restrictive assumptions to develop more practical algorithms. As for the lower bound, we are sorry that we cannot provide a definitive answer, since the lower bound for online clustering of bandit problem remains an open challenge in the field.

---

> ### Author Response · Authors · 2024-11-18
>
> **Q7: L219: Could the authors please clarify with respect to which variables the expectation is taken?**\
> **A7:** The expectation is taken over the randomness of both the arm selected by the learner ($a_t$) and the user selected by the environment ($i_t$). We will clarify this point in the revised manuscript.
>
> ---
> **Q8: For typical adversarial setups, as contexts are not random, they cannot have expectation and variance.**\
> **A8:** We would like to clarify that our *smoothed adversarial* setting is different from the *fully adversarial* setting you mentioned. In our setting, as defined in Assumption 5 (Lines 388-392), the feature vectors (context) are first selected by an adversary deterministically, and then perturbed by additive Gaussian noise before being presented to the learner. **The randomness introduced by the noise allows the contexts to have well-defined expectation and variance**.
>
> Developing online clustering of bandits algorithms in a fully adversarial context without any randomness remains an open problem.
>
> ---
> We appreciate your valuable review of our paper and are more than happy to answer any further questions.

---

> ### Author Response · Authors · 2024-11-25
>
> Dear Reviewer BvdA,
>
> As the discussion phase is drawing to a close, we kindly ask whether our responses have resolved your concerns or if there are any remaining issues that we can further clarify. Your insights are invaluable in refining our work, and we are eager to ensure that all your concerns are fully addressed.
>
> Thank you once again for your time and effort in reviewing our manuscript.
>
> Best regards,\
> Authors

---

> ### Author Response · Authors · 2024-12-02
>
> Dear Reviewer BvdA,
>
> As today is the final day of the discussion phase, we wanted to kindly follow up on our rebuttal.
>
> In our rebuttal, we clarified the following points:
> 1. The bounded context assumption is not stronger than the subGaussian assumption used in prior works.
> 2. The description of our algorithms is complete and includes the clustering procedures.
> 3. Our work provides tighter regret bounds for both settings.
>
> Additionally, we appreciate your reference to the work on logarithmic regret in contextual linear bandits (Ghosh et al., 2022), which further supports our point that the assumptions in prior works are extremely strong.
>
> We would greatly appreciate it if you could let us know whether our response resolves your concerns.
>
> Best regards,\
> Authors

---

### Official Review · Reviewer_KU8f · 2024-11-09

**Soundness:** 3
**Presentation:** 3
**Contribution:** 2
**Rating:** 5
**Confidence:** 3

**Summary:**

This paper studies online clustering of bandits, with the goal of relaxing the strong context regularity condition adopted in prior works like Gentile, et al. (2014).
For this purpose, the authors first added a uniform exploration phase to the existing algorithms. With the additional knowledge about the gap parameter $\gamma$, an appropriate value for the uniform exploration length $T_{0}$ can be chosen to ensure accurate cluster estimation.
On a parallel direction, the authors adopted the perturbed adversary assumption as Kannan, et al. (2018), and showed that CLUB and SCLUB algorithms by Gentile, et al. (2014) and Li, et al. (2019) now incurs less regret due to failed cluster detection (the first term in regret upper bound).

**Strengths:**

Relaxation of the strong context regularity assumption adopted in online clustering bandits is a well-motivated and important problem.

The paper is technically sound and easy to follow.

**Weaknesses:**

1. My primary concern with this paper is its contribution, as it may overstate the extent to which it relaxes the assumptions used in prior works. I wouldn’t consider adding the gap parameter as a relaxation. With this parameter as input, we can incorporate a dedicated uniform exploration phase of sufficient duration (determined by the gap parameter) to ensure accurate cluster estimation. Gentile et al. (2014) required the additional assumption on variance precisely because they did not have access to such information. Without knowledge of the gap, it is impossible to determine the adequate amount of uniform exploration, necessitating a stronger assumption to ensure that the minimum eigenvalue of the design matrix grows rapidly enough, even under UCB exploration.

2. The use of the perturbed adversary assumption in the context of online clustering of bandits appears to be novel, and I believe it represents a meaningful contribution of this paper. However, the discussion on the technical innovation here is limited, and I would appreciate more clarity on this aspect from the authors.
For instance, since a key part of the proof involves demonstrating that the minimum eigenvalue of the design matrix grows sufficiently under the perturbed adversary assumption (as shown in Lemma 11), how does this approach differ technically from Lemma 3.2 of Kannan et al. (2018)? Could the authors elaborate on the primary differences in the proof structure when applied to online clustering of bandit algorithms as opposed to greedy algorithms?

**Questions:**

Please see my questions above.

Additionally, could the authors provide more details on the application of the self-normalized bound from Abbasi-Yadkori et al. (2011) to the first term in Equation 1? Specifically, I would appreciate a clearer explanation of how the filtration is defined in this context, given that the summation is taken over a set of time steps corresponding to the cluster, which is itself random (cluster estimation is based on observed noisy feedback).

---

> ### Author Response · Authors · 2024-11-18
>
> Thank you for your insightful comments! Here are our responses:
>
> ---
> **Q1: Does adding the gap parameter relax the assumptions used in prior works?**\
> **A1:** Yes, introducing the gap parameter $\widetilde{\gamma}$ relaxes the assumptions used in prior works. Specifically, prior studies rely on extremely strong assumptions about the arm generation distribution $\boldsymbol{X}$ (as detailed in Lines 279-284 of our paper). In fact, it has been proven that **such a distribution does not exist** when the number of arms $K$ is larger than, say, 8 -- an easily attainable size for most applications.
>
> Your explanation of why Gentile et al. (2014) requires these strong assumptions is absolutely correct. Specifically, Gentile et al. (2014) aimed to demonstrate that the UCB algorithm could simultaneously perform clustering inference and regret minimization, even without any prior knowledge about the underlying clusters. However, as discussed in Remark 1 (Lines 230-238), we argue that the goal set by Gentile et al. (2014) is unrealistic, as any clustering algorithm requires some prior knowledge to determine an appropriate stopping criterion (for example, the K-means algorithm needs the number of clusters to be specified). Also, in real-world scenarios, it is highly unlikely that users within the same cluster would have **exactly identical** feature vectors, as assumed by Gentile et al. (2014). Therefore, in practice, a known minimum gap parameter $\widetilde{\gamma}$ is necessary to determine whether two users belong to the same cluster.
>
> In summary, by replacing these strong (or even **unattainable**) assumptions with the gap parameter, we offer a more realistic and practical trade-off.
>
> ---
> **Q2: How does Lemma 11 differ technically from Lemma 3.2 of Kannan et al. (2018)?**\
> **A2:** We think you may want to compare our Lemma 10 with Lemma 3.2 of Kannan et al. (2018), where both lemmas establish lower bounds on the smallest eigenvalue related to $\mathbb{E}[\boldsymbol{x}\_{i}\boldsymbol{x}\_{i}^\top]$ (where $i$ is the selected arm). While our Lemma 10 follows the same idea as Kannan et al. (2018), there are specific differences in their proof approaches: (In the following, we omit the subscript $t$ for simplicity. Each feature vector $\boldsymbol{x}$ is composed as $\boldsymbol{x}=\boldsymbol{\mu} + \boldsymbol{\varepsilon}$, where $\boldsymbol{\mu}$ is chosen by an adversary and $\boldsymbol{\varepsilon}$ is a Gaussian perturbation.)
> 1. **Different Conditioning in the Expectation:** In our Lemma 10, we establish a lower bound on $\lambda\_{\text{min}}(\mathbb{E}[\boldsymbol{x}\_{i}\boldsymbol{x}\_{i}^\top])$ without additional conditions, while in Lemma 3.2 of Kannan et al. (2018), the expectation is conditioned on the event $\max_{j \neq i} \boldsymbol{x}_j^\top\widehat{\boldsymbol{\theta}} \leq \boldsymbol{\mu_i}^\top \widehat{\boldsymbol{\theta}} + r\\|\widehat{\boldsymbol{\theta}}\\|$ for some constant $r$, which intuitively means that the perturbation term for arm $i$ does not need to be too large in order for arm $i$ to be selected by the greedy strategy.
> 2. **Different Analysis:** In Lemma 3.2 of Kannan et al, since the conditioning depends on all arms, their analysis splits the expectation into two parts: one for arm $i$ and the other for all the other arms. Then they reduce the proof to a diversity condition on arm $i$ (Lemma 3.7 of Kannan et al. (2018)). The proof of our Lemma 10 does not require these steps, as it does not condition on any event.
>
> Next, we compare our Lemma 10 with Lemma 3.7 of Kannan et al. (2018). The primary differences are:
> 1. Our Lemma 10 is for the UCB strategy. It lower bounds $\lambda\_{\text{min}}(\mathbb{E}[\boldsymbol{x}\_{i}\boldsymbol{x}\_{i}^\top])$ where $i$ is the selected arm by the UCB strategy: $i=\text{argmax}\_{a \in \mathcal{A}_t}(\boldsymbol{x}_a^\top\widehat{\boldsymbol{\theta}} + C_a)$, where $C_a$ is the confidence term.
> 2. Lemma 3.7 of Kannan et al. (2018) bounds $\lambda_{\text{min}}(\mathbb{E}[\boldsymbol{x}\boldsymbol{x}^\top \mid \widehat{\boldsymbol{\theta}}^\top \boldsymbol{\varepsilon}\geq b])$ for some $b$, for any feature vector $\boldsymbol{x}$ (not necessarily associated with the selected arm).
>
> In summary, while our analysis follows the idea of Kannan et al. (2018), it includes significant modifications to accommodate our setting.

---

> ### Author Response · Authors · 2024-11-18
>
> **Q3: What are the primary differences in the proof structure when applied to online clustering of bandit algorithms as opposed to greedy algorithms?**\
> **A3:** By leveraging the fact that at each time $t$, the vector $\boldsymbol{\mu}_a$ chosen by an adversary and the confidence term $C_a$ are deterministic given the history up to time $t$, we can adapt the analysis used for greedy algorithms to the UCB strategy in our online clustering of bandit algorithms.
>
> Specifically, the greedy strategy selects arms by: $\text{argmax}\_{a \in \mathcal{A}_t} \boldsymbol{x}_a^\top \widehat{\boldsymbol{\theta}} = \text{argmax}\_{a \in \mathcal{A}_t} (\boldsymbol{\mu}_a^\top \widehat{\boldsymbol{\theta}} + \boldsymbol{\varepsilon}_a^\top \widehat{\boldsymbol{\theta}})$, where $\boldsymbol{x}_a = \boldsymbol{\mu}_a + \boldsymbol{\varepsilon}_a$, with $\boldsymbol{\mu}_a$ being deterministic and $\boldsymbol{\varepsilon}_a$ representing random perturbations.
>
> To analyze this, existing studies (e.g., Kannan et al. (2018), Sivakumar et al. (2020)) convert the expectation $\mathbb{E}[\boldsymbol{x}\_{a_t}\boldsymbol{x}\_{a_t}^\top]$ to a variance expression. They then perform transformations to ensure that the randomness lies only in the second term $\boldsymbol{\varepsilon}_a^\top \widehat{\boldsymbol{\theta}}$. This allows them to ignore the deterministic first term and analyze the second term using properties of the Gaussian distribution.
>
> For the UCB strategy used in our algorithms, it selects arms by $\text{argmax}\_{a \in \mathcal{A}_t} (\boldsymbol{x}_a^\top \widehat{\boldsymbol{\theta}} + C_a)=\text{argmax}\_{a \in \mathcal{A}_t} ((\boldsymbol{\mu}_a^\top \widehat{\boldsymbol{\theta}} + C_a) + \boldsymbol{\varepsilon}_a^\top \widehat{\boldsymbol{\theta}}).$ We similarly transform this formula by considering $(\boldsymbol{\mu}_a^\top \widehat{\boldsymbol{\theta}} + C_a)$ as a single term, ensuring that the randomness resides only in the second term $\boldsymbol{\varepsilon}_a^\top \widehat{\boldsymbol{\theta}}$. By doing so, we can follow the same analytical approach as in the greedy algorithms.
>
> ---
> **Q4: Could the authors provide more details on the application of the self-normalized bound from Abbasi-Yadkori et al. (2011) to the first term in Equation 1? How is the filtration defined in this context, given that the summation is taken over a set of time steps corresponding to the cluster, which is itself random.**\
> **A4:** Thank you for your insightful question. The first term in Equation 1 is:
> $$\sum\_{\tau \in [t]: i\_{\tau}=i} \boldsymbol{x}\_{a\_{\tau}} \eta\_{\tau},$$
> where $a\_{\tau}$ is the selected arm at time $\tau$ with associated feature vector $\boldsymbol{x}\_{a\_{\tau}}$, $\eta\_{\tau}$ is the noise term at time $\tau$, and $i\_{\tau}$ is the user at time $\tau$.
>
> First, we would like to clarify that the summation is taken over time steps where user $i$ appears (i.e., when $i_{\tau}=i$) and is **unrelated to the estimated clusters**. In fact, Equation 1 is used to bound the estimation error of the feature vector for each **individual user $i$**, independent of any estimated clusters.
>
> For the filtration definition, we can follow Abbasi-Yadkori et al. (2011) and define it as
> $$\mathcal{F}\_t=\sigma(i\_1,\boldsymbol{x}\_{a\_1},\eta\_1, \dots, i\_{t},\boldsymbol{x}\_{a_t}, \eta_t, i\_{t+1},\boldsymbol{x}\_{a\_{t+1}}),$$
> where $i_{t}$ and $\boldsymbol{x}\_{a_t}$ are $\mathcal{F}\_{t-1}$-measurable, and $\eta_t$ is $\mathcal{F}\_{t}$-measurable. Then we can rewrite the summation in a form that aligns with Theorem 1 of Abbasi-Yadkori et al. (2011):
> $$\sum\_{\tau \in [t]: i\_{\tau}=i} \boldsymbol{x}\_{a\_{\tau}} \eta\_{\tau} = \sum\_{\tau=1}^t \mathbb{1}\_{i\_{\tau}=i}\boldsymbol{x}\_{a\_{\tau}} \eta\_{\tau},$$
> which allows us to directly apply the self-normalized bound.
>
> We will provide more details on this in the revised manuscript.
>
> ---
> We appreciate your valuable review of our paper and are more than happy to answer any further questions.

---

> ### Author Response · Authors · 2024-11-25
>
> Dear Reviewer KU8f,
>
> As the discussion phase is drawing to a close, we kindly ask whether our responses have resolved your concerns or if there are any remaining issues that we can further clarify. Your insights are invaluable in refining our work, and we are eager to ensure that all your concerns are fully addressed.
>
> Thank you once again for your time and effort in reviewing our manuscript.
>
> Best regards,\
> Authors

---

> ### Author Response · Authors · 2024-12-02
>
> Dear Reviewer KU8f,
>
> As today is the final day of the discussion phase, we wanted to kindly follow up on our rebuttal.
>
> You mentioned that your primary concern was whether we overstate the extent to which our work relaxes the assumptions used in prior works. In our rebuttal, we clarified that the assumptions in prior works are **extremely strong** and become **unattainable when the number of arms $K>8$**. We also explained why assuming knowledge of the gap parameter is a reasonable trade-off.
>
> We would greatly appreciate it if you could let us know whether our response resolves your concerns.
>
> Best regards,\
> Authors

---

### Official Review · Reviewer_sepG · 2024-11-12

**Soundness:** 3
**Presentation:** 2
**Contribution:** 3
**Rating:** 6
**Confidence:** 3

**Summary:**

The paper proposes an algorithm for the linear bandits with clustered users, relaxing assumption on the data diversity and achieving less regret incurred by mis-clustering under both stochastic and smoothed adversarial context settings. Empirical performances of the proposed algorithms shows the efficacy and practicality on the real datasets.

**Strengths:**

1. This paper improves the practicality of the algorithms for the clustering bandit problem by relaxing the strong assumption and analyzing the adversarial context setting.
2. Experiments are abundant to validate the performance of the proposed algorithms.

**Weaknesses:**

1. The theoretical analysis seems to have limited novelty and heavily relies on the previous theoretical results.
2. Some parts of the presentation is hard to read, i.e., variables are defined under Table 1, the regret bounds are stated before defining the variables $\tilde{\gamma}$, $u$, and etc...
3. It would be better to prove an $\tilde{O}(T^{2/3})$ lower bound to show the impossibility results when the $\tilde{\gamma}$ is unknown.

**Questions:**

1. Could the authors explain on the novelty of the theoretical analysis?
2. What happens when $\tilde{\gamma}$ is very small but $\gamma$ is large? Can $\tilde{\gamma}$ be estimated in some ways?

---

> ### Author Response · Authors · 2024-11-18
>
> Thank you for your insightful comments! Here are our responses:
>
> ---
> **Q1: The theoretical analysis seems to have limited novelty and heavily relies on the previous theoretical results. Could the authors explain the novelty of the theoretical analysis?**\
> **A1:** While we agree that our theoretical analysis builds upon many ideas from existing studies, we have made significant changes to integrate these ideas into the online clustering of bandits framework. These changes also allow us to relax previous assumptions and achieve improved regret bounds. The novelty of our theoretical analysis mainly lies in the following aspects:
> 1. For the stochastic context setting, we eliminate the strong assumption used in prior studies by introducing a pure exploration phase. The key novelty in our analysis lies in Lemma 2, where we utilize the matrix Chernoff bound to prove that the minimum eigenvalue of the design matrix grows sufficiently fast. This ensures that the algorithm gathers enough statistical information to accurately infer the underlying clusters. This leads to a tighter regret bound for our proposed UniCLUB and UniSCLUB.
> 2. For the smoothed adversarial context setting, the definition of Gaussian perturbation is the same as Kannan et al. (2018). However, the analysis of Kannan et al. (2018) focuses on how greedy strategies behave in smoothed contexts, while we adapt the analysis to handle UCB strategies in Lemmas 10 and 11. This also leads to a tighter regret bound for our proposed SACLUB and SASCLUB.
> 3. For the case when $\widetilde{\gamma}$ is unknown, our algorithm Phase-UniCLUB effectively mitigates the regret caused by potential misclusterings. The key innovation lies in the carefully designed phase lengths and the incorporation of uniform exploration within each phase. This uniform exploration allows us to estimate users' preference vectors to a certain accuracy, ensuring that users within a cluster have true preference vectors that are close to each other, bounding their 2-norm distance by $\gamma_s$ (Lemma 8). With our refined analysis, we can bound the regret caused by misclusterings and achieve a sublinear regret of $\widetilde{O}(T^{\frac{2}{3}})$ (Theorem 3). Without these components, the regret caused by misclusterings would **grow linearly with $T$**, as shown in the second term of Equation (7) in existing work [1].
>
> [1] Wang Z, Xie J, Liu X, et al. Online clustering of bandits with misspecified user models. NeurIPS 2023.
>
> ---
> **Q2: Some parts of the presentation is hard to read, i.e., variables are defined under Table 1, the regret bounds are stated before defining the variables $\widetilde{\gamma}$, $u$, and etc...**\
> **A2:** Thank you for pointing this out. We stated the regret bound in the introduction before elaborating on the definition of each variable to give experienced readers a quick overview of our contributions. For readers unfamiliar with this topic, we have also provided references to the detailed definitions of each variable in subsequent sections under Table 1.
>
> We understand that this approach might make the manuscript harder to read for some. We will continue to polish the manuscript to ensure that both readers familiar with the topic and those new to it can follow the presentation more easily.
>
> ---
> **Q3: It would be better to prove an $\widetilde{O}(T^{2/3})$ lower bound to show the impossibility results when $\widetilde{\gamma}$ is unknown.**\
> **A3:** Thank you for your insightful suggestion. We completely agree that establishing a $\widetilde{O}(T^{2/3})$ lower bound would strengthen the understanding of the problem. However, currently, we are unable to prove such a lower bound. The lower bound for the online clustering of bandits problem when $\widetilde{\gamma}$ is unknown remains an open challenge in the field. We acknowledge the significance of this issue and consider it a valuable direction for future research.
>
> ---
> **Q4: What happens when $\widetilde{\gamma}$ is very small but $\gamma$ is large? Can $\widetilde{\gamma}$ be estimated in some ways?**\
> **A4:** If $\widetilde{\gamma}$ is very small but $\gamma$ is large, our algorithms UniCLUB and UniSCLUB would incur a larger regret due to unnecessary exploration when learning the underlying clusters. However,
> 1. From the theoretical perspective, since $\widetilde{\gamma}$ is treated as a constant in our theoretical analysis, a small value does not affect the asymptotic order of the regret bound stated in Theorems 1 and 2.
> 2. From the practical perspective, as discussed in Remark 1 (Lines 230-238), $\gamma$ is usually a predefined parameter used to determine whether two items belong to the same cluster. So its lower bound, $\widetilde{\gamma}$, is intrinsically known. If $\widetilde{\gamma}$ is not directly known, it can be estimated, for example, from historical data.
>
> ---
> We appreciate your valuable review of our paper and are more than happy to answer any further questions.

---

> > ### Comment · Reviewer_sepG · 2024-11-25
> >
> > Thank you for the response. I would like to keep my score as is.

---

### Author Response · Authors · 2024-11-24
**Revised Manuscript**

Dear Reviewers,

We have revised our manuscript in response to your valuable comments and have uploaded the updated version for your review. All changes have been highlighted in red for your convenience. Below is a summary of the updates:

1. **Reviewer sepG:**\
(Lines 113-114) In the introduction, we emphasize the variables defined below Table 1 and refer readers to the detailed definitions in Section 3.1.

2. **Reviewer KU8f:**\
(Lines 754-755) We elaborate on the filtration definition and clarify how the self-normalized bound is applied to the first term in Equation 1.

3. **Reviewer BvdA:**\
(Line 216) We explicitly clarify which variables the expectation is taken over in the regret definition.

4. **Reviewer wksx:**\
(Lines 464-468, 1794-1847) We include additional experiments to compare our algorithms with GOB.Lin and GraphUCB, as suggested.

We hope these revisions address your comments and improve the manuscript. Please let us know if further clarification or adjustments are needed.

Thank you again for your constructive feedback.

Best regards,\
Authors

---

### Meta-Review · Area_Chair_ohGJ · 2024-12-24

**Metareview:**

This paper aims to relax the assumption about the "stochastic/diverse" context commonly used in existing online clustering of bandits literature. The authors consider stochastic context (similar setting as Gentile, et al. (2014) with different assumptions on context diversity and cluster gap) and smoothed adversary (more relaxed setting assuming Gaussian perturbation, same as Kannan et al. (2018)), The authors proposed new algorithms for these settings and analyzed their regrets.  The reviewers recognized the following strengths:
- Relaxing the stochastic/diverse context assumption is well-motivated. The most commonly used setting of contextual/linear bandits is adversarial context. Stochastic context has been used in literature to conquer bottlenecks in theoretical analysis but is often considered too strong and hard to achieve in practice.
- Theoretical results are sound.

The reviewers mainly raised these concerns:
- Technical novelty and challenges are not clear in both settings (Reviewer sepG, KU8f)
- Contribution regarding the relaxed assumption might be misleading or overstated (Reviewer KU8f, BvdA)
- Regret bound is not tight (Reviewer sepG, BvdA)

There are other concerns regarding experiments and presentation, which are addressed during rebuttal. Overall, this is a borderline paper with mixed opinions: all reviewers agree with the importance of the problem (relaxing the assumption for online clustering of bandits) which the AC fully agrees with, but two reviewers remain negative opinions. AC evaluates each concern carefully (see below for detailed comments on each weakness) and recommends acceptance. However, careful revision is needed to clarify the novelty and contributions, avoiding concerns about misleading or overstated claims.

**Additional Comments On Reviewer Discussion:**

Regarding the concerns:
- Technical novelty and challenges are not clear in both settings (Reviewer sepG, KU8f). The authors explained the novelty in both stochastic context and smoothed adversary settings. AC believes this concern is partially resolved: for stochastic context, the novelty lies in identifying relaxed assumptions while the proof technique is rather straightforward; for smoothed adversarial context, the settings and fundamental techniques follow Kannan et al. (2018) and proofs are modified for the UCB-based strategy instead of greedy method.
- Contribution regarding the relaxed assumption might be misleading or overstating (Reviewer KU8f, BvdA). The concern is also partially addressed. The authors argued that assumptions in previous work are too strong. However, Reviewer KU8f's question that adding the cluster gap parameter as a new assumption is also strong, is not addressed. AC believes that studying the two settings are small but meaningful step towards the most commonly studied setting without any assumption: adversarial context.  Since the paper is titled "Demystifying Online Clustering of Bandits", it is important to clarify the assumption and contribution in each setting to avoid concerns about misleading or overstated claims.
- Regret bound is not tight (Reviewer sepG, BvdA). This is a difficult technical challenge that would be good to solve. However, this should not be the reason to reject the paper.

---

### Decision · Program_Chairs · 2025-01-22

Accept (Poster)